# Intratumoral delivery of FLT3L with CXCR3/CCR5 ligands promotes XCR1+ cDC1 infiltration and activates anti-tumor immunity

Louise Gorline [1,2,3,16], Fillipe Luiz Rosa do Carmo[1,2,3,16], Pierre Bourdely [4,5], Jérémie Bornères[1,2,3], Nathan Vaudiau [1,2,3], Aurélie Semervil [1,2,3,5], Mathias Vetillard[6], Aboubacar Sidiki K Coulibaly[1,2,3], Natacha Jugniot[1,2,3], Agathe Ok [6], Mathilde Bausart[7], Oriane Fiquet[8], Marine Andrade[8], Dicken Fardol [9], Ikrame Haddar[1,2,3], Zeina Abou Nader[5], Judith Weber [5], Hannah Theobald[10], Matthieu Collin[11], Joseph Calmette[11], Giorgio Anselmi [12], Flavia Fico[7], Florent Ginhoux[10], Laleh Majlessi [13], Emmanuel L. Gautier[9], Loredana Saveanu [6], Julie Helft [5], Marc Dalod [14], Mathilde Dusseaux [8], James P. Di Santo [3,15], Stéphanie Hugues[7] & Pierre Guermonprez [1,2,3] ✉

Tumor infiltration by XCR1+ conventional dendritic cells (cDC1) correlates strongly with favorable prognosis and improved responses to immunotherapy. Yet, tumor-driven immunosuppressive programs restrict efficient cDC1 recruitment, highlighting the need for strategies to increase cDC1 access to the tumor microenvironment. Here, we establish a proof-of-concept cell-based immunotherapy that enhances the infiltration of circulating cDC1 progenitors and supports their local expansion. Intratumoral engraftment of autologous mesenchymal stromal cells engineered to express membrane bound FLT3L promotes cDC1 recruitment when combined with poly(I:C). We identify poly(I:C)-induced CXCL9 and CCL5 as essential chemokines controlling intratumoral cDC1 infiltration. Stromal cell−mediated local delivery of FLT3L together with CXCL9 and CCL5 is sufficient to enhance cDC1 infiltration in mice or humanized mice settings. Finally, this approach activates antitumor immunity and partially overcomes resistance to immune checkpoint blockade. Collectively, our data support the therapeutic potential of expanding intratumoral cDC1s through local and sustained delivery of FLT3L, CXCL9, and CCL5.

An important factor limiting the development of anti-tumor immunity lies in the extent of tumor infiltration by immune cells, including dendritic cells (DCs). DCs are sentinel cells of the immune system initiating the tumor-immunity cycle[1,2].

Tumor-associated DCs (TADCs) up-take antigens by phagocytosis and migrate to tumor draining lymph nodes (tdLNs) after up-regulating CCR7. If engaged in immunogenic maturation by the activation of proper innate signalling pathways, DCs can prime anti-tumor CD4+ and CD8+ T cells in tdLNs[2,3]. Once activated, effector T lymphocytes can migrate back to tumors via efferent lymph and then blood[2]. All these activities define DCs as initiators of the cancer-immunity cycle.

In addition to this virtuous cycle of T cell activation in tdLN, intratumoral DCs also have local roles. Indeed, they participate to the recruitment of CXCR3[+] effector T cells via the release of CXCL9 chemokines[4–6]. Also, intratumoral DCs shape niches supporting the survival of tumor-infiltrating T cells (TILs) by delivering specific trophic cues. For instance, CXCL16[+] TADCs attract CXCR6[+]TCF1[−] effector and promote their survival by trans-presenting IL15[7]. DCs establish local interactions with CD28[+] T cells and preserve their fitness[8]. Finally, CCR7[+] LAMP3[+] DCs also populate Tertiary Lymphoid Structures[9], some "stem immunity hubs" related-lymphoid aggregates where they colocalize with TCF1[+]CD8[+] TILs[6,10] and establish some tripartite interactions with CXCL13[+]CD4[+] T cells and TCF1[+]CD8[+] TILs[11].

DCs are classified into two major subsets: XCR1[+] type 1 conventional DCs (cDC1s), which express IRF8 and an heterogenous population of CD11b[+] DCs expressing IRF4 and encompassing cDC2s and DC3s. cDC1s play a non-redundant role in the rejection of immunogenic tumors[12] and are essential for the efficacy of immune checkpoint blockade (ICB) therapies, as demonstrated in pre-clinical models[13,14]. Furthermore, cDC1 infiltration in human tumors correlates with favorable clinical outcomes and response to ICB[15,16]. Mechanistically, cDC1 would fine tune the CD8[+] T cell response by maintaining the pool of TCF1[+] TILs along tumor development thereby fueling a reservoir of T cells potentially promoting effector anti-tumor responses[17]. However, cDC1 infiltration within tumors remains generally limited[18–20]. COX enzymes in tumor cells limit the activation of NK-dependent DCs recruitment[16,21]. Also, oncogenic pathways such as β-catenin limit the recruitment of cDC1s by limiting CCL4 production[19]. Therefore, enhancing the recruitment and activation of intratumoral cDC1s remains an unmet therapeutic need to overcome cancer-induced immune suppression.

Mechanistically, recruitment of circulating DCs have been investigated in the context of pre-clinical models of immunogenic tumors[15,19,21]. Böttcher et al. have identified that cDC1 recruitment depends on NK-cell derived CCR5 ligands (CCL4, CCL5), as well as XCR1 ligands (XCL1, XCL2)[21]. It is unclear if infection activated CCR2-dependent pathways[22] also control cDC1 recruitment.

The FLT3L-FLT3 signaling pathway is essential for the homeostasis of conventional DCs, including cDC1s, in both mice[23,24] and humans[25]. Intratumoral FLT3L is an important biomarker associated to the density of TADCs, particularly cDC1s, in human melanoma[15]. FLT3L-FLT3 pathway supports DC development and maintenance in both lymphoid organs and peripheral tissues[24]. Tissue FLT3L supports the proliferation of DC precursors (pre-DCs) in bone marrow and peripheral sites[24,26]. Of note, radio-resistant cells control homeostatic levels of FLT3L[27]. Within secondary lymphoid organs, stromal cells contribute to DC pool maintenance. For example, a CCL19[+]GREMLIN[+] fibroblastic niche support DC survival at homeostasis[28] via FLT3L production[29]. However, the cellular sources of FLT3L within tumors are not very well known. Barry et al. have shown that NK cells represent a substantial source of FLT3L[15] but other sources like T cells[30] or mast cells[27] might participate to local FLT3L production.

Administration of recombinant soluble FLT3L is an effective approach to expand circulating DCs[31] in humans but it faces questionable safety issues with general modification of haematopoiesis and expansion of tolerogenic DCs[32]. Furthermore, systemic recombinant FLT3L might promote immune tolerance, as it leads to the expansion of regulatory T cells[33,34]. Lastly, combination therapies of FLT3L with innate agonists such as anti-CD40[20] or poly(I:C)[14] raise some toxicity concerns associated to systemic inflammation.

In this work, we have reasoned that targeted, local interventions in the tumor microenvironment (TME) could optimize local generation of immunogenic DCs[35], while avoiding most systemic effects underpinning immune regulation or toxicity. Here, we explore the therapeutic potential of delivering engineered autologous stromal cells within the TME. Engrafted stromal cells are expressing chemotactic factors supporting DC infiltration, differentiation, proliferation, and maintenance. We identify that stromal membrane bound form of FLT3L combined with poly(I:C) or CXCL9 and CCL5 is sufficient to activate the local recruitment and accrual of cDC1s, T and NK/ILC1 cells within tumors. Furthermore, we show that these immunotherapies stimulate anti-tumor immunity and partially overcome resistance to ICB. Altogether, these findings open an avenue to rewire the tumor microenvironment by local cell-based therapeutic intervention.

## Results

### Intratumoral delivery of engineered mesenchymal stromal cells expressing membrane bound FLT3L does not alter intratumoral and systemic cDC1 populations

We have previously demonstrated that engineered stromal cells expressing human FLT3L act as "synthetic niches" able to support the differentiation of human tissue DCs in immunodeficient mice[36]. In a first attempt to stimulate intratumoral cDC accumulation, we generated stable, autologous, fibroblastic cell lines from embryonic fibroblasts of C57BL/6 mice. These cells are characterized as CD45[−]CD31[−]CD44[+]CD140a[+]gp38/PDPN[+] (Supplementary Fig. 1A) and termed as engineered mesenchymal stromal cells thereafter (eMSC). eMSC-FLT3L express homogeneously the membrane form of human FLT3L, as detected by flow cytometry (Fig. 1A). We found that eMSC-FLT3L but not eMSC-control (eMSC-ctrl) stimulated the differentiation of cDC1s and cDC2s, when co-injected with CD45.1 bone-marrow hematopoietic stem cell progenitors (HSPCs) in basal membrane extract plugs in the dermis of tumor free mice, showing that FLT3L expressed in eMSC-FLT3L was functional (Fig. 1B, C and Supplementary Fig. 1B). On the contrary, recombinant FLT3L (rec-FLT3L) did not stimulate cDC differentiation within the plugs. We next decided to inject eMSC-FLT3L within subcutaneous B16 melanoma tumors expressing ovalbumin (B16-OVA), which have a primary resistance to immune checkpoint blockade (ICB). First, we found that gp38[+]GFP[+] eMSC-FLT3L were still detectable on site 14 days after intratumoral injection, validating their persistence within the TME (Fig. 1D, E). To further evaluate the long-term biodistribution and persistence of eMSCs, we engineered nano-luciferase-expressing eMSCs and monitored their persistence using in vivo imaging for 8 days post engraftment in the B16-OVA model (Fig. 1F). We observed that eMSCs remained within the tumor for at least 8 days (Fig. 1G), with their activity confined to the tumor site (Fig. 1H). These observations indicate that, once injected, eMSCs stay localized within the tumor and do not disseminate systemically. We next compared intratumoral engraftment of eMSC-FLT3L to B16F10 cells overexpressing human FLT3L (B16-huFLT3L) and assessed their impact on cDC differentiation. In contrast with B16-huFLT3L, B16F10 tumor bearing mice that had received intratumoral engraftment of eMSC-FLT3L at day 7 did not display an elevation of systemic FLT3L serum levels at day 12 (Fig. 1I, J). Similar results were obtained using the B16-OVA model (Supplementary Fig. 1C). However, we did not observe an increase of cDC1s in the tumor (Fig. 1K), the tumor draining lymph node (tdLN) (Fig. 1L) nor in the spleen (Fig. 1M).

We conclude that intratumoral injection of eMSC-FLT3L alone does not alter intratumoral and systemic cDC1 populations.

### Intratumoral delivery of eMSC-FLT3L synergizes with poly(I:C) to activate anti-tumor immunity

We next wondered if innate agonist administration would circumvent the inability of eMSC-FLT3L to expand intratumoral DCs. We chose to use poly(I:C) because it had previously been shown to stimulate anti-tumor immunity in combination with recombinant, soluble FLT3L[14]. We designed our intratumoral cell-based therapy as two injections of eMSC-FLT3L and one injection of poly(I:C) within B16-OVA tumors (Fig. 2A). We found that only eMSC-FLT3L in combination with poly(I:C) induced tumor regression as assessed by the tumor growth

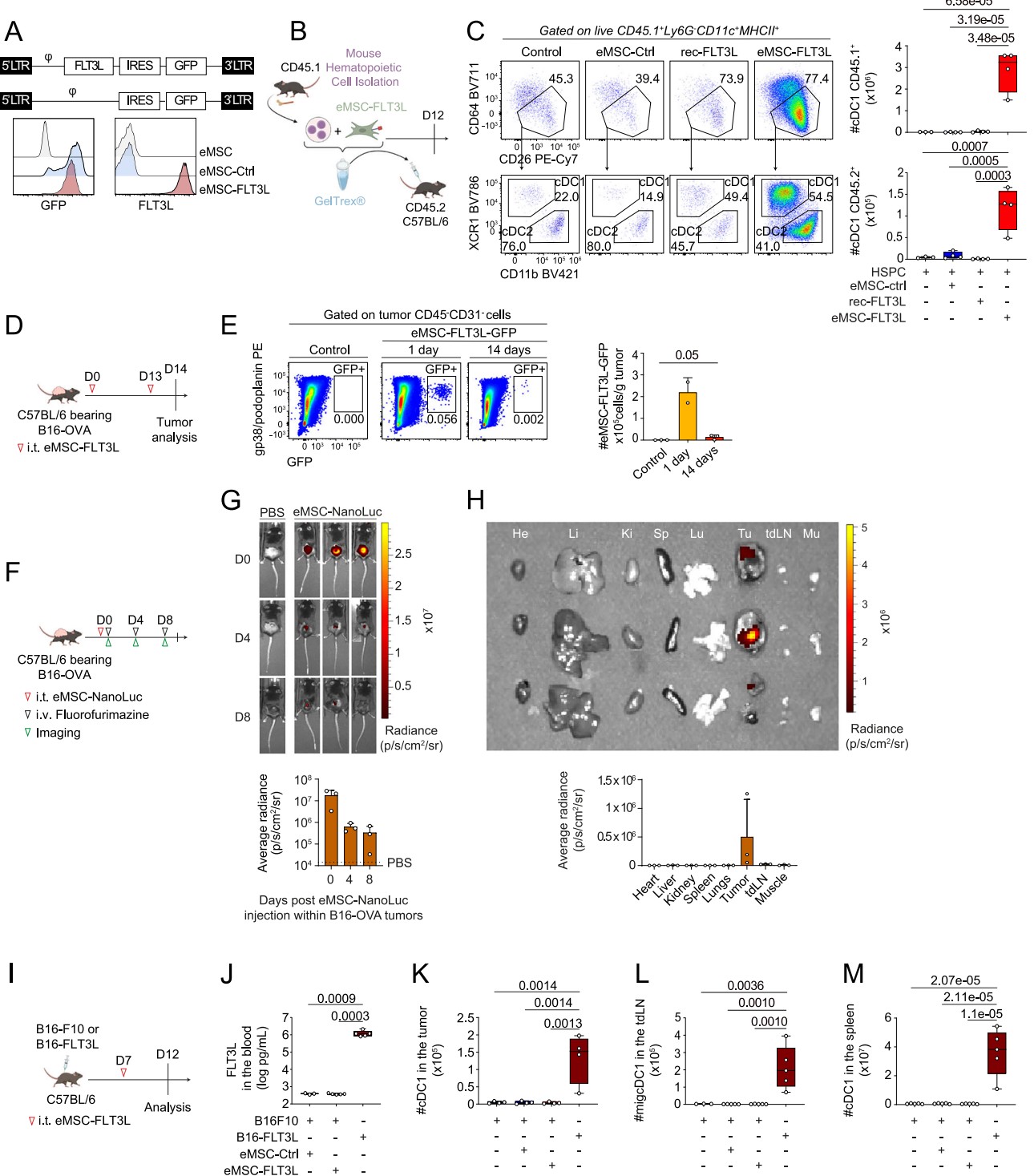

(Fig. 2B) and the tumor weight (Fig. 2C). In addition, the combination therapy extended the survival of mice. Importantly, 10% of the mice treated with the combination therapy eliminated the tumor without relapses until day 90 (Fig. 2D). Interestingly, intratumoral administration of recombinant FLT3L in combination with poly(I:C) also delayed tumor growth (Supplementary Fig. 2A) and improved survival (Supplementary Fig. 2B) but less efficiently than eMSC-FLT3L. Furthermore, mice treated with the therapy did not exhibit weight loss, unlike those receiving anti-CD40 treatment as reported elsewhere[37–39] (Fig. 2E, F). In addition, serum levels of ALT (Fig. 2G) and AST (Supplementary Fig. 2C), markers of hepatocellular injury, remained unchanged in mice treated with eMSC-FLT3L + poly(I:C) compared to the anti-CD40

group. Furthermore, mice having received the anti-CD40 antibody exhibited necrotic lesions in the liver (Supplementary Fig. 2D). These results demonstrate the lack of noticeable toxicity associated to immunotherapy by eMSC-FLT3L + poly(I:C) engraftment. In a clinical translational point of view, we asked if adult, autologous fibroblasts would represent a suitable option for eMSC-based immunotherapy. To test this, we generated a CD45⁻CD31⁻ stromal fraction from adult murine visceral and inguinal adipose tissues. We transduced them to express membrane bound human FLT3L (eAMSC-FLT3L) (Fig. 2H). Following the same experimental plan as previously described (Fig. 2A), we found that eAMSC-FLT3L displayed similar efficiency as eMSCs-FLT3L, when combined with poly(I:C), as showed by the

**Fig. 1 | Intratumoral delivery of engineered mesenchymal stromal cells expressing membrane bound FLT3L does not alter intratumoral and systemic cDC1 populations. A** Expression of GFP (eMSC-ctrl) and human membrane-bound FLT3 ligand in mouse embryonic fibroblasts (eMSC-FLT3L). φ: packaging signal essential for retroviral genome packaging; IRES internal ribosome entry site. **B** Experimental design to assess the ability of eMSC-FLT3L to drive DC differentiation in vivo. **C** Representative flow cytometry plots and quantification at day 12 of CD45.1 and CD45.2 cDC1s in the synthetic niches containing HSPC only (n = 3 plugs) or with eMSC-FLT3L, eMSC-ctrl or rec-FLT3L (n = 4 plugs), one experiment, one-way ANOVA-test with Tukey's multiple comparisons. **D** Experimental design to evaluate the persistence of eMSC-FLT3L within B16-OVA tumors. **E** Representative flow cytometry plots and quantification of the absolute number of eMSC-FLT3L-GFP/g tumor, 1 day (n = 2 mice) and 14 days (n = 3 mice) after eMSC-FLT3L-GFP injection, one experiment, one-tailed Mann–Whitney test. **F** Experimental design to evaluate the biodistribution and persistence of eMSCs within B16-OVA tumors. **G** Bioluminescence imaging and quantification of the average radiance of eMSC-NanoLuc at days 0, 4 and 8 following their injection, n = 3 mice per group, one experiment. **H** Bioluminescence imaging and quantification of the average radiance of eMSC-NanoLuc in individual organs of mice having received eMSC-NanoLuc, n = 3 mice per group, one experiment. He heart; Li liver; Ki kidney, Sp spleen, Lu lungs, Tu tumor; tdLN tumor-draining lymph node, Mu muscle. **I** Experimental design to evaluate cDC1s in tumors, tumor-draining lymph nodes and spleens. **J** Circulating huFLT3L levels measured by ELISA at day 12 in mice bearing B16F10 and eMSC-ctrl (n = 3), B16F10 and eMSC-FLT3L (n = 5) and B16-FLT3L (n = 5), one experiment, one-way ANOVA-test with Tukey's multiple comparisons. Quantification of the absolute numbers of cDC1s in the tumors (**K**) (n = 4 mice), in the skin tumor-draining lymph nodes (tdLN) (**L**) (n = 3 mice for B16F10 and n = 5 mice for the others) and in the spleens (**M**) (n = 5 mice), one experiment, one-way ANOVA-test with Tukey's multiple comparisons. **C, J–M** Box plots show median, 25th–75th percentiles, minimum–maximum whiskers, with all data points displayed. **E, G, H** Data are presented as mean values ± SD. **C, E, G, H, J–M** Source data are provided as a Source data file. **B, D, F, I** Created in BioRender. Guermonprez, P. (2025) https://BioRender.com/o76t012. i.t.: intratumoral; i.v. intravenous.

---

reduced tumor growth (Fig. 2I) and tumor weight (Fig. 2J). We conclude that adult fibroblasts are amenable to engineering for local immunotherapy. Lastly, to further validate the therapeutic efficacy of the eMSC-FLT3L + poly(I:C) strategy, we tested it across multiple tumor models, including MC38 colon carcinoma (Fig. 2K–N), TC-1 lung carcinoma transformed with HPV16 E6/E7 (Fig. 2O–R), and the orthotopic E0771 breast carcinoma (Fig. 2S–V). In each model, treatment delayed tumor growth (Fig. 2L, P, T), reduced individual tumor size variation relative to the day of the first intratumoral injection (Fig. 2M, Q, U), and improved overall survival (Fig. 2N, R, V).

Altogether, these experiments show that local cell-based delivery of FLT3L within the TME in combination with poly(I:C) is a potent therapy to control tumor growth in multiple tumor contexts.

## Intratumoral delivery of eMSC-FLT3L and poly(I:C) stimulates the infiltration of cDC1s, that are required for immunotherapy efficiency

Having established the immunotherapeutic potential of eMSC-FLT3L + poly(I:C) combination, we next sought to address if this intervention would modify cDC infiltration in the TME. To do so, we analysed intratumoral cDCs 24 h and 6 days after a poly(I:C) injection (Fig. 3A). We observed that 24 hours post poly(I:C) injection (day 10), tumor cDC1s underwent phenotypic maturation, as probed by CD40 and CD86 up-regulation within both eMSC-Ctrl and eMSC-FLT3L deliveries (Fig. 3B, C). Poly(I:C) injection also induced a decrease in the number of cDC1s in both eMSC-Ctrl and eMSC-FLT3L treated tumors (Fig. 3C). In contrast, 6 days post poly(I:C) injection (day 15) cDC1s underwent expansion in mice treated with both poly(I:C) and eMSC-FLT3L but not in mice treated with poly(I:C) or eMSC-FLT3L alone (Fig. 3C). This shows that sustained cDC1 expansion at the tumor site, uniquely achieved by the synergy of eMSC-FLT3L and poly(I:C), correlates with the efficiency of the therapy. Of note, similar results were found for cDC2s (Supplementary Fig. 3A). In order to test the role of cDC1s in the efficiency of the therapy, we injected B16-OVA in WT and in cDC1-deficient mice (Xcr1^cre × Rosa^lsl-DTA (Xcr1^DTA)[40]) and treated the tumors with eMSC-FLT3L and poly(I:C) immunotherapy. We found that the tumor control (Fig. 3D) and the survival extension (Fig. 3E) provided by the therapy, were totally abrogated in the absence of cDC1s.

We conclude that the synergy between eMSC-FLT3L and poly(I:C) expands intratumoral cDC1s, that are required for the therapeutic benefits of this intervention.

## Intratumoral delivery of eMSC-FLT3L and poly(I:C) increase mature migratory cDC1s in tumor-draining lymph nodes

Having identified cDC1s anti-tumor role in our therapeutic approach, we next wondered if this effect relied on tumor-draining lymph nodes (tdLNs). As previously described, we analysed the tdLNs 24 h (day 10) and 6 days (day 15) after the poly(I:C) administration (Fig. 4A). At 24 h post poly(I:C) injection, we observed an increase in the migration of mature (CD40^high, CD86^high) cDC1s in the tdLNs of tumors treated with eMSC-FLT3L and poly(I:C) (Fig. 4B, C). However, 6 days after poly(I:C) injection, no significant differences were observed between groups in either the number of migratory cDC1s or their maturation status, suggesting that the effect of poly(I:C) is transient (Fig. 4C). Therefore, and in correlation with the previous results (Fig. 3C), we conclude that eMSC-FLT3L combined with poly(I:C) initially promotes the migration of mature cDC1s to the tdLN, followed by the maintenance of a pool of immature cDC1s within the tumor.

## Intratumoral delivery of eMSC-FLT3L and poly(I:C) enables cross-priming of tumor-specific CD8+ T cells

Since cDC1s at day 10 in the tdLNs are mature and that these cells play a crucial role in the therapy, we investigated whether there was an induction of the adaptive immune response in the tdLN. At day 15, we found that the absolute number of activated CD8+CD44+ T cells (T_ACT), exhausted CD8+CD44+Tim-3+PD-1+ T cells (T_EX) and stem-like CD8+CD44+Tim-3^PD-1+Slamf6+ T cells (T_SL) significantly increased upon eMSC-FLT3L and poly(I:C) therapy (Fig. 4D, E). This effect was associated with an increase in proliferation, as probed by Ki67 staining, in both TCF1+CD44+ stem-like CD8+ and more differentiated TCF1^−TIM3+ T cells (Supplementary Fig. 4A, B). Functionally, CD8+ T cells increased their capacity to secrete IFN-γ and effector to T regulatory cells (Tregs) ratio increased (Fig. 4F, G and Supplementary Fig. 4C, D). Interestingly, we observed a significant increase in the absolute number of CD4+Foxp3^− cells but not in Tregs (Supplementary Fig. 4D, E). Furthermore, CD4 + Foxp3-cells of tumors treated with eMSC-FLT3L and poly(I:C) increased their secretion of IFN-γ (Supplementary Fig. 4F). Finally, the eMSC-FLT3L and poly(I:C) immunotherapy induced the expansion of tumor specific CD8+ T cells that recognize the OVA_{257–264} immunodominant peptide, as shown with the tetramer staining (Fig. 4H, I). Altogether, we conclude that the therapeutic efficiency of eMSC-FLT3L and poly(I:C) associates with increased cross-priming of anti-tumor CD8+ T cells.

## Intratumoral delivery of eMSC-FLT3L and poly(I:C) stimulates the infiltration of T and NK cells within tumor beds, that are required for immunotherapy efficiency

We next wondered if eMSC-FLT3L and poly(I:C) immunotherapy had an impact on the infiltration of intratumoral lymphocytes. First, we characterized lymphoid infiltrates at day 15 within B16-OVA tumors, treated with eMSC-FLT3L and poly(I:C) (Fig. 5A). We found that eMSC-FLT3L and poly(I:C) synergized to expand intra-tumoral NK1.1+ conventional NK cells (cNK) and CD62L^−CD49a+ type 1 innate lymphoid cells (ILC1) (Fig. 5B, C). We also observed an increase in the

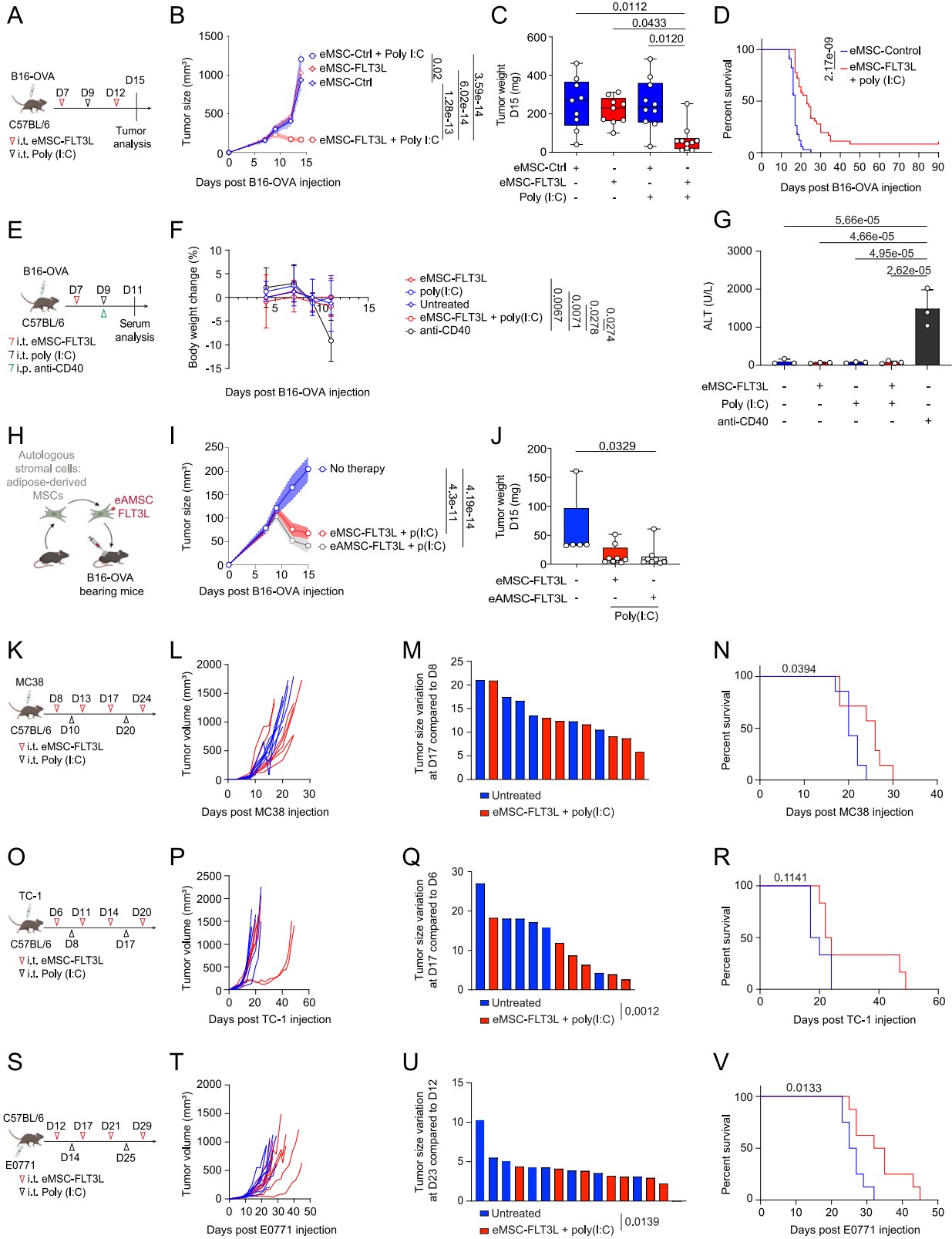

proliferation of these two cell types, as probed by Ki67 staining (Fig. 5B, C). Furthermore, we noticed an increase in intratumoral CD8⁺CD44⁺ activated T cells ($T_{ACT}$) (Fig. 5D, E). By looking closer to the phenotype of these cells, we found that eMSC-FLT3L + poly(I:C) immunotherapy increased both CD8⁺CD44⁺TIM3⁺PD-1⁺ exhausted T cells ($T_{EX}$) and CD8⁺CD44⁺TIM3⁻TCF1⁺ stem-like T cells ($T_{SL}$) (Fig. 5D, E). Of note, the intratumoral abundance of $T_{SL}$ is associated to

responses to immunotherapy in multiple cancer types[41–43]. Interestingly, expansion of $T_{SL}$ cells was associated with a significant increase in cell cycling activity, as probed by Ki67 staining (Supplementary Fig. 5A, B). By contrast, $T_{EX}$ cells displayed a high cycling activity both in control and treated groups. Strikingly, while intratumoral CD4⁺ T cells were increased and showed enhanced proliferation (Supplementary Fig. 5C, D), the abundance and cycling activity of intratumoral

**Fig. 2 | Intratumoral engraftment of autologous mesenchymal stromal cells expressing membrane bound FLT3L stimulates anti-tumor immunity in the presence of poly(I:C). A** Experimental design to evaluate anti-tumoral effects of eMSC-FLT3L + poly(I:C) in B16-OVA model. Tumor growth curves (n = 10 mice) (**B**) and tumor weight (**C**) at day 15. n = 9 (eMSC-ctrl, eMSC-FLT3L), n = 10 (eMSC-ctrl + poly(I:C), eMSC-FLT3L + poly(I:C)) mice per group, two independent experiments, two-way ANOVA-test with Tukey's multiple comparisons, Kruskal–Wallis' test with Dunn's multiple comparisons. **D** Survival curves. n = 34 (eMSC-ctrl), n = 36 (eMSC-FLT3L + poly(I:C)) mice, six independent experiments, log-rank (Mantel–Cox) test. **E** Experimental design to evaluate the immunotherapy's toxicity. **F** Body weight loss of mice throughout the experiment. n = 4 (eMSC-FLT3L + poly(I:C), anti-CD40), n = 3 (other groups) mice per group, two-way ANOVA-test with Tukey's multiple comparisons. **G** ALT level in serum 48 h after poly(I:C) or anti-CD40 treatment, one-way ANOVA-test with Tukey's multiple comparisons. n = 4 mice (eMSC-FLT3L + poly(I:C)), n = 3 mice (other ones) per group, one experiment. **H** Experimental design. Adipocytes precursors (CD45-CD31-) were isolated from inguinal and visceral fat of C57BL/6 mice and transduced with membrane-bound

FLT3 ligand (eAMSC-FLT3L). These cells were injected as shown in (**A**). **I** Tumor growth curves, two-way ANOVA-test with Dunnett's multiple comparisons. **J** Tumor weight at day 15, Kruskal–Wallis' test with Dunn's multiple comparisons. **I**, **J** n = 5 (no therapy), n = 8 (other groups) mice per group, one experiment. Experimental design to evaluate anti-tumoral effects of eMSC-FLT3L and poly(I:C) in MC38 (**K**), TC-1 (**O**) and E0771 (**S**) tumor models. **L**, **P**, **T** Tumor growth curves showing individual mice. **M**, **Q**, **U** Individual tumor size variation compared to the day of the first eMSC-FLT3L injection. Two-way ANOVA-test with Šídák's multiple comparisons. **N**, **R**, **V** Survival curves. Log-rank (Mantel–Cox) test. **L**, **N** n = 7 mice per group, **M** n = 6 (untreated), n = 7 (eMSC-FLT3L + poly(I:C)) mice, one experiment. **P**, **R** n = 6 mice per group, one experiment. (**T**–**V**) n = 8 mice per group, one experiment. **B**, **I** A line represents the mean, and SEM is shown with the colored area. **C**, **J** Box plots show median, 25th–75th percentiles, minimum–maximum whiskers, with all data points displayed. **F**, **G** Data are presented as mean values ± SD. **B**–**D**, **F**, **G**, **I**, **J**, **L**–**N**, **P**, **R**, **T**–**V** Source data are provided as a Source data file. **A**, **E**, **H**, **K**, **O**, **S** Created in BioRender. Guermonprez, P. (2025) https://BioRender.com/o76t012. i.p. intraperitoneal.

Tregs were not up-regulated following eMSC-FLT3 + poly(I:C) immunotherapy (Supplementary Fig. 5C, D). We next wondered whether these changes would associate to increased tumor-antigen-specific T cell response. We found that eMSC-FLT3L + poly(I:C) immunotherapy induced a striking expansion of tumor specific CD8+ T cells for the OVA$_{257-264}$ peptide, as detected by the tetramer staining (Fig. 5F) and the IFNγ-secreting T cells upon peptide-specific (SIINFEKL) ex vivo restimulation (Fig. 5G). Having established the local infiltration of tumor-specific T cells during the onset of tumor growth, we next wondered if the few survivor mice that had undergone eMSC-FLT3L + poly(I:C) immunotherapy displayed evidence of tumor-specific T cell memory. To test this, we injected the SIINFEKL synthetic peptide in tumor-free or B16-OVA survivor mice (d90 post primary tumor injection) (Fig.5H). We found that this re-challenge induced the expansion of CD8+ T cells specific for the OVA$_{257-264}$-H2K$^b$ complex (Fig. 5I) in inguinal lymph nodes from survivor mice but not from tumor free naïve mice. Furthermore, those CD8+ T cells displayed a circulating effector memory phenotype (Fig. 5J). We conclude that eMSC-FLT3L + poly(I:C) immunotherapy induces a sustained memory CD8+ T cell response. Finally, in order to probe the requirement for conventional CD4 and CD8 T cells as well as NK cells, we performed eMSC-FLT3L + poly(I:C) immunotherapy in the presence of control, NK1.1 or CD4 and CD8 depleting antibodies, alone or in combinations (Fig. 5K). We found that both NK and CD4/CD8 contributed to the therapeutic effect of eMSC-FLT3L + poly(I:C), as probed by the reversed inhibition of the tumor growth (Fig. 5L) and of the extension of survival (Fig. 5N). Importantly, inhibition of both NK and CD4/CD8 T cells achieved the maximal inhibition of eMSC-FLT3L + poly(I:C) immunotherapy. To address the involvement of CD8+ T cells and to differentiate total CD4+ T cells from Tregs, a second experiment was conducted in which CD8+, total CD4+, and Treg cells were individually depleted. CD8+ T-cell depletion confirmed their critical role, as evidenced by the loss of tumor control (Supplementary Fig. 5E) and reduced survival (Supplementary Fig. 5F). In contrast, depletion of CD4+ T cells or Tregs did not affect tumor growth or survival, indicating that CD8+ T cells are the primary mediators of the anti-tumor response in this context. Altogether, we conclude that eMSC-FLT3L + poly(I:C) immunotherapy activates and relies on both innate and adaptive anti-tumor immune response.

### Intratumoral delivery of eMSC-FLT3L and poly(I:C) overcomes resistance to anti-PD-1/CTLA-4 immune checkpoint blockade
Lastly, we wondered if stimulation of T and NK-dependent anti-tumor immunity by eMSC-FLT3L and poly(I:C) immunotherapy would bypass resistance to immune checkpoint blockade (ICB). To test this, we submitted B16-OVA to aPD-1/aCTLA-4 ICB immunotherapy (Fig. 5N).

We found that ICB slightly expanded the survival of mice as compared to isotype control-treated mice. However, mice treated with both ICB and eMSC-FLT3L + poly(I:C) immunotherapies exhibited a significant reduction of tumor growth (Fig. 5O) and had extended survival as compared to isotype-treated control groups (Fig. 5P). Of note, the combination of both ICB and eMSC-FLT3L + poly(I:C) improved eMSC-FLT3L + poly(I:C) alone in terms of survival, indicating a proper synergistic effect between both interventions.

Altogether, these data indicate that eMSC-FLT3L + poly(I:C) immunotherapy induce T and NK-dependent immunity against B16-OVA tumor and this defines a way to partially overcome resistance to ICB immunotherapy.

### CXCR3 and CCR5 ligands are induced by poly(I:C) and control its impact on cDC1 and T cell infiltration
We next wondered which mechanism induced by poly(I:C) could explain the synergy between eMSC-FLT3L and poly(I:C). Since eMSC-FLT3L alone are able to recapitulate DC development from hematopoietic progenitors when engrafted in the dermis of tumor free mice (Fig. 1C and Supplementary Fig. 1B), we asked whether eMSC-FLT3L would also recapitulate DC development from DC progenitors within the TME. To test this, we engrafted eMSC-FLT3L together with bone marrow hematopoietic stem and progenitor cells (HSPCs), containing DC precursors and obtained from CD45.1 mice, within B16-OVA tumors (Fig. 6A). Three days after the engraftment of HSPCs, we found that CD45.1+ cDC1s had differentiated from progenitors co-engrafted with eMSC-FLT3L but not eMSC-ctrl (Fig. 6B). In the same tumors, eMSC-FLT3L did not increase cDC1s from the recipient mice, consistently with previous results (Fig.6C). We conclude that availability of DC precursors and not specific features of the TME restrict the differentiation of pre-DCs into cDC1s within B16-OVA tumors. We next wondered if poly(I:C) would regulate the infiltration of pre-cDC1s via chemokine regulation. Using a chemokine multiplex array, we found that poly(I:C) up-regulated multiple chemokines within the TME of eMSC-FLT3L-treated tumors (Fig. 6D). This includes CXCL9 and CXCL11, which both act as ligands for the CXCR3 receptor, and CCL5, that acts as ligand for CCR5[44] (Fig. 6E). Notably, poly(I:C) monotherapy also elicited an upregulation of CXCL9, as evidenced by ELISA measurements performed on tumor homogenates (Supplementary Fig. 6A). CXCR3 and CCR5 are found on activated T and NK cells but also on circulating DC precursors[4,45,46]. To test the contribution of these chemokine signalling pathways to the eMSC-FLT3L + poly(I:C) therapy, we treated B16-OVA bearing mice receiving the therapy with CXCR3 and/or CCR5 antagonists (Fig. 6F). We found that CXCR3 and CCR5 inhibition prevented the anti-tumoral impact of eMSC-FLT3L + poly(I:C) immunotherapy, as shown with the tumor

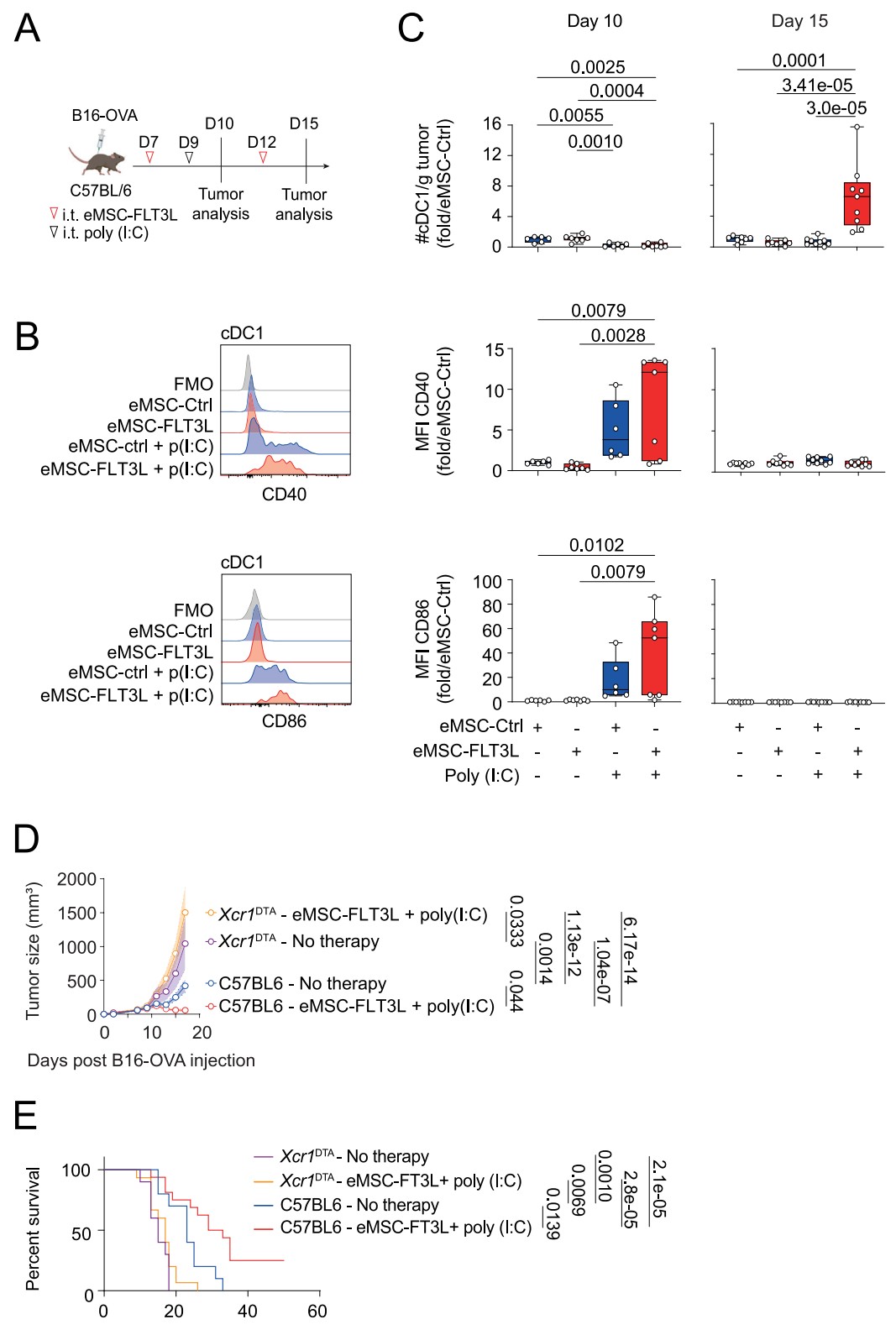

growth (Fig. 6G) and the tumor weight (Fig. 6H). Furthermore, blockade of CXCR3 and/or CCR5 inhibited the accrual of intratumoral cDC1s (Fig. 6I) and the increase of activated CD8[+] T cells within tdLN (Fig. 6J).

Altogether, these data highlight the crucial role of CXCR3 and CCR5 signalling pathways in mediating the immunotherapeutic effect of eMSC-FLT3L + poly(I:C) immunotherapy.

## Intratumoral delivery of eMSC co-expressing FLT3L, CXCL9 and CCL5 stimulates the infiltration of immature pre-cDC1s and cDC1s in WT and BRGSF human immune system (HIS) mice

We next reasoned that stromal cell-based delivery of CXCL9 and/or CCL5 could substitute the effect of poly(I:C) on eMSC-FLT3L therapy. To test this, we produced eMSCs engineered to express CXCL9 and CCL5, either alone or in combination with FLT3L (Fig. 7A). As

**Fig. 3 | Intratumoral delivery of eMSC-FLT3L and poly(I:C) stimulates the infiltration of cDC1s, that are required for immunotherapy efficiency.**
**A** Experimental design to evaluate cDC1 infiltration in B16-OVA tumors. Created in BioRender. Guermonprez, P. (2025) https://BioRender.com/o76t012. Representative flow cytometry plots at day 10 (**B**) and quantification (**C**) of the absolute number of intratumoral cDC1/g tumor and the mean fluorescence intensity (MFI) of CD40 and CD86. Results are shown as fold change to control (eMSC-ctrl). Day 10: n = 6 (eMSC-ctrl, eMSC-ctrl + poly(I:C)), n = 7 (eMSC-FLT3L, eMSC-FLT3L + poly(I:C)) mice per group, two independent experiments, one-way ANOVA-test with Tukey's multiple comparisons. Day 15: n = 8 (eMSC-ctrl, eMSC-FLT3L), n = 9 (eMSC-ctrl + poly(I:C), eMSC-FLT3L + poly(I:C)) mice per group, two independent experiments, one-way ANOVA-test with Tukey's multiple comparisons. Box plots show median, 25th–75th percentiles, minimum–maximum whiskers, with all data points displayed. **D** Tumor growth curves of $Xcr1^{DTA}$ mice or C57BL/6 WT mice treated or not with eMSC-FLT3L + poly(I:C). n = 4 mice $Xcr1^{DTA}$-No therapy and n = 8 mice for the three other groups, representative of two independent experiments, two-way ANOVA-test with Tukey's multiple comparisons. A line represents the mean and SEM is shown with the colored area. **E** Survival curves. n = 10 (C57BL/6-No therapy, $Xcr1^{DTA}$-No therapy), n = 15 ($Xcr1^{DTA}$-eMSC-FLT3L + poly(I:C)), n = 16 (C57BL/6-eMSC-FLT3L + poly(I:C)) mice, two independent experiments, log-rank (Mantel–Cox) test. **C**–**E** Source data are provided as a Source data file.

confirmed by ELISA on culture supernatants, these cells secrete high concentrations of both chemokines when expressed alone or in combination (Fig. 7B, C). Then, we asked whether eMSC-FLT3L-CXCL9-CCL5 would allow the entry of circulating pre-cDC1s within B16-OVA tumors. To test this, we injected IRF8-GFP reporter mice with two tumors on either flank and injected eMSC-FLT3L on one side (left) and eMSC-FLT3L-CXCL9-CCL5 on the other side (right) (Fig. 7D). IRF8-GFP mice are useful to identify IRF8-expressing pre-cDC1s and cDC1s. Twenty four hours after the injection, we found that CCL5 and CXCL9 enriched tumors had an increased infiltration of pre-cDC1s, as defined by Lin- CD45RB⁺FLT3⁺MHCII⁺CD26⁺Ly6C⁻CD172a⁻ XCR1⁻ IRF8^high SIRPa⁻ cells (Fig. 7E). We next wondered if this infiltration of pre-cDC1s would translate into local cDC1 expansion at a later time point and following the same experimental design as previously but substituting poly(I:C) with the engineered cells (Fig. 7F). We found that tumors treated with eMSC-FLT3L-CXCL9-CCL5 had a significant increase in both the absolute number (Fig. 7G) and the frequency (Fig. 7H) of cDC1s compared with untreated tumors or those treated with eMSC-FLT3L alone. Furthermore, we noticed that upon the therapy, tumor cDC1s tend to be more activated, as shown by CD40 and CD86 up-regulation (Supplementary Fig. 7A). Altogether, these data show that stromal cells expressing CXCL9 and CCL5 can substitute the effect of poly(I:C) on cDC recruitment and maturation.

We then wondered whether eMSC-FLT3L-CXCL9-CCL5 co-localized with cDC1s within the tumors and whether they were more effective than eMSC-FLT3L in recruiting cDC1s. We injected eMSC-FLT3L or eMSC-FLT3L-CXCL9-CCL5 in $Xcr1^{cre}xRosa^{LsL-tdTomato}$ mice bearing melanoma tumors. In these mice, cDC1s are genetically labeled with tdTomato fluorescence. First, we found that cDC1s established contact with eMSCs at the border of tumor sites (Supplementary Fig. 7B). Next, we observed that cDC1s were more efficiently attracted by eMSC-FLT3L-CXCL9-CCL5 than by eMSC-FLT3L (Fig. 7I, J), underscoring the critical role of these chemokines in supporting interaction between FLT3⁺ cDC1s and membrane bound FLT3L-expressing eMSCs.

Since CCR5 and CXCR3 are expressed by activated T and NK cells[4], we wondered if the infiltration of pre-cDC1s/cDC1s was indirect, due to the recruitment, at first, of activated lymphoid cells. To test this, we used BRGS mice that are deficient in murine T and B cells ($Rag2^{-/-}$), murine NK cells ($IL2r\gamma_c^{-/-}$)–and macrophage phagocytosis of human cells ($Sirpa^{NOD}$)[47]. BRGS mice were injected subcutaneously with basal membrane extract plugs containing either eMSC-ctrl (left flank), or eMSC-FLT3L-CXCL9-CCL5 (right flank). (Fig. 7K). First, we found that eMSC-FLT3L-CXCL9-CCL5 triggered a local accumulation of cDC1s in the absence of lymphocytes (Fig. 7L). We next wondered if this response was actually dependent on the FLT3 receptor tyrosine kinase that is required for the homeostasis of tissue cDC1s in non-lymphoid tissues[24]. To test this, we used BRGSF mice deficient in FLT3 (BRGSF $Flt3^{-/-}$)[48]. We observed no cDC1 recruitment in mice deficient in tissue cDCs, both in control and treated plugs (Fig. 7L). From these results, we conclude that the recruitment of cDC1 induced by eMSC-FLT3L-CXCL9-CCL5 is independent on B, T and NK/ILCs cells but dependent on FLT3 signalling. Finally, we wondered whether our findings would

be transferable to human settings. To test this in controlled experimental settings, BRGSF mice were engrafted with CD34⁺ HSPCs from human cord blood to create HIS mice. Seventeen-week-old BRGSF HIS mice showed multi-lineage human immune reconstitution comprising B, T, NK and DC. Reconstituted BRGSF HIS mice were injected subcutaneously with basal membrane extract plugs containing either eMSC-Ctrl (left flank) or eMSC-FLT3L-CXCL9-CCL5 (right flank) (Fig. 7M). After 10 days, mice were sacrificed, and the cellular infiltrates populating the plugs in the dermis of the mice were analysed by flow cytometry (Fig. 7N). We found that plugs containing eMSC-FLT3L-CXCL9-CCL5 had a significant increase in human cDC1s, as compared to plugs containing eMSC-ctrl, both in absolute number (Fig. 7O) and frequency (Fig. 7P). Of note, similar trends were observed for human cDC2s (Supplementary Fig. 7C, D).

Altogether, we conclude that the engraftment of eMSC expressing FLT3L, CXCL9 and CCL5 represent an efficient platform to promote the recruitment and expansion of cDC1s within tissues and that the system is amenable to human cDC1s.

## Intratumoral delivery of eMSC co-expressing FLT3L, CXCL9 and CCL5 stimulates the infiltration of activated CD4 and CD8 T cells in the tumor

We next wondered if the secretion of CXCL9 and CCL5 would also impact the attraction of activated T cells. To test this, we first set-up an in vitro migration assay in Boyden chambers, to study the chemo-attraction of activated polyclonal CD4⁺ and CD8⁺ T cells towards eMSCs expressing CCL5 and/or CXCL9 (Fig. 8A). As expected, after 3 hours of migration, we observed a synergistic effect of eMSC-CCL5 and eMSC-CXCL9 in their ability to attract activated CD4⁺ (Fig. 8B) and CD8⁺ T cells (Fig. 8C). We then wanted to know if this attraction could allow the recruitment of activated T cells in B16-OVA tumors. To test this, we injected eMSC expressing CCL5 and/or CXCL9 in the tumors and adoptively transferred ex vivo activated polyclonal CD45.1 T cells intravenously (Fig. 8D). A synergistic effect between eMSC-CCL5 and eMSC-CXCL9 was again observed in their ability to recruit activated CD4⁺ (Fig. 8E) and CD8⁺ (Fig. 8F) CD45.1 T cells in the tumor. In addition, we observed that eMSC-CCL5 and eMSC-CXCL9, either individually or in combination, tended to promote the infiltration of endogenous activated CD44⁺ T cells, particularly the CD44⁺CD8⁺ T-cell subset (Fig. 8G, H). We conclude that eMSC-CXCL9 and eMSC-CCL5 represent a workable platform to deliver locally bioactive chemokines. Lastly, we wanted to assess if the eMSC-FLT3L-CXCL9-CCL5 immunotherapy would have an impact on the infiltration of intratumoral lymphocytes. To test this, we injected eMSC-FLT3L-CXCL9-CCL5 in B16-OVA tumors and looked at a late time point the immune infiltrate (Fig. 8I). We observed that tumors treated with eMSC-FLT3L-CXCL9-CCL5 were more infiltrated with NK, ILC1, CD4⁺ and CD8⁺ T cells (Fig. 8J). Of note, absolute number of activated and exhausted CD8⁺ T cells were also increased in treated tumors.

Altogether, these data show that eMSC-FLT3L-CXCL9-CCL5 immunotherapy induce the recruitment of lymphoid cells in B16-OVA tumors.

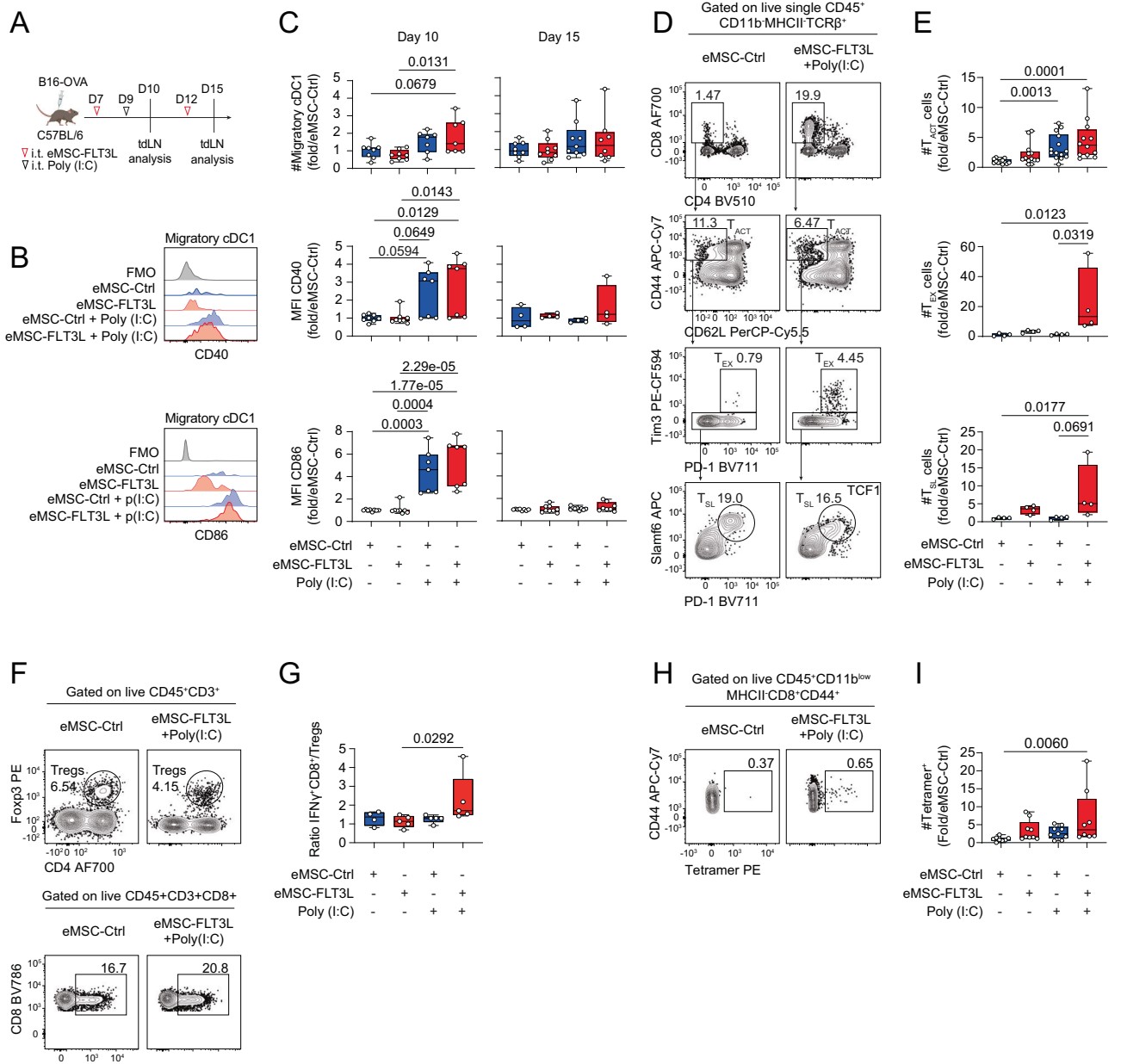

**Fig. 4 | Intratumoral delivery of eMSC-FLT3L and poly(I:C) increase mature migratory cDC1s in tumor-draining lymph nodes and enables cross-priming of tumor-specific CD8+ T cells. A** Experimental design to evaluate migratory cDC1s in the tumor-draining lymph nodes. Created in BioRender. Guermonprez, P. (2025) https://BioRender.com/o76t012. Representative flow cytometry plots at day 10 (**B**) and quantification (**C**) of the absolute number of migratory cDC1s and the mean fluorescence intensity (MFI) of CD40 and CD86. Results are shown as fold change to control (eMSC-ctrl). Day 10: n = 8 (eMSC-ctrl, eMSC-FLT3L), n = 7 (eMSC-ctrl + poly(I:C), eMSC-FLT3L + poly(I:C)) mice per group, two independent experiments, one-way ANOVA-test with Tukey's multiple comparisons. Day 15: n = 8 (eMSC-ctrl, eMSC-FLT3L + poly(I:C)), n = 9 (eMSC-FLT3L, eMSC-ctrl + poly(I:C)), two independent experiments, for the MFI CD40, n = 4 mice per group, representative of two independent experiments, one-way ANOVA-test with Tukey's multiple comparisons. **D** Representative flow cytometry plots of CD8 T cells in the tumor-draining lymph nodes at day 15. **E** Quantification of the absolute number of activated ($T_{ACT}$),

exhausted ($T_{EX}$) and stem-like ($T_{SL}$) CD8 T cells. Results are shown as fold change to control (eMSC-ctrl). $T_{ACT}$: n = 12 (eMSC-ctrl, eMSC-FLT3L + poly(I:C)), n = 13 (eMSC-FLT3L), n = 14 (eMSC-ctrl + poly(I:C)) mice per group, three independent experiments, Kruskal–Wallis test with Dunn's multiple comparisons. $T_{EX}$ and $T_{SL}$: n = 4 mice per group, Kruskal–Wallis test with Dunn's multiple comparisons. Representative flow cytometry plots (**F**) and quantification (**G**) of the ratio of IFNy+CD8+ T cells over T regulatory cells at day 15. n = 4 (eMSC-ctrl), n = 5 (other groups) mice per group, one experiment, Kruskal–Wallis test with Dunn's multiple comparisons. Representative flow cytometry plots (**H**) and absolute number quantification (**I**) of OVA-specific CD8+ Tetramer+ T cells at day 15. Results are shown as fold change to control (eMSC-ctrl). n = 8 (eMSC-ctrl, eMSC-FLT3L + poly(I:C)), n = 9 (eMSC-FLT3L, eMSC-ctrl + poly(I:C)) mice per group, two independent experiments, Kruskal–Wallis test with Dunn's multiple comparisons. **C**, **E**, **G**, **I** Box plots show median, 25th–75th percentiles, minimum–maximum whiskers, with all data points displayed. Source data are provided as a Source data file.

## Intratumoral delivery of eMSC co-expressing FLT3L, CXCL9 and CCL5 recapitulates anti-tumor immunity

Having shown that eMSC-FLT3L-CXCL9-CCL5 immunotherapy had an impact on the infiltration of immune cells in the tumors, we wondered

if it would also confer anti-tumor immunity. To test this, we did three injections of eMSC-FLT3L-CXCL9-CCL5 in B16-OVA tumors (Fig. 9A). We observed a significant reduction in tumor growth (Fig. 9B), tumor size (Fig. 9C) and tumor weight (Fig. 9D) at a late time point.

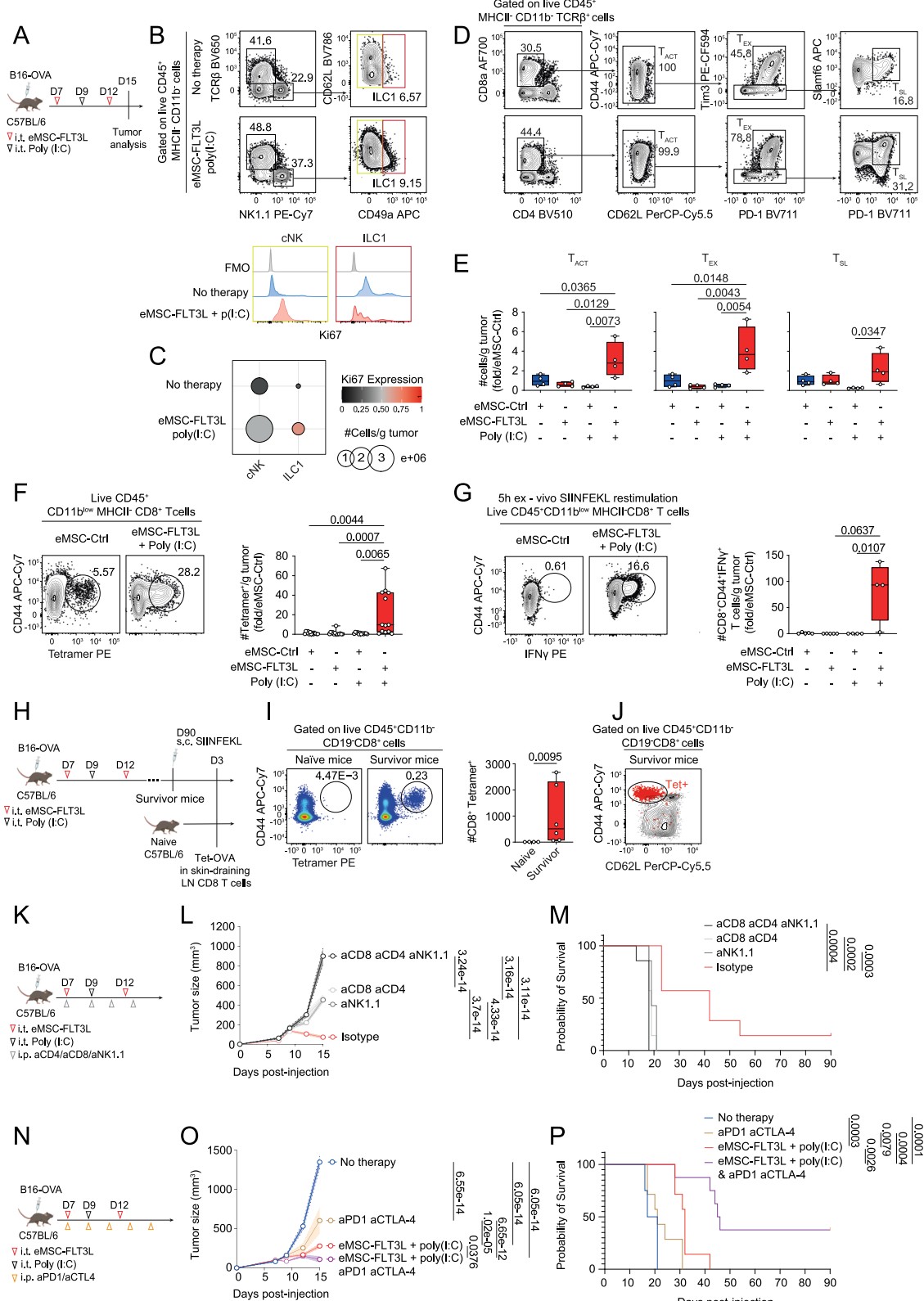

Finally, we wanted to assess if eMSC-FLT3L-CXCL9-CCL5 immunotherapy would also overcome resistance to ICB, by treating the mice with or without aPD-1/aCTLA-4 immunotherapy (Fig. 9E). We found that even if ICB alone reduced the tumor growth (Fig. 9F) and expand the survival of mice (Fig. 9G), the synergy of both eMSC-FLT3L-CXCL9-CCL5 and ICB was needed to eradicate the tumor in a fraction of treated mice (Fig. 9G).

Altogether, these data indicate that eMSC-FLT3L-CXCL9-CCL5 immunotherapy activates anti-tumor immunity and partially overcome resistance to ICB.

## Discussion

In this study, we developed an immunotherapeutic platform based on the intratumoral engraftment of ex vivo−engineered, autologous

**Fig. 5 | Intratumoral delivery of eMSC-FLT3L and poly(I:C) stimulates the infiltration of T and NK cells within tumor beds that are required for immunotherapy efficiency and overcomes primary resistance to PD-1/CTLA-4 blockade. A** Experimental design to evaluate lymphocyte infiltration within tumor. Representative flow cytometry plots (**B**) and quantification (**C**) of NK1.1 and ILC1 subsets. Dots represent the absolute number of cells/g tumor, colors represent the expression level of Ki67. n = 4 mice per group, one experiment. Representative flow cytometry plots (**D**) and quantification (**E**) of activated (T$_{ACT}$), exhausted (T$_{EX}$), and stem-like (T$_{SL}$) CD8$^+$ T cells. n = 4 mice per group, one experiment, one-way ANOVA with Tukey's multiple comparisons test. **F** Representative flow cytometry plots and quantification of OVA-specific CD8$^+$Tetramer$^+$ T cells/g tumor. n = 12 (eMSC-ctrl + poly(I:C)), n = 13 (eMSC-ctrl, eMSC-FLT3L + poly(I:C)), n = 14 (eMSC-FLT3L) mice per group, three independent experiments, Kruskal–Wallis' test with Dunn's multiple comparisons. **G** Representative flow cytometry plots and quantification of IFNγ$^+$CD44$^+$CD8$^+$ T cells after 5 h ex vivo restimulation with SIINFEKL. n = 4 (eMSC-ctrl + poly(I:C), eMSC-FLT3L + poly(I:C)), n = 5 (eMSC-ctrl, eMSC-FLT3L) mice per group, one experiment, Kruskal–Wallis' test with Dunn's multiple comparisons. **H** Experimental design to evaluate T cell memory generation.

**I** Representative flow cytometry plots and quantification of CD44$^+$CD8$^+$Tetramer$^+$ cells in the skin-draining lymph nodes after in vivo SIINFEKL restimulation. n = 4 naïve, n = 6 survivor mice, two-tailed Mann–Whitney test. **J** Representative flow cytometry plots of Tet$^+$CD8$^+$ T cells (red dots) overlaid on total CD8$^+$ T cells of survivor mice. **K** Experimental design to assess the role of CD4, CD8 and NK cells in the anti-tumor effect induced by eMSC-FLT3L + poly(I:C). **L** Tumor growth curves. Two-way ANOVA-test with Tukey's multiple comparisons. **M** Survival curves. Log-rank (Mantel–Cox) test. **L, M** n = 7 mice per group, one experiment. **N** Experimental design to evaluate eMSC-FLT3L + poly(I:C) synergy with immune checkpoint blockade. **O** Tumor growth curves. Two-way ANOVA-test with Tukey's multiple comparisons. **P** Survival curves. Log-rank (Mantel–Cox) test. **O, P** n = 7 (aPD1-aCTL4, eMSC-FLT3L + poly(I:C)), n = 8 (No therapy, eMSC-FLT3L + poly(I:C) + aPD1-aCTLA4) mice per group, one experiment. **E–G** Results are shown as fold change to control (eMSC-ctrl). **E–G, I** Box plots show median, 25th–75th percentiles, minimum–maximum whiskers, with all data points displayed. **L, O** A line represents the mean, and SEM is shown with the colored area. **C, E–G, I, L, M, O, P** Source data are provided as a Source data file. **A, H, K, N** Created in BioRender. Guermonprez, P. (2025) https://BioRender.com/o76t012.

mesenchymal stromal cells (eMSCs). In the first part, we demonstrate a potent therapeutic synergy between eMSCs expressing a membrane-bound form of FLT3 ligand (FLT3L) and the adjuvant poly(I:C). These findings are consistent with previous reports showing that combined administration of recombinant FLT3L and poly(I:C) enhances cDC1 expansion and maturation, promotes tumor regression, and improves responses to immune checkpoint blockade[14,49,50]. However, systemic delivery or secretion of FLT3L has been shown to perturb hematopoiesis and induce peripheral and intratumoral Treg expansion[33,34]. Moreover, the requirement for repeated injections of poly(I:C) and/or FLT3L in these studies raises safety and feasibility concerns for clinical translation. This calls for the development of delivery methods limiting FLT3L protein production to the TME. Here, we show that intratumoral delivery of eMSCs expressing membrane bound FLT3L elicits potent anti-tumor immunity while restricting FLT3L activity to the tumor site, thereby avoiding systemic toxicities. Treated mice exhibited no elevation in serum FLT3L levels or Treg frequencies, and no evidence of weight loss or liver injury, suggesting that this localized, cell-based strategy mitigates systemic adverse effects commonly associated with DC-targeted immunotherapies[38]. Autologous eMSC-FLT3L proposes an alternative option to the intratumoral injection of recombinant FLT3L[14], viral vectors encoding FLT3L[13], CAR-T cell transduced FLT3L[51] and Herpes Virus oncolytic virus encoding FLT3L[52]. Indeed, local retention of eMSC has the advantage to restrict expression of FLT3L to the TME, a feature not afforded by CAR T cell mediated systemic delivery[51]. In addition, our system, in contrast to viral vectors, is not susceptible to immunity.

This study expands the therapeutic scope of engineered mesenchymal stromal cells (eMSCs) to include modulation of the dendritic cell (DC) lineage. Previous studies have shown that IL-2−based, T cell-targeting eMSCs confer substantial benefits by promoting the expansion of non-exhausted PD-1$^+$TIM-3$^-$ CD8$^+$ T cells[53,54]. Likewise, eMSCs engineered to express type I interferons or IL-12 have been reported to enhance the infiltration and cytotoxic effector functions of intratumoral CD8$^+$ T cells[55–58] and NK cells[59]. It remains to be determined whether eMSCs that act directly on lymphocytes could synergize with DC-targeting eMSCs, such as those developed in this study. Exploring such combinatorial strategies may further amplify both antigen presentation and effector responses, providing a versatile platform for next-generation cell-based immunotherapies.

An important finding of this study is that elevation of FLT3L levels alone in the TME is not sufficient to induce DC recruitment and bring immunotherapeutic benefits. We found that poly(I:C) addition synergizes with eMSC-FLT3L. Therefore, we next investigated the mechanisms underlying the enhanced efficacy of the eMSC-FLT3L + poly(I:C)

therapy and identified a marked upregulation of multiple chemokines within the tumor microenvironment (TME), notably CXCL9 and CCL5, which serve as ligands for CXCR3 and CCR5, respectively. Pharmacological blockade experiments demonstrated that both CXCR3 and CCR5 are essential for the intratumoral accumulation of cDC1s and for the establishment of effective antitumor immunity. To minimize the potential toxicities associated with poly(I:C), we replaced poly(I:C) by the co-expression of CXCL9 and CCL5 on eMSC-FLT3L. This substitution similarly enhanced the infiltration of cDC1s, T cells, and NK cells, while maintaining robust antitumor activity. These results align with previous studies highlighting the role of CXCR3- and CCR5-mediated recruitment in sustaining immune effector infiltration into the TME[4,46]. Importantly, although CXCR3 and CCR5 are expressed on circulating pre-DCs[45] as well as on activated T and NK cells, our findings indicate that T and NK cells are dispensable for the CXCL9/CCL5-driven recruitment of DCs. It is known that intratumoral chemokines may derive from multiple sources, including NK cells and cDC1s themselves[4,6,21] and are essential to support the sustained infiltration of cDC1 within the TME. Here, we show that this pathway of cDC1 recruitment associated to immunogenic tumors is activatable in the context of poorly immunogenic tumors by eMSC-FLT3L-CCL5-CXCL9 engraftment. We demonstrate that local chemokine release by eMSC increase the interaction of cDC1s with eMSC-FLT3L. Translatability of the approach to human cDC1s is demonstrated because eMSC-FLT3L-CCL5-CXCL9 treatment supports the local recruitment of human cDC1s in the dermis of BRGSF immunodeficient mice reconstituted with human CD34$^+$ hematopoietic stem cells. However, the identity of the circulating progenitor population responding to this therapy remains to be elucidated, as several candidate precursors have been proposed as human equivalents of murine pre-DCs[31,60,61].

When combining eMSC-FLT3L immunotherapies to aPD-1/aCTLA-4 ICB immunotherapy, we manage to partially overcome primary resistance to immune checkpoint blockade and extend survival. We hypothesize that at least 3 mechanisms could account for this effect of eMSC-FLT3L immunotherapy:

i. eMSCs might activate DCs engaging with poorly differentiated TCF1$^+$ T$_{SL}$ CD8$^+$ T cells enabling to generate ICB-responding pool within the tumors[43,62]. In support of this, multiple studies have identified that mature DCs (and cDC1s in particular) participate to a niche supporting TCF1$^+$ poorly differentiated T cells[6,10,11,17]. This might possibly happen via the accumulation of mature DCs within TLS for instance[63].

ii. eMSC might promote the formation of a pool of activated DCs within the tumor engaging intratumoral effector CD8$^+$ T or NK cells. Garris et al. have shown that ICB rapidly activates

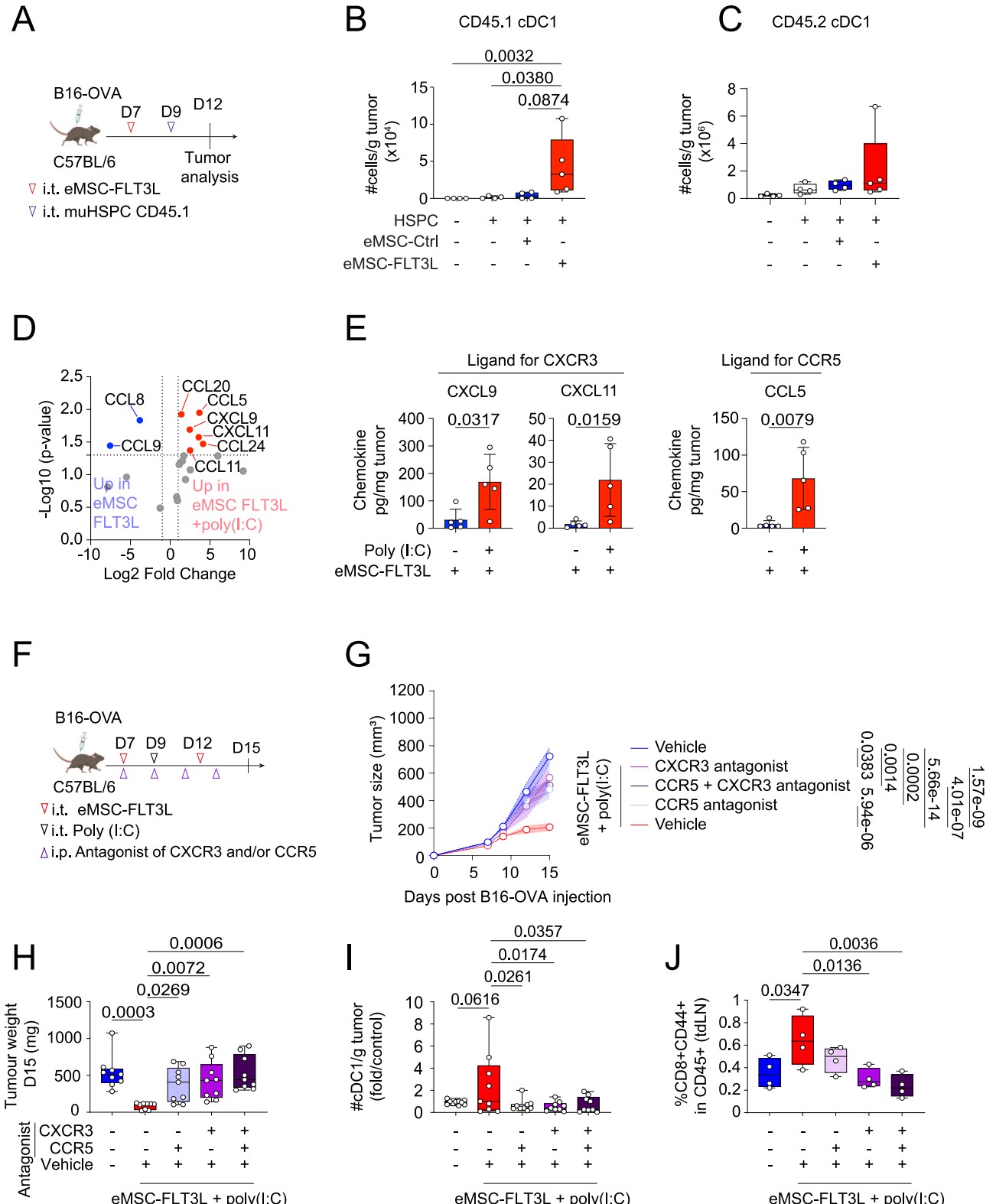

intratumoral DCs via the IFNγR to become IL12/IL15⁺ DCs[64]. By increasing DC density, eMSC could promote the generation of intratumoral IL12/IL15/CXCL16⁺ perivascular DCs, which have been shown to play a key role in maintaining the fitness of intratumoral CXCR6⁺ CD8⁺ T cell effectors[7].

iii. Lastly, eMSC might increase the number of migratory DCs cross-priming novel CD8⁺ T cells in tumor draining lymph nodes. This might help broadening and diversifying T cell response which is a feature of productive ICB immunotherapy[65,66]. In support of this notion, we have evidenced that eMSC-FLT3L + poly(I:C) therapy increases the flux of migratory DCs to tumor-draining lymph nodes.

Altogether, this study provides a proof-of-concept for the implementation of eMSC in tissue engineering approaches purposed to induce anti-tumor immunity via increased DC infiltration within

**Fig. 6 | CXCR3 and CCR5 ligands are induced by poly(I:C) and control its impact on cDC1 and T cell infiltration. A** Experimental design to assess the ability of eMSC-FLT3L to drive DC differentiation in B16-OVA tumors. Quantification of the absolute number of CD45.1 cDC1/g tumor (**B**) and CD45.2 cDC1/g tumor (**C**). n = 5 (HSPC + eMSC-FLT3L), n = 4 (other groups) mice per group, one experiment, Kruskal–Wallis' test with Dunn's multiple comparisons. **D** Volcano plot for the chemokine multiplex array, where each dot represents one gene, plotted based on log2 fold change and −log10 of adj p-value. Genes in blue are up-regulated in tumors treated with eMSC-FLT3L, while genes in red are up-regulated in tumors treated with eMSC-FLT3L + poly(I:C). n = 5 mice per group, one experiment. **E** Individual selected chemokines of interest, based on up-regulated chemokines in eMSC-FLT3L + poly(I:C) found in volcano plot. Values were obtained with a Legendplex. n = 5 mice per group, one experiment, two-tailed Mann–Whitney test. Data are presented as mean values ± SD. **F** Experimental design to block chemokine receptors CXCR3 and/or CCR5. **G** Tumor growth curves. n = 9 (antagonist groups),

n = 10 (untreated), n = 11 (eMSC-FLT3L + poly(I:C)) mice per group, two independent experiments, two-way ANOVA-test with Tukey's multiple comparisons. A line represents the mean and SEM is shown with the colored area. **H** Tumor weight at day 15. n = 9 mice per group, two independent experiments, Kruskal–Wallis test with Dunn's multiple comparisons. **I** Quantification of the absolute number of cDC1/g tumor. Results are shown as fold change to control (untreated). n = 10 (untreated), n = 9 (other groups) mice per group, two independent experiments, one-way ANOVA test with Dunnett's multiple comparisons. **J** Quantification of the frequency of activated CD8+ T cells in live CD45+ cells in the tumor-draining lymph node. n = 4 mice per group, one experiment, one-way ANOVA test with Dunnett's multiple comparisons. **B, C, H–J**) Box plots show median, 25th–75th percentiles, minimum–maximum whiskers, with all data points displayed. **B, C, E, G, H–J** Source data are provided as a Source data file. **A, F** Created in BioRender. Guermonprez, P. (2025) https://BioRender.com/o76t012.

tumors, based on synergistic activation of chemokine receptors (CXCR3, CCR5) and the FLT3 receptor tyrosine kinase.

The prospect of locally implanted, eMSCs acting as versatile, sustained factories for therapeutic molecule delivery is highly attractive for cancer immunotherapy. This is particularly relevant for chemokines, which generate spatially organized gradients governing leukocyte trafficking and tissue compartmentalization. Nonetheless, several technological and translational challenges must be addressed to realize the clinical potential of eMSCs.

First, standardized and GMP-compatible protocols for autologous eMSCs isolation, engineering, and production must be established. Such methods should ensure reproducible recovery of stromal populations of controlled complexity that are amenable to ex vivo expansion, viral transduction, and post-transduction enrichment. Although the present study primarily employed murine embryonic fibroblasts as an MSC source, we demonstrate that adipose-derived stromal cells can be expanded, transduced, and used to generate immunologically competent eMSCs. Other groups have successfully implemented bone marrow-derived eMSCs for immunotherapy[53,57–59]. Validation in human systems will be critical for clinical translation. Furthermore, the phenotypic heterogeneity resulting from ex vivo expansion and transduction remains to be fully characterized. Despite homogeneous transgene expression (e.g., via IRES-linked fluorescent reporters), we cannot exclude that this cellular preparation is functionally heterogeneous and that only a subset of eMSCs may be functionally active in vivo. Integration of single-cell RNA sequencing and spatial transcriptomics in eMSC-treated tumors could provide essential insight into stromal diversity, fate, and functional states.

Second, optimization of eMSC delivery methods could improve therapeutic persistence and efficacy. In our current model, intratumoral injection of cell suspensions resulted in transient eMSC persistence, with only a minor fraction detectable beyond ten days. Future studies should aim at enhancing eMSC residency through tissue engineering strategies, including synthetic scaffolds or extracellular matrix integration[56]. Cell-intrinsic manipulation of the stress pathway might also offer some opportunities to improve the survival of eMSC in the TME. For instance, NQO1 expression might improve resistance to oxidative stress[54]. Would these manipulations improve persistence, it would be important to determine whether prolonged eMSC survival improves T and NK cell−dependent tumor control. While intratumoral delivery provides localized control, it may not be feasible for all cancer types. Developing systemic delivery approaches for eMSCs could extend therapeutic reach to deep or inaccessible lesions while improving patient tolerability. Indeed, intravital imaging studies have shown that intravenously administered MSCs exhibit intrinsic tropism for sites of tissue remodeling, including wounds and solid tumors[67,68]. The molecular mechanisms underlying this selective homing remain poorly defined but could be exploited to deliver eMSCs to otherwise unreachable tumor sites.

Third, eMSC-based therapies designed to stimulate DC-dependent antitumor immunity must also incorporate cues that promote antigen release and DC maturation, the two essential steps required to initiate the cancer-immunity cycle. In our study, eMSCs expressing FLT3L and CXCL9/CCL5 effectively enhanced local DC infiltration but did not efficiently drive DC maturation, potentially limiting migration to tumor-draining lymph nodes and cross-priming of CD8+ T cells. Embedding maturation-inducing signals within eMSCs may overcome this limitation. In parallel, promoting local tumor cell death could enhance antigen availability for DC uptake. One promising avenue involves combining eMSCs with oncolytic virotherapy, which induces immunogenic cell death and antigen release. eMSCs have been widely explored as delivery vehicles for oncolytic viruses, leveraging their tumor-homing capacity[69,70]. This approach has been implemented for several viral platforms, including herpes simplex virus[71], myxoma virus[72], Newcastle disease virus[73], measles virus[74], and adenovirus[74,75]. We propose that eMSC-based oncolytic virotherapy could be synergistically combined with DC-targeting eMSCs, as suggested by Svensson-Arvelund et al., who observed cooperation between recombinant FLT3L administration and NDV-mediated oncolysis[76]. Moreover, type I IFN−releasing eMSCs could complement DC-directed eMSCs to further enhance local immune activation[56,58,59].

In summary, while technical and regulatory limitations remain, the translational potential of eMSC-based immunotherapies is considerable. Their inherent tumor tropism, compatibility with ex vivo genetic engineering, and capacity for localized, sustained delivery make eMSCs a promising cellular platform for next-generation combination immunotherapies aimed at amplifying both innate and adaptive antitumor immunity.

## Methods
### Mice
Mice were housed at the INSERM U1149 CRI, Medicine school site Bichat or at the Paris Institut Pasteur animal house facilities. Wild type C57BL/6J and CD45.1 (CByJ.SJL(B6) Ptprc^a/J) mice were purchased from Janvier and kept in our facility. *Irf8*-GFP (B6.Cg *Irf8*^tm2.1Hm^/J), *Xcr1*^cre^ (B6 *Xcr1*^tm1Ciphe^), ROSA26-LSL-RFP (B6.Cg *Gt(ROSA)26Sor*^tm1Hjf^/J) and ROSA26-LSL-DTA (B6.129P2 *Gt(ROSA)26Sor*^tm1(DTA)Lky^/J)[77] mice were kindly provided by Marc Dalod ROSA36-LSL-tdTomato (B6.Cg *Gt(ROSA)26Sor*^tm14(CAG-tdTomato)Hze^/J) were kindly provided by Dr. Tessa Bergsbaken. *Xcr1*^DTA^, *Xcr1*^RFP^, *Xcr1*^tdTomato^ mice were generated as described[40]. OT-I *Rag2*^−/−^ CD45.1 mice were kindly provided by Dr. Sebastian Amigorena. BALB/c *Rag2*^−/−^ *Il2ry*^−/−^ *Sirpα*^NOD^ (BRGS), BALB/c *Rag2*^−/−^ *IL2ry*^−/−^ *Sirpα*^NOD^ *Flt3*^−/−^ (BRGSF) mice and BRGSF-HIS mice were provided by the Human Disease Model Core facility (Institut Pasteur, Paris). The study was approved by the local ethics committee (Comité d'éthique Paris Nord n°121 or CEEA89) and the French Ministry of Education and Research under the authorization number

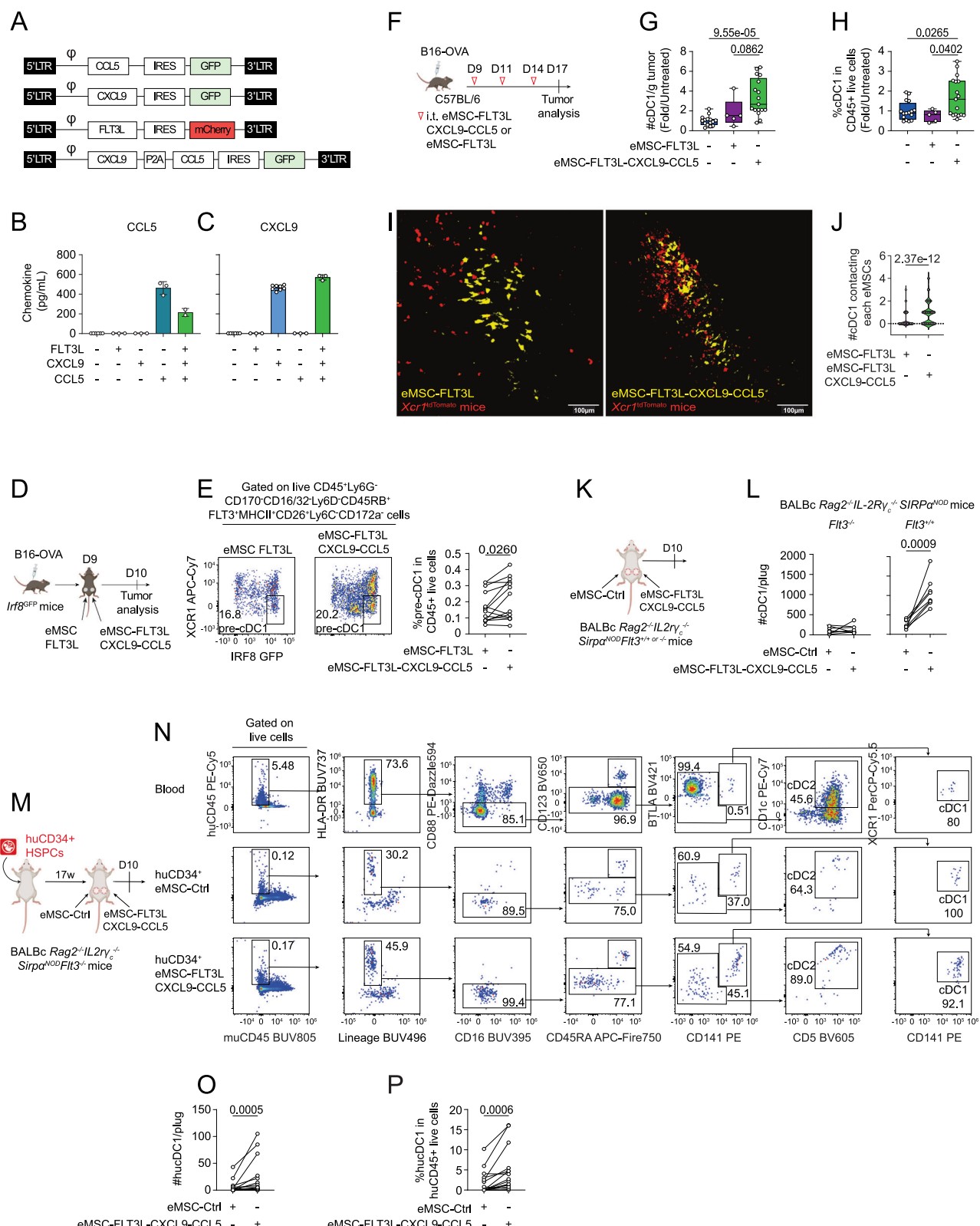

APAFIS#15373 and APAFIS #48865. Animal care and treatment were conducted with national and international laws and policies (European Economic Community Council Directive 86/609; OJL 358; December 12, 1987). All experiments were performed in specific pathogen-free animal facilities, under controlled environmental conditions with ambient temperature maintained at 22 ± 2 °C, a relative humidity between 40 and 60% and on a 12 h light/dark cycle, in accordance with the Federation of European Laboratory Animal Science Association (FELASA) guidelines, institutional guidelines and the French law.

### BRGSF humanized immune system mice (BRGSF-HIS)

Briefly, cord blood CD34+ cells (Biological Resource Center (CRB)-Banque de Sang de Cordon, AP-HP, Hôpital Saint-Louis, Unité de Thérapie Cellulaire, Paris, France) were isolated with affinity matrices according to the manufacturer's instructions (Miltenyi Biotec) and

**Fig. 7 | Intratumoral delivery of eMSC co-expressing FLT3L, CXCL9 and CCL5 stimulates cDC1 infiltration in mice and humanized mice. A** Constructions to produce eMSC-CCL5, eMSC-CXCL9 and eMSC-FLT3L-CXCL9-CCL5. ELISA of CCL5 (**B**) and CXCL9 (**C**) released by transduced eMSCs. CCL5: n = 6 (untransduced), n = 3 (FLT3L, CXCL9, CCL5), n = 2 (FLT3L-CXCL9-CCL5), CXCL9: n = 6 (untransduced), n = 3 (FLT3L, CCL5, FLT3L-CXCL9-CCL5), n = 8 (CXCL9) technical replicates, one experiment. Data are presented as mean values ± SD. **D** Experimental design to evaluate eMSC-FLT3L-CXCL9-CCL5 ability to recruit pre-cDC1s within tumors. IRF8-GFP mice bearing two tumors received eMSC-FLT3L (left) and eMSC-FLT3L-CXCL9-CCL5 (right). **E** Representative flow cytometry plots and frequencies of intratumoral pre-cDC1s upon CD45⁺ live cells. n = 14 mice, three independent experiments, two-tailed paired t-test. **F** Experimental design to evaluate cDC1 recruitment within tumors. Flow cytometry quantification of the absolute number of cDC1/g tumor (**G**) and their frequency in CD45⁺ live cells (**H**). Results are shown as fold change to control (untreated). **G:** n = 15 (Untreated), **H:** n = 16 (Untreated), **G**, **H:** n = 17 (eMSC-FLT3L-CXCL9-CCL5), n = 5 (eMSC-FLT3L) mice, three independent experiments, one-way ANOVA with Tukey's multiple comparisons test. Box plots show median, 25th–75th percentiles, minimum–maximum whiskers, with all data points displayed. **I** Representative images of YUMM-OVA tumors injected in $Xcr1^{tdTomato}$ mice and treated with eMSC-FLT3L or eMSC-FLT3L-CXCL9-CCL5. Images represent masks for cDC1s (red) and eMSCs (yellow). n = 2 mice per group, one experiment. **J** Violin plots showing the number of cDC1s contacting an eMSC. n = 159 (eMSC-FLT3L), n = 109 (eMSC-FLT3L-CXCL9-CCL5) individual eMSC, one experiment, two-tailed Mann–Whitney test. **K** Experimental design to evaluate cDC1 recruitment in tissues in BALB/c $Rag2^{-/-}$ $IL2r\gamma_c^{-/-}$ $Sirpa^{NOD}$ $Flt3^{+/+ or -/-}$ (BRGS and BRGSF) mice. **L** Flow cytometry quantification of the absolute number of cDC1/plug in BRGS $Flt3^{-/-}$ or $Flt3^{+/+}$ mice. n = 8 plugs, one experiment, two-tailed paired t-test. **M** Experimental design to evaluate human cDC1 recruitment in tissues of BRGSF humanized mice. **N** Gating strategy used to identify human cDC1s in plugs. Lineage: NKp46, CD3, CD19, CD56, CD66b, CD203c, CD20. **O**, **P** Quantification of the absolute number (**N**) and frequencies (**O**) of human cDC1/plug. n = 16 plugs, three independent experiments, two-tailed Wilcoxon matched-pairs signed rank test. **B**, **C**, **E**, **G–J**, **L**, **O**, **P** Source data are provided as a Source data file. **D**, **F**, **K**, **M** Created in BioRender. Guermonprez, P. (2025) https://BioRender.com/o76t012.

subsequently phenotyped for CD34 and CD38 expression. Newborn (5–9 days old) BRGSF mouse pups received sublethal irradiation (3 Gy) and were injected intra-hepatically with $3 \times 10^5$ CD34⁺ human cord blood cells. Twelve and 16 weeks after graft, reconstitution of the human immune system was confirmed by analysis of human CD45, CD19, CD3, CD4, and CD8 expression on live leukocytes in the blood using flow cytometry. All animal studies were approved by the Pasteur Institute Safety Committee in accordance with French and European Guidelines (Committee for Ethics in Animal Experimentation 240006) and validated by the French Ministry of Education and Research (APAFIS #49211-2024042612197924 v3). The use of human CD34⁺ cells was ethically and scientifically approved, as the CRB providing these samples is AFNOR-certified under the ISO 20387 standard. Furthermore, the scientific project was approved by the CRB management and a scientific board. Lastly, sample transfer for research purposes was authorized by the French Ministry of Higher Education and Research and the Île-de-France Regional Health Agency (authorization AC-2024-6640). Written informed consent was obtained from all patients, and a formal agreement between AP-HP and the Pasteur Institute governed the transfer and use of these samples.

## Tumor cell lines
B16-OVA and YUMM-OVA cell lines were generated by lentiviral transduction using a pCDH_CMV7-OVA-EFI-G418 plasmid. TC-1-Luc cells were kindly provided by Dr. Alexandre Boissonnas, E0771 cells were kindly provided by Stéphanie Hugues, and MC38 cells were kindly provided by Dr. Philippe Bousso. B16F10, B16-FLT3L[78], B16-OVA and TC-1-Luc cell lines were cultured in RPMI Medium 1640 (ThermoFisher Scientific) supplemented with GlutaMAX, 10% heat-inactivated fetal bovine serum (FBS) (ThermoFisher Scientific), 50 μM β-mercaptoethanol (ThermoFisher Scientific) and 100 U/mL penicillin-streptomycin (ThermoFisher Scientific). YUMM-OVA, MC38, and E0771 cell lines were cultured in DMEM (ThermoFisher Scientific) supplemented with GlutaMAX, 10% heat-inactivated FBS (ThermoFisher Scientific), 50 μM β-mercaptoethanol (ThermoFisher Scientific) and 100 U/mL penicillin-streptomycin (ThermoFisher Scientific). Cells were maintained at 37 °C and 5% $CO_2$.

## Tumor models
For the grafted tumor model, tumor cells were harvested during the exponential growth phase using 0.05% Trypsin-EDTA (Gibco), washed twice in PBS, and counted using trypan blue exclusion. 8- to 12-week-old male (B16-OVA, MC38, YUMM-OVA) and female (E0771, TC-1) mice were subcutaneously inoculated with $5 \times 10^5 – 1 \times 10^6$ tumor cells suspended in 100 μL of not supplemented RPMI or DMEM medium. Tumor dimensions were recorded three times per week using a digital caliper, and tumor volume (V) was calculated using the formula $V = (L \times W \times W)/2$, with volume expressed in mm³. Mice were monitored daily, and endpoints were established based on signs of distress, including pain, lethargy, dehydration, ulceration, or tumor necrosis. To generate survival curves, mice were sacrificed at ethical endpoints, the main one being the tumor volume (2000 mm³). In some cases, this limit has been exceeded the last day of measurement, and the mice were immediately euthanized.

## Generation of engineered mesenchymal stromal cells
**Generation of mesenchymal stromal cells.** Mesenchymal stromal cells (MSC) were generated from mouse embryonic fibroblasts (MEFs) from individual C57/BL6 embryos at embryonic day 13 and were kindly provided by Loredana Saveanu. Practically, the embryo body was digested with 0.05% trypsin and 0.02% EDTA in Hanks balanced salt solution at 37 °C for 30 min. The single cell suspension obtained after digestion was resuspended in high glucose DMEM medium containing 10% FCS, 2mM L-glutamine, 100 U/ml penicillin and 100 μg/ml streptomycin and seeded on 0.1% gelatin (Sigma G1890) coated tissue culture dishes at $5 \times 10^4$ cells/cm². Cells growing on the gelatin-coated dishes were harvested, and cell passages were repeated in DMEM medium containing 10% FCS, 2mM L-glutamine, 100 U/ml penicillin and 100 μg/ml streptomycin.

**Generation of adipose-derived mesenchymal stromal cells.** Method was adapted from Dr. Geneviève Marcelin. Inguinal and visceral fat of C57BL/6 mice were cut in small pieces and digested with 6–8 mL of digestion mix, containing high glucose DMEM (ThermoFisher Scientific), 1 mg/mL collagenase A (Roche), resuspended in HBSS (ThermoFisher Scientific), 100 U/mL HEPES (ThermoFisher Scientific), 100 U/mL penicillin/streptomycin (ThermoFisher Scientific) and 1 mg/mL BSA. After 40 min digestion at 37 °C, the digestion was mechanically ended with a syringe and a needle of 18 G. Cell suspension was then filtered on a 100 μm cell strainer and centrifuged. Floating adipocytes and red blood cells (ACK lysis, ThermoFisher Scientific) were removed. Cells were enriched in CD45-Ter119- cells using a StemCell Magnet and biotin anti-CD45 and anti-Ter119 antibodies. Enriched cells were then cultured in an adherent flask with DMEM (Gibco®) supplemented with GlutaMAX, 20% heat-inactivated FBS (Gibco®), 50 μM β-mercaptoethanol (ThermoFisher Scientific) and 100 U/mL penicillin-streptomycin (ThermoFisher Scientific). Cells were maintained at 37 °C and 5% $CO_2$.

**Engineering of mesenchymal stromal cells.** Human FLT3L, CXCL9 and CCL5 were amplified by PCR from previously obtained or commercial cDNA (Horizon Discovery) and cloned into pMX retroviral

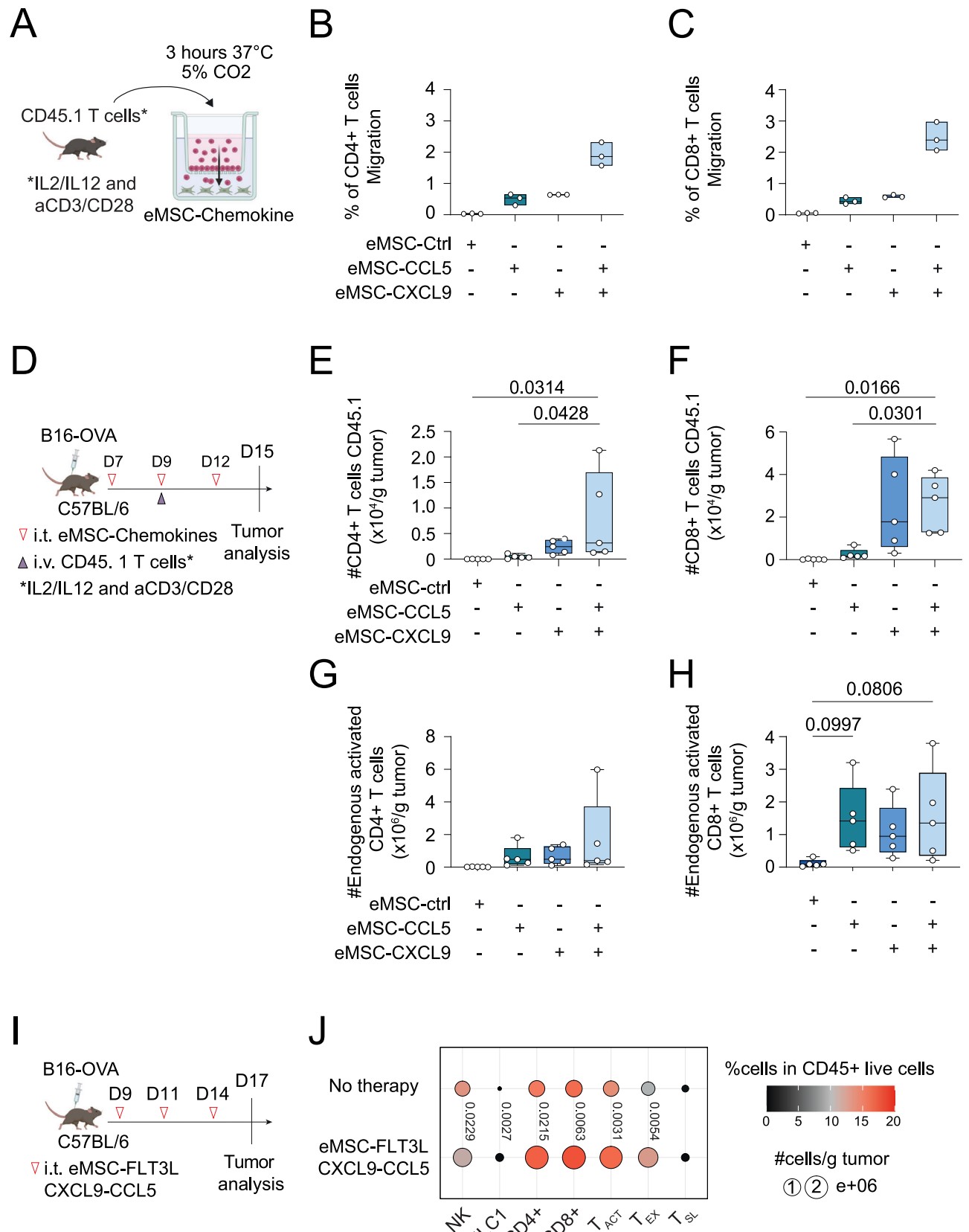

vectors (detailed in the Supplementary Table 1). To generate viral particles, the Platinum-E (Plat-E) Retroviral Packaging Cell Line (kindly provided by Dr. Laurie Menger) was transfected with 1, 2 μg plasmid, 21 mM CaCl$_2$ and HEPES-Buffered Saline (HBS, homemade solution). MSCs were then transduced with the viral supernatant and 8 μg/mL

polybrene (Santa-Cruz Biotechnology). Transduced MSCs were selected using 2 μg/mL puromycin (ThermoFisher Scientific) and sorted on their expression of membrane bound FLT3L (antibody details provided in the Supplementary Table 1) and/or their expression of eGFP or mCherry reporters. Engineered MSCs were cultured in DMEM (Gibco®)

**Fig. 8 | Intratumoral delivery of eMSC co-expressing FLT3L, CXCL9 and CCL5 stimulates the infiltration of activated CD4+ and CD8+ T cells.**
**A** Experimental scheme of a migration assay. eMSC-CXCL9, eMSC-CCL5 and eMSC-ctrl were plated in the lower compartment of Boyden chambers, 24 h before the migration assay. Ex vivo polarized CD45.1 T cells were then transferred to the upper compartment of the culture system and incubated at 37 °C for 3 h. Quantification of the migration index of CD4+ (**B**) and CD8+ (**C**) T cells. The migration index was obtained by counting the number of T cells that migrated to the bottom chambers, regarding the number of T cells initially put on top of the chambers. n = 3 technical replicates, one experiment. Data are represented as floating bars with individual data points and a line at median. **D** Experimental design for adoptive Th1-CD45.1 T cells transfer in B16-OVA-bearing mice. Quantification of the absolute number of CD45.1 CD4+ T cells/g tumor (**E**) and CD45.1 CD8+ T cells/g tumor (**F**). n = 5 mice per group, one experiment, one-way ANOVA test with Dunnett's multiple comparisons. Quantification of the absolute number of endogenous CD44+CD4+ T cells/g tumor

(**G**) and endogenous CD44+CD8+ T cells/g tumor (**H**). n = 5 mice per group, one experiment, one-way ANOVA test with Dunnett's multiple comparisons.
**I** Experimental design to assess cDC1 intratumoral infiltration after administrating eMSC-FLT3L-CCL5-CXCL9. **J** Quantification of the absolute number and the frequencies in CD45+ live cells of NK, ILC1, CD4+, CD8+CD44+ activated (T_{ACT}), CD8+CD44+PD1+Tim3+ exhausted (T_{EX}), CD8+CD44+PD1+Tim3− stem-like (T_{SL}) T cells. Dots represent the absolute number of cells/g tumor while the colors represent the frequencies of cells in CD45+ live cells. n = 11 (No therapy), n = 12 (eMSC-FLT3L-CXCL9-CCL5) mice per group, two independent experiments, statistics are shown for the quantification of the absolute numbers, two-tailed unpaired t-test. **E**–**H** Box plots show median, 25th–75th percentiles, minimum–maximum whiskers, with all data points displayed. **B**, **C**, **E**–**H**, **J** Source data are provided as a Source data file. **A**, **D**, **I** Created in BioRender. Guermonprez, P. (2025) https://BioRender.com/o76t012.

supplemented with GlutaMAX, 10% heat-inactivated FBS (Gibco®), 50 μM β-mercaptoethanol (ThermoFisher Scientific), 2 μg/mL puromycin (ThermoFisher Scientific) and 100 U/mL penicillin-streptomycin (ThermoFisher Scientific), under standard culture conditions of 37 °C and 5% $CO_2$. MSCs expressing the nano-luciferase were obtained by transducing MSCs with a lentiviral vector kindly provided by Laleh Majlessi.

### In vitro experiments
**T cells isolation, activation and polarization.** CD4+ and CD8+ T cells were isolated from the draining lymph nodes and spleens of 10–12-week-old CD45.1 mice. Purification was performed using negative selection with a biotin-conjugated antibody cocktail, followed by anti-biotin magnetic beads and separation with a Miltenyi magnet and LS columns (Miltenyi Biotec). For in vitro stimulation, purified T cells were cultured on plates coated with anti-CD3 (2 μg/mL; clone 17A2) and anti-CD28 (2 μg/mL; clone 3751; BioLegend) in the presence of recombinant IL-2 (5 μg/mL; BioLegend). To promote Th1 differentiation, cultures were supplemented with recombinant IL-12 (10 ng/mL; BioLegend) and blocking antibody anti-IL-4 (5 μg/mL; clone MP4-25D2; BioLegend).

**Migration assay.** Cell migration assays were conducted using 24-well transwell inserts with 5-μm pore polycarbonate membranes (Corning). Engineered mesenchymal stromal cells (eMSCs) expressing CCL5 or CXCL9 were seeded at a density of 250,000 cells per well in the lower chamber. Once the eMSCs reached confluency, the culture medium was replaced with serum-free DMEM. A total of 100,000 CD45.1 Th1-polarized T cells were resuspended in serum-free DMEM, and 100 μL of the cell suspension was carefully added to the upper chamber of the Transwell insert. The upper chamber was then gently placed into the lower chamber using sterile forceps, ensuring proper immersion in the culture medium. The plate was incubated at 37 °C for 3 h. Following incubation, the supernatant was collected to quantify T-cell migration by flow cytometry.

### In vivo experiments
**HSPC differentiation in membrane extract plugs.** Immunomagnetic negative selection was used to isolate mouse stem and progenitor cells (HSPC) from CD45.1 mice using the Mouse Hematopoietic Progenitor Cell Isolation Kit (StemCell), following the manufacturer's instructions. A total of $10^4$ hematopoietic stem and progenitor cells (HSPCs) per plug were subcutaneously injected alongside engineered mesenchymal stromal cells (eMSCs) at a 1:50 ratio (HSPC/eMSC) in 200 μL of ice-cold Geltrex LDEV-Free Reduced Growth Factor Basement Membrane Matrix (Thermo Fisher). Mice were euthanized on day 12 of differentiation via cervical dislocation, and GelTrex plugs were excised for further analysis. The plugs were sectioned and enzymatically digested in HBSS (Life Technologies) containing 1% FBS, 0.37 U/mL Collagenase D (Roche), 10 μg/mL DNase I (Roche), and 1 mg/mL Dispase (Sigma-

Aldrich) at 37 °C for 30 min. Following digestion, the tissue was smashed on a 100-μm strainer (Corning), and the resulting cell suspension was collected and resuspended in FACS buffer for flow cytometry analysis. For experiments involving recombinant FLT3L, human FLT3L was used at a concentration of 100 ng/mL (Celldex).

**In vivo treatment with engineered stromal cells.** Intratumoral injections of eMSCs started when the tumors were between 50 and 100 mm³. Mice were treated with $5 \times 10^5$–$1 \times 10^6$ eMSC-FLT3L, in combination with 25 μg/mouse of poly I:C LMW (InVivoGen). We designed our intratumoral cell-based therapy as two injections of eMSC-FLT3L (day 7 and day 12) and one injection of poly(I:C) (day 9). Mice that were later treated with $5 \times 10^5$–$1 \times 10^6$ eMSC-FLT3L-CXCL9-CCL5 received three injections of eMSCs. In the latest experiments, mice developed tumors with the desired injectable size few days later than the initial schedule. Thus, the start of injections was postponed but the number and frequencies of injections was kept.

**In vivo bioluminescence.** A total of $1 \times 10^6$ eMSC-NanoLuc cells were administered intratumorally into palpable B16-OVA tumors. Mice were imaged using the IVIS SpectrumCT 2 In Vivo Imaging System 4 h after injection and again at days 4 and 8. Prior to imaging, mice were anesthetized and received an intravenous injection of 0.44 μmoles fluorofurimazine substrate (Promega). On day 8, mice were sacrificed, and various organs were collected and imaged ex vivo to assess bioluminescence. Photon flux was quantified using the Living Image 4.8.2 software. Regions of interest (ROIs) were quantified as average radiance (photons/s/cm²/sr) represented as color-scaled images superimposed on grayscale photos of mice using Living Image software.

**In vivo treatment with recombinant FLT3L.** Intratumoral injections of recombinant FLT3L started when the tumors were between 50 and 100 mm³. Mice were treated 4 times (every 2 days) with 10 μg/mouse of recombinant FLT3L (Celldex).

**Serum alanine aminotransferase (ALT) and aspartate transaminase (AST) assay and hematoxylin-eosin coloration.** Serum was collected 48 h after poly(I:C) intratumoral or anti-CD40 (100 μg/mice, clone FGK 4.5) intraperitoneal injection. Blood was allowed to clot at room temperature for 30 min and centrifuged two times at 1000 × g for 10 min. ALT and AST levels were measured using the ALT/GPT kit (ThermoFisher Diagnostics) following the manufacturer's instructions. Samples were run on a Konelab analyzer (ThermoFisher Scientific). Liver of the same mice were harvested and put in 4% paraformaldehyde (ThermoFisher Scientific) overnight. The next day, livers were transferred in 70% ethanol, paraffin embedded, cut, and stained with hematoxylin and eosin (H&E) at the histology platform of the Pasteur Institute. Slides were scanned using the Axioscan (Zeiss).

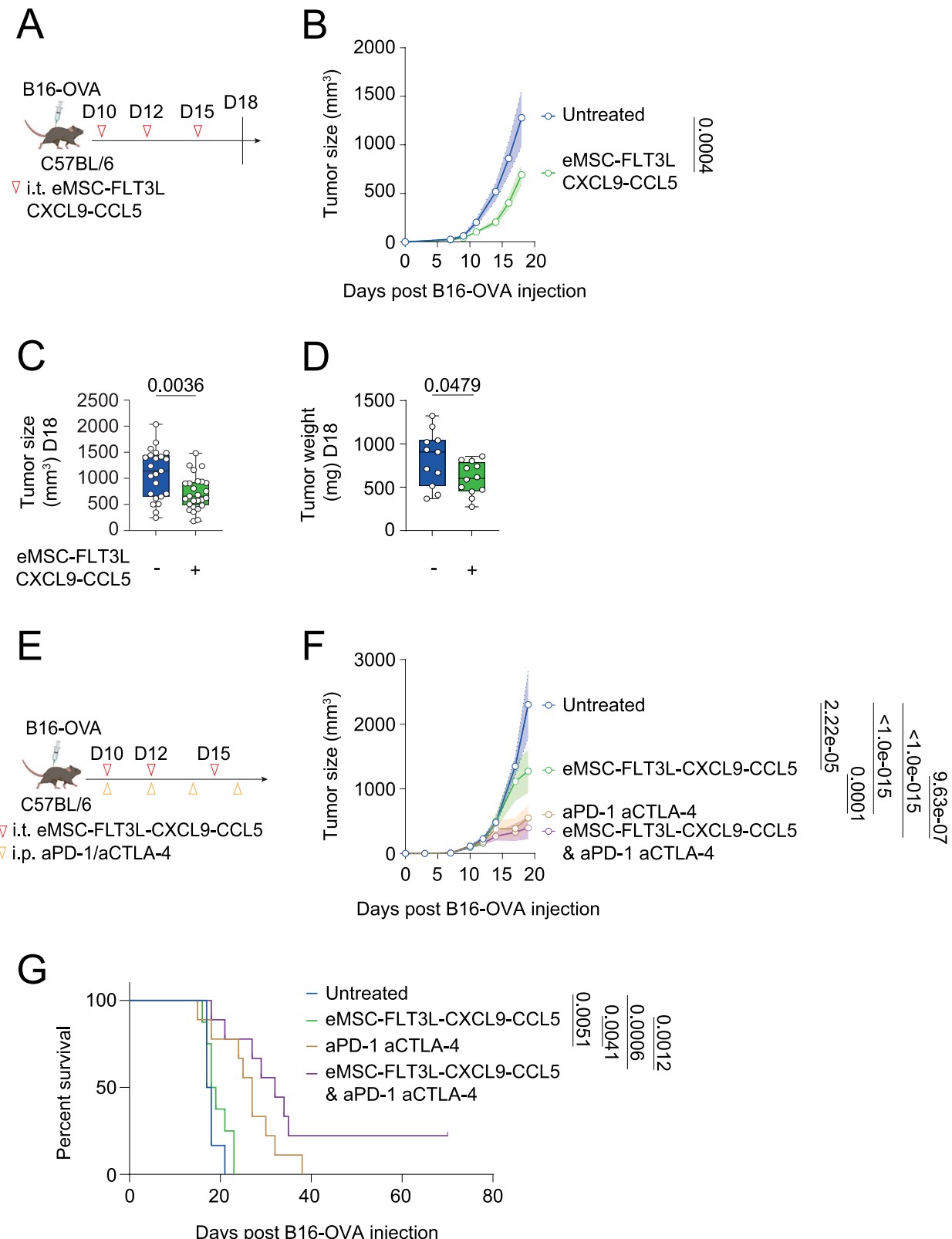

**Immune checkpoint blockade (ICB) treatment.** Mice received intraperitoneal (i.p.) injections of 200 μg anti-PD-1 (clone 29F.1A12, BioXcell) and 200 μg of anti-CTLA-4 (clone 9H10, BioXCell). An isotype antibody (clone 1-113, BioXcell) and IgG control (clone MPC11, BioXCell) was used as a control. All antibodies were administered intraperitoneally at treatment initiation and subsequently every 2 days until reaching 5 injections.

**CD4, CD8, CD25 and NK cells depletion.** For the depletion of CD4⁺ T cells, CD8⁺ T cells, Tregs and NK1.1⁺ cells, mice were treated with a first dose of 500 μg of anti-CD4 (clone GK1.5, BioXCell), anti-CD8 (clone 2.43, BioXCell), anti-CD25 (clone PC-61.5.3, BioXCell), and anti-NK1.1 (clone PK136, BioXCell). As controls, 300 μg of IgG2a isotype (clone C1.18.4, BioXCell) and 500 μg of rat IgG2b isotype (clone LFT-2, BioXCell) were used. All antibodies were administered

**Fig. 9 | Intratumoral delivery of eMSC co-expressing FLT3L, CXCL9 and CCL5 is sufficient to activate anti-tumor immunity. A** Experimental design to assess the anti-tumor effect of eMSC-FLT3L-CCL5-CXCL9. **B** Tumor growth curves. n = 5, representative of four independent experiments, two-way ANOVA-test with Šídák's multiple comparisons. **C** Tumor size at day 18. n = 23 (untreated), n = 25 (eMSC-FLT3L-CXCL9-CCL5) mice per group, four independent experiments, two-tailed unpaired t-test. **D** Tumor weight at day 18. n = 11 (untreated), n = 12 (eMSC-FLT3L-CXCL9-CCL5) mice per group, two independent experiments, two-tailed unpaired t-test. **E** Experimental design to evaluate the synergy of eMSC-FLT3L-CXCL9-CCL5 with immune checkpoint blockade. **F** Tumor growth curves. n = 6 (untreated), n = 9

(aPD-1-aCTLA-4, eMSC-FLT3L-CXCL9-CCL5 + aPD-1-aCTLA-4), n = 8 (eMSC-FLT3L-CXCL9-CCL5), two-way ANOVA-test with Tukey's multiple comparisons. **G** Survival curves. n = 6 (untreated), n = 9 (aPD-1-aCTLA-4, eMSC-FLT3L-CXCL9-CCL5 + aPD-1-aCTLA-4), n = 8 (eMSC-FLT3L-CXCL9-CCL5), one experiment, log-rank (Mantel–Cox) test. **B–F** A line represents the mean, and SEM is shown with the colored area. **C**, **D** Box plots show median, 25th–75th percentiles, minimum–maximum whiskers, with all data points displayed. **B–D**, **F**, **G** Source data are provided as a Source data file. **A**, **E** Created in BioRender. Guermonprez, P. (2025) https://BioRender.com/o76t012.

intraperitoneally at the initiation of treatment and subsequently every 3 days at a dose of 200 μg, until reaching 4–6 injections.

**CXCR3- and CCR5-antagonists treatments.** Mice received intraperitoneal injections of either maraviroc (500 μg/mouse, MedChemExpress), a CCR5 antagonist, or AMG 487 (200 μg/mouse, MedChemExpress), a CXCR3 antagonist. Both agents were dissolved in a vehicle of dimethyl sulfoxide (DMSO). Administration started at the initiation of treatment and continued every two days until day 15, where samples were collected for flow cytometry analysis.

**In vivo re-challenge OVA peptide in mice.** Survivor mice that fully rejected B16-OVA tumors treated with eMSC-FLT3L + poly(I:C) underwent in vivo restimulation via subcutaneous injection of 1 μg OVA 257–264 peptide (SIINFEKL, IBA Lifesciences) dissolved in 100 μl phosphate-buffered saline (PBS). This experiment was conducted 90 days after primary tumor injection. The injection site was located in the dorsal flank region. Three days post-restimulation, mice were euthanized, and inguinal draining lymph nodes were excised for subsequent analysis.

**Adoptive T cell transfer.** $1 \times 10^5$ eMSCs were injected in B16-OVA tumors at day 7, 9 and 12. At day 9, $2 \times 10^6$ Th1-polarized CD45.1 T cells (previously described) were intravenously administered.

**Synthetic niches (plugs).** $1 \times 10^6$ eMSC-ctrl or eMSC-FLT3L-CXCL9-CCL5 were injected subcutaneously (s.c.) in 200 μL solution, containing 150 μL of Geltrex LDEV-free reduced growth factor basement membrane matrix and 50 μL of PBS. The synthetic niches (plugs) were harvested and analyzed by flow cytometry 10 days after the injection.

## Mouse tissues processing

Murine tumors, tumor-draining lymph nodes (LNs), spleens, or synthetic niches were harvested and subjected to enzymatic digestion. Tissues were incubated in 3 mL of digestion buffer, comprising Hank's Balanced Salt Solution (HBSS, ThermoFisher Scientific) supplemented with calcium and magnesium (ThermoFisher Scientific), 75 μg/mL Collagenase D (Roche), and 0.02 mg/mL DNase I (ThermoFisher Scientific), at 37 °C for 30 min. Single-cell suspensions were then generated by smashing the digested tissues through 70 μm cell strainers (Corning). Spleen red blood cells were lysed using ACK lysis buffer (ThermoFisher Scientific). The absolute number of live immune cells in each tissue cell suspension was assessed using AccuCheck Counting Beads (ThermoFisher Scientific) along with anti-CD45 and DAPI stainings on BD LSR-Fortessa X20, BD Symphony A5 (BD Biosciences) or spectral SONY ID7000 (SONY).

## Flow cytometry analysis and fluorescent-activated cell sorting

After tissue processing and cell counting, Fc-receptor were blocked using FcBlock for 20 min at 4 °C. Viability was assessed using 7-AAD (Biolegend), Zombie dyes (BioLegend), or Live/dead fixable stains (ThermoFisher Scientific). Extracellular staining was performed in FACS buffer (PBS (Life Technologies), 3% FBS (ThermoFisher Scientific), and 2 mM EDTA (Life Technologies)) during 30 min at 4 °C.

Labelling with H2K$^b$-SIINFEKL tetramers (MBL) was performed at room temperature for 30 min. For intra-cellular stainings, single cell suspensions were then fixed during 30 min and stained during 30 min using a fixation/permeabilization kit (eBiosciences, transcription factor staining buffer). For intra-cellular IFNγ staining, single-cell suspensions were first restimulated ex vivo with 2 μM OVA257–264 peptide (SIINFEKL, IBA Lifesciences) for 5 h at 37 °C in 5% CO$_2$, followed by a 4-h incubation with 5 μg/mL Brefeldin A (Sigma Aldrich). Cells were subsequently labeled with surface antibodies, fixed either with the Cytofix/Cytoperm kit (BD, cytokines stainings) or the Transcription Factor Staining Buffer Set (Thermo Fisher Scientific, nuclear intracellular stainings), and stained with an intracellular cytokine staining (ICS) antibody panel. A comprehensive list of antibodies used is provided in Supplementary Table 1. Flow cytometry data were acquired using BD LSR-Fortessa X20, BD Symphony A5 (BD Biosciences) or ID7000 Spectral (Sony). Cells sorting was performed using a BD FACSMelody™ Cell Sorter equipped with BD FACSDiva software. Data were analyzed with FlowJo software.

## Multiplex chemokines analysis

Dissected tumor samples were weighted and homogenized in 1 mL of 1× phosphate-buffered saline (PBS) supplemented with protease inhibitor, using a Fisherbrand™ 150 handheld homogenizer. Homogenates were then centrifuged at $14,000 \times g$ for 15 min, and the resulting supernatants were collected. The concentrations of pro-inflammatory chemokines within the supernatants were quantified using the LEGENDplex™ MU Proinflammatory Chemokine Panel 1 and 2 (BioLegend), following the manufacturer's instructions. Samples were analyzed by flow cytometry on a BD FACS LSRFortessa. Data were analyzed using LEGENDplex™ software (BioLegend). Chemokine concentrations were normalized to total protein concentration, which was estimated using the initial tumor mass.

## Immunofluorescence

$1 \times 10^6$ YUMM-OVA tumor cells were injected in the flank of Xcr1$^{RFP}$ or Xcr1$^{tdTomato}$ mice. Intratumoral injection of eMSC-ctrl, eMSC-FLT3L or eMSC-FLT3L-CXCL9-CCL5 started when tumors were between 50 and 100 mm$^3$. Mice were sacrificed two days later. Protocol was adapted from Laforêts et al.[79]. Tumors were harvested and embedded in 5% agarose. 300–350 μm thick slices of non-fixed fresh samples were obtained using a vibratome (LEICA, VT1200S), with blade speed between 0.5 and 0.75 mm/s and the amplitude at 1 mm. Samples were then collected in phenol-red free DMEM medium, containing 10% FBS, 1% penicillin/streptomycin and 4mM L-glutamine. Live tumor samples were then put in a 35 mm glass bottom dish, filled with the before mentioned medium. Images were taken with a Nikon Ti2E, equipped with a 20× objective, a spinning disk and a temperature-controlled chamber at 37 °C. Images were analyzed with Fiji software. To quantify the number of cDC1s surrounding each eMSC, a threshold was applied to each channel of every image, based on the maximum intensity projection, generating one channel containing cDC1s and another containing eMSCs. For eMSC-FLT3L-CXCL9-CCL5, which co-express Cherry and GFP, cDC1s were isolated by subtracting the green channel from the red, while eMSCs were obtained by merging the two channels.

The number of cDC1s around each eMSC was then manually determined.

## Statistical analysis

Statistical analysis was performed using Prism 10 (GraphPad Software Inc., USA). Statistical analysis and verification of normal distribution were performed using GraphPad Prism software. Outliers were detected using the ROUT method in GraphPad Prism (Q = 0, 5%) and subsequently removed from the dataset prior to analysis. For comparisons of two groups, two-tailed paired or unpaired Student's *t* test was used. When three or more groups were compared, a one-way or two-way analysis of variance (ANOVA) test was used. Precise tests are described in the figure legends. A p-value of less than 0.05 was considered as significant. The number of biological as well as experimental replicates is indicated in figure legend. Data are presented as means ± SD or SEM, box and whiskers, or violin plots as described in the figure legends. SD standard deviation; SEM standard error of the mean.

## Reporting summary

Further information on research design is available in the Nature Portfolio Reporting Summary linked to this article.

## Data availability

Data supporting the findings of this study are available within the article, Supplementary Information or Source data file. Source data are provided as a Source data file. Source data are provided with this paper.

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

## Acknowledgements

P.G. is a CNRS investigator (Centre National de la Recherche Scientifique). This work was supported by Fonds National Suisse (SINGERGIA SNF CRSII5_202246/1, P.G., S.H.), INSERM transfert CO-POC grant (P.G.), Agence Nationale pour la Recherche (ANR-17-CE11-0001-01 and ANR-18-IDEX-0001, P.G.), La Ligue contre le cancer (P.G.), Institut National du Cancer INCA (PL-BIO22-147, P.G.), Fondation pour la Recherche Médicale (EQU202203014687, P.G.), Fondation pour la Recherche sur le Cancer ARC (PJA2021060003913, P.G.) and the Pasteur Institute. We would like to thank the staff of the flow cytometry and the animal facility platforms from the CRI (Centre de Recherche sur l'Inflammation, Faculté de Médecine, Bichat) and from the Pasteur Institute for all the work, help and support in most of the experiments presented in this paper. We also like to thank the staff from the histopathology core facility from the Pasteur Institute for the liver H&E images. We thank Rémy Yim (Human Disease Models Core Facility, Institut Pasteur, Université Paris Cité, 75015 Paris, France), the Human Disease Models Core Facility (Institut Pasteur, Paris) and the CRB-Banque de Sang de Cordon, AP-HP, Hôpital Saint-Louis, Unité de Thérapie Cellulaire, Paris, France for the experiments on the BRGSF reconstituted mice. We thank the units U1149 and U1016 of Institut National de la Santé et de la Recherche Médicale. We thank the unit UMR3738 of the Centre National de la Recherche at the Pasteur Institute. We also would like to thank Laleh Majlessi for providing the NanoLuciferase vector, Alexandre Boissonnas (Sorbonne Université, INSERM U1135, CNRS, Centre d'Immunologie et des Maladies Infectieuses, Cimi-Paris, Paris 75013, France), Dr. Philippe Bousso (Dynamics of Immune Responses Unit, Institut Pasteur, Université de Paris Cité, INSERM U1223, F-75015 Paris, France), Loredana Saveanu and Dr. Laurie Menger (Inserm U1015, Gustave Roussy Cancer Campus, Villejuif 94800, France) for providing us various cell lines, Marc Dalod, Dr. Tessa Bergsbaken (Department of Pathology, Immunology and Laboratory Medicine, Center for Immunity and Inflammation, Rutgers New Jersey Medical School, Rutgers-The State University of New Jersey, Newark, NJ) and Dr. Sebastian Amigorena (Institut Curie, PSL University, INSERM U932, Immunity and Cancer, Paris, France) for providing us mice strains and Dr. Geneviève Marcelin (Sorbonne Université, INSERM U1269, Nutrition and obesities: systemic approach research group, Nutriomics, Paris F-75013, France) for providing us protocols.

## Author contributions

Conceptualization: S.H., P.G. Investigation: L.G., F.L.R.D.C., P.B., J.B., N.V., A.S., M.V., A.S.K.C., N.J., A.O., M.B., O.F., M.A., D.F., I.H., Z.A.N., J.W., H.T., G.A., F.F. Resources: F.G., L.M., E.L.G., L.S., J.H., MDalod, MDusseaux, J.P.D.S., S.H., P.G. Writing: L.G., F.L.R.D.C., S.H., P.G., Supervision: S.H., P.G. Funding acquisition: M.C., J.C., J.H., S.H., P.G.

## Competing interests

The authors declare no competing interests.

## Additional information

[1]Dendritic Cells and Adaptive Immunity Unit, Immunology Department, Institut Pasteur, Université de Paris Cité, Paris, France. [2]CNRS UMR3738, Developmental Biology and Stem Cells, Institut Pasteur, Université de Paris Cité, Paris, France. [3]Centre for Vaccine and Immunotherapy, Institut Pasteur, Paris, France. [4]Laboratory of Tumor Inflammation and Angiogenesis, Center for Cancer Biology, VIB, Leuven, Belgium. [5]INSERM UMR 1016, CNRS UMR 8104, Institut Cochin, Université Paris Cité, Paris, France. [6]Bichat Medical School, INSERM UMR1149, CNRS EMR8252, Université Paris Cité, Paris, France. [7]Department of Pathology and Immunology, Geneva Medical School, Geneva, Switzerland. [8]Human Disease Models Core Facility, Institut Pasteur, Université Paris Cité, Paris, France. [9]Nutrition and Obesity: Systemic Approaches, Inserm, Sorbonne University, Paris, France. [10]Inserm U1015, Institut Gustave Roussy, Université Paris-Saclay, Villejuif, France. [11]Inserm Transfert, Paris, France. [12]Deparment of Cellular Immunology, International Centre for Genetic Engineering and Biotechnology, Trieste, Italy. [13]Virology Department, Pasteur-TheraVectys Joint Laboratory, Institut Pasteur, Université Paris Cité, Paris, France. [14]Centre d'Immunologie de Marseille-Luminy, CIML, CNRS, INSERM, Aix Marseille Université, Marseille, France. [15]Immunology Department, Immunité Innée, Institut Pasteur, Paris, France. [16]These authors contributed equally: Louise Gorline, Fillipe Luiz Rosa do Carmo. ✉e-mail: pierre.guermonprez@pasteur.fr

