## [Transparent Peer Review File · Nature Communications]

Intratumoral delivery of FLT3L with CXCR3/CCR5 ligands promotes XCR1+ cDC1 infiltration and activates anti-tumor immunity

Corresponding Author: Dr Pierre Guermonprez

Version 0:

Reviewer comments:

Reviewer #1

(Remarks to the Author)

The work by Gorline et al. investigates the use of a cell therapy-based approach to remodel the TME through the delivery of FLT3L as a means to enhance the presence of XCR1+DC1+ cells in the tumor vicinity. Well designed mechanistic studies were also conducted and identified CXCR3/CCR5 ligands as major players in the induction of an anti-tumoral effect. The paper is well laid-out overall.

Although the study will have a great impact on the field, it does have some pitfalls that need to be addresses prior to publication. Here are the major points that need to be worked-out:

1 - The authors refer to the used cells as mesenchymal stromal cells (MSCs) yet, there is no evidence provided to prove that these cells are indeed MSCs, which is surprising! For instance, the authors refer to these cells as "fibroblastic cell lines", which indicates that they are not MSCs. The authors should therefore clarify this point. If these are indeed MSCs, then they need to show their phenotypic analysis (CD31-, CD44+, CD45-, CD73+, CD90+ and CD105+).

2 - What is the expression profile of PD-1 and PD-L1 on these modified cells? This is important as pro-inflammatory signals will trigger or enhance the expression of these ICs on the cell surface.

3 - The authors gene-engineered their cells to express FTL3L and other chemokines. They should however provide evidence that there is no autocrine loop taking place whereby the gene-engineered cells express the cognate receptors and respond to the produced ligands.

4 - In figure 1, the authors seem to compare human to murine FLt3L. Is that correct? If so, why is that?

5 - The authors always show tumor growth up to day 15. This is not impressive at all. Why not showing the growth until day 30 or 40?

6 - By looking at Fig. 1L and N, one can say that the tumors that seem to be controlled by the eMSC-FLT3L + Poly I:C have regrown, which goes with my previous concern.

7 - Why choosing the B16F10 model? The authors should validate their therapeutic in vivo potency in at least 2 additional tumor models.

8 - The authors used the term cross-priming to refer to CD8 T-cell activation. Usually this process occurs if a given antigen is cross-presented. Have the authors provided any evidence for that?

9 - Why are TIM3-TCF1+CD8+ T_{pex} referred to as "exhausted". No convincing data are presented in that regard.

10 - Why are surviving animals not rechallenged with tumors?

11 - Are the eMSCs presented in Fig. 6A expressing FLT3L?

12 - Fig. 6B/C are missing numbers on the Y axis. Plz fix it.

13 - Figure 6M is confusing. We see percentages between 80-100% in all groups, which is high! The authors should better explain this figure and its relation to Fig. 6N and O.

14 - Fig. 8F is disappointing especially if compared to the first set of in vivo data in Fig. 1. Once would have expected better tumor control at that point... In fact, the authors should combine the eMSC-FLT3L + Poly I:C with the anti-PD1 and anti-CTLA4 to see if this combo therapy induces stronger or more effective anti-tumoral responses.

15 - The discussion is very thin and needs more development within the context of cancer immunotherapy. In this current format, it looks more like a summary of findings. Plz discuss the drawback, limits, pitfalls of this technology and put it in a clinical context or compare it to other experimental approaches.

16 - Why was the use of MSCs in the context of cancer vaccination not discussed? There are series of papers describing the use of gene-engineered or pharmacologically-reprogrammed MSCs in the context of cancer vaccination. Such concept should be discussed as well.

Reviewer #2

(Remarks to the Author)

This manuscript by Dr. Guermontprez and colleagues presents an elegant and innovative strategy for enhancing anti-tumor immunity through intra-tumoral delivery of engineered mesenchymal stromal cells expressing FLT3L and the chemokines CXCL9 and CCL5. The experiments are well-designed, and the mechanistic insights into cDC1 recruitment and T cell priming are compelling. I particularly appreciated the use of both mouse and humanized models, which enhances the translational relevance of the findings.

Nevertheless, a few key points could be addressed to strengthen the manuscript:

1. Cellular source of chemokines: The cellular source of CXCL9 and CCL5 following poly(I:C) treatment remains undefined. Further characterization of this axis would help clarify the underlying mechanism.
2. Safety considerations: Although systemic cytokine levels appear unaffected, the long-term biodistribution, persistence, and potential immunogenicity of the eMSCs warrant additional safety evaluation.
3. Model generalizability: The study would benefit from validation in additional tumor models to demonstrate generalizability beyond B16-OVA. This model is somewhat peculiar, as most OVA-expressing tumors are highly immunogenic and are often spontaneously rejected in immunocompetent mice. This limitation could be addressed in a follow-up study in which intra-tumoral delivery of FLT3L-expressing eMSCs is tested in other transplantable or spontaneous tumor models.
4. Mechanistic synergy with checkpoint blockade: The authors should discuss more thoroughly the mechanism by which this therapy synergizes with checkpoint blockade.
5. Clinical implications: The authors should discuss better the translational potential of eMSC therapy, as the intratumor route of delivery would not be amenable to most human cancers

Addressing these points will enhance the impact of the study. However, given the novelty and important implications of the work, I recommend the manuscript undergo minor revision prior to acceptance.

Minor specific points:

In the experiments shown in Figures 3B, S2, and S3, the observed effects on cDC1 maturation and T cell activation could be attributed to poly I:C treatment alone. The authors should include a comparison between eMSC + poly I:C and poly I:C alone to demonstrate that the presence of eMSCs is essential for enhancing cDC1 maturation and IFN- γ secretion by T cells.

Reviewer #3

(Remarks to the Author)

The manuscript by Gorline et al analyses the anti-cancer efficacy of engineered mesenchymal stem cells expressing factors that enhance DC and T/NK cell activation/infiltration in a mouse tumour model harbouring an ectopic immunodominant antigen. The use of humanized mice/human cells is highly appreciated and the authors perform a comprehensive analysis of the majority of relevant aspects of DC-mediated induction of anti-cancer T cell immunity. The authors basically report the effective repurposing of existing knowledge, the combination of a previously reported system (eMSCs) with known anti-cancer factors for the treatment of cancer in mice. Despite an eventual lack of novelty, the manuscript is clearly interesting from a translational point of view.

In light of this, the choice of mouse cancer models that are most closely resembling human disease to demonstrate therapeutic efficacy may be of high relevance. Unfortunately, the manuscript practically exclusively relies on the B16-OVA cancer model. Moreover, several treatment controls are missing, which are relevant to understand if the efficacy of the new treatment strategy developed by the authors outcompetes previously used therapies. If those points are strengthened (see my suggestions to do so below), the manuscript will be a valuable addition to current efforts to improve cancer immunotherapy.

Main comment 1 – only one pre-clinical cancer model expressing an ectopic immuno-dominant antigen is used in the manuscript

The strongest suit of the present manuscript is its therapeutic relevance. The key findings of every Figure should, therefore, be demonstrated in one or more cancer model that closely resembles human disease. Ideally not a cell line (for example spontaneous models, if possible, or at least engraftment of primary cancer cells) and certainly not cancer cells expressing ectopic antigens.

I believe the only Figure that didn't use B16-OVA is Figure 1F-J with B16/F10 tumours, which is an experiment to "exclude" undesired effects. Probably in exactly such "excluding experiments", the authors should be consistent in tumour model and use B16-OVA.

Major comment 2 – missing controls (especially known factors as mono-treatment) and no repetition of experiments throughout the manuscript

Intratumoral injection of PolyI:C and its clinically approved derivatives is a frequently explored approach and known to be effective in pre-clinical mouse models – via a plethora of pathways (PMID: 31046839, 34824158, 34531246, 40079116). The authors only explore chemokine induction by polyI:C, which raises a few questions on the effects of polyI:C (dependent or not on eMSC-FLT3L combination treatment):

- (1) How do eMSC-FLT3L compare with intratumoral injection of rec-huFLT3L in combination with polyI:C?
- (2) Figure 4A-D and O-Q: the eMSC-FLT3L and, most importantly, the reference-control polyI:C mono-therapy groups are missing. That is important because many of those effects have previously been shown to be induced by polyI:C alone (see citations above)
- (3) Figure 5D – is the upregulation of CXCL9, CXCL11 and CCL5 unique to eMSC-FLT3L + poly(I:C) or also found be polyI:C mono-treatment?

Many experiments using treatment with eMSC-FLT3L-CXCL9-CCL5 are missing the eMSC-FLT3L control, for example:

- (1) Figure 6D-I: are those untreated compared to treated tumours? The control of eMSC-FLT3L-treated tumours is missing, especially for Fig. 6H-I.
- (2) Figure 6L-O: highly appreciated, but probably eMSC-FLT3L would have been the appropriate control here? I understand this is very challenging to repeat, but could the authors explain their choice of control here?

Finally, the majority of experiments have only been performed once. Every experiment (especially in vivo tumour-treatments) have to be performed 2 independent times to generate robust and reproducible data.

Major comment 3 – some conclusions should be revised and may require further testing

The unique induction of activated and non-exhausted CD8+ T cells in the eMSC-FLT3L+ PolyI:C-treated group is not clear, because a lot of data from the other groups are omitted/not shown. Also, it is not clear if the efficacy of eMSC-FLT3L+ PolyI:C treatment is CD8+ T cell dependent or Treg cell dependent. Please consider the following for a better understanding of the effects of the therapy:

- (1) Figure 3F – please add a quantification of all experimental groups – not only the untreated and combo
- (2) Figure 3G – please add a quantification of Tregs and IFN γ -producing CD8+ T cells for all experimental groups – not only the ratio with CD8 T cells
- (3) In Figure 4K-N, CD8 blockade should be tested alone to confirm an involvement of CD8 T cells. Also, to distinguish total CD4 from Treg cells, please block Tregs specifically using Foxp3-ko mice or a blocking antibody (for example anti-CD25).

Additional comments

In their explanations on conventional DCs (cDCs) the authors should probably adhere to the currently established nomenclature as much as possible and use the abbreviations cDC1 and cDC2 (instead of DC1 and DC2) to prevent confusion for the reader.

Define what "rec-huFLT3L" is. I understand recombinant human Flt3L.

The authors assess "Intra-tumoral delivery of eMSC-FLT3L" in Figure 1A-J. However, the experimental setup is a co-injection of tumour cells and MSCs from the beginning. The caveat of this strategy should be stated.

As the treatment with eMSC-FLT3L is of a transient nature, how to the authors envisage clinical translation? Please discuss and explain.

Figure 4 is missing panel M

Figure 4D – add percentages to representative FACS plot please

Results from Figure 4H-J should be reinterpreted: it is clear that tumour-experienced mice will develop a T cell memory and it is not clear how much the therapy contributed to that. For confirmation, tumour-resected mice would be necessary. And many more markers than displayed in Figure 4J would be necessary to confirm a resident-memory phenotype. I do not think this is a necessary message of the manuscript, but please omit any statements like "We conclude that eMSC-FLT3L +

poly(I:C) immunotherapy induce local, systemic and long- term immunity against tumor antigens in the CD8+ T cell compartment” throughout the manuscript.

Figure 7D-F: could the authors also show non-transferred, endogenous T cell infiltration into tumours of this experiment? That would be more relevant than quantifying adoptively transferred T cells.

Figure 7G-H: at day 17, tumours already have a significantly different size (according to Figure 8). Hence, the differences in T cell infiltration could be a consequence rather than a cause. T cell infiltration into tumours should be analyzed at an earlier time-point, when tumours have equal sizes between groups.

The authors state: From these results, we conclude that the recruitment of DC1 induced by eMSC-FLT3L-CXCL9-CCL5 is independent on B, T and NK/ILCs cells but dependent on FLT3 signalling. Yet, FLT3L overexpression alone is not sufficient to induce cDC1 recruitment by itself. This is puzzling and the authors should comment on this. To decipher this and understand if Flt3L is actually required (but not sufficient), the authors could consider to show if eMSC-CXCL9-CCL5 (without FLT3L expression) are able to recruit cDC1s to tumours (but that is not vital for the manuscript).

Figure 8E-F: a tumor model that is less susceptible to aPD1/aCTLA4 blockade should be used to evaluate synergy with eMSC-FLT3L-CXCL9-CCL5 treatment. I don't think this is necessary for the manuscript and the data could be removed.

Version 1:

Reviewer comments:

Reviewer #1

(Remarks to the Author)

The authors have greatly addressed each of my comments. I have no more requests nor additional clarifications.

Reviewer #2

(Remarks to the Author)

The authors have extensively addressed my comments and those of the other reviewers. I have no additional suggestions.

Reviewer #3

(Remarks to the Author)

I would like to congratulate the authors for strongly improving the manuscript and erasing most of my comments. Additional analyses and controls strengthen the scientific soundness of the data and conclusions. Also, it is appreciated that the authors validated the efficacy of eMSC-FLT3L + poly(I:C) in 3 more grafted cancer models. Yet, the key findings in the remaining paper are unfortunately still limited to B16-OVA.

However, more importantly, in light of the new data in response to my major comment 2, the comparison of Recombinant FLT3L + poly(I:C) (which are known and previously described treatments) and eMSC-FLT3L + poly(I:C) and mono-treatments, the therapeutic improvement of using eMSC-FLT3L (over recombinant Flt3l or polyI:C alone) is unfortunately overall less convincing in the revised version of the manuscript. Indeed, this direct comparison was not tested in more physiological cancer models (without ectopic antigen), where the therapeutic efficacy may actually be equal. Hence, based on the provided data in Supplementary Figure 2 (that shows no significant difference between Recombinant FLT3L + poly(I:C) and eMSC-FLT3L + poly(I:C)), I am unfortunately not convinced that there is a strong therapeutic advantage of using eMSC-FLT3L or eMSC-FLT3L-CXCL9-CCL5 of the other treatment (combinations) of polyI:C and rec-Flt3L. In my point of view, this should be better shown experimentally in at least a second physiological cancer model (not only B16-OVA). Other advantages of eMSC-FLT3L compared with rec-Flt3L could also be more closely explored or at least explained in detail to justify the use of an adoptive transfer of cells versus a recombinant protein/adjuvant in clinical practice (e.g. eventual toxicity, Treg induction in a comparative experiment, etc) – but those experiments are just a suggestion. Erasing this last doubt would really strengthen the relevance of the presented manuscript, that otherwise is very interesting.

Point-by-point response to the reviewers

We sincerely thank the reviewers for their thorough and constructive evaluation of our manuscript. In this point-by-point reply, we address each of the reviewers' comments in detail, which have helped us to substantially improve the quality of our work. The authors' responses are written in **blue**, and the revised or newly added text in the manuscript is highlighted in **green**. Figures (including supplemental figures) are numbered according to their order of appearance in the revised manuscript, with the corresponding numbers of the original figures indicated after, where applicable.

REVIEWER COMMENTS

Reviewer #1:

The work by Gorline et al. investigates the use of a cell therapy-based approach to remodel the TME through the delivery of FLT3L as a means to enhance the presence of XCR1+DC1+ cells in the tumor vicinity. Well-designed mechanistic studies were also conducted and identified CXCR3/CCR5 ligands as major players in the induction of an anti-tumoral effect. The paper is well laid-out overall.

We would like to sincerely thank the reviewer for their positive and encouraging assessment of our work. We greatly appreciate the recognition of our study's design and the mechanistic insights into the role of CXCR3/CCR5 ligands in mediating the anti-tumoral effect.

Although the study will have a great impact on the field, it does have some pitfalls that need to be addresses prior to publication. Here are the major points that need to be worked-out:

1- The authors refer to the used cells as mesenchymal stromal cells (MSCs) yet, there is no evidence provided to prove that these cells are indeed MSCs, which is surprising! For instance, the authors refer to these cells as "fibroblastic cell lines", which indicates that they are not MSCs. The authors should therefore clarify this point. If these are indeed MSCs, then they need to show their phenotypic analysis (CD31-, CD44+, CD45-, CD73+, CD90+ and CD105+).

We appreciate the reviewer's insightful comment and the opportunity to clarify this point. We fully acknowledge that the terminology surrounding *mesenchymal stem cells* and *mesenchymal stromal cells (MSCs)* can lead to confusion. In our study, we specifically used mesenchymal stromal cells (MSCs) as a platform for cell-based delivery. We chose this broader terminology because it encompasses a variety of cell types and origins.

In our work, we engineered primary mouse embryonic fibroblasts (MEFs) that were propagated from individual C57BL/6 embryos at embryonic day 13. These cells are characterized as CD45⁻, CD31⁻, CD44⁺, CD73⁻, CD90.2⁻, CD140a⁺, and gp38/PDPN⁺, as shown in the histograms provided in the **supplementary figure S1A**. Stromal cells from inguinal fat pad were used as positive control in those experiments. Nonetheless, it is difficult to ascertain negativity for CD73 and CD90, as the protein could have been cleaved by trypsin during cell detachment prior to analysis.

Sup Figure 1: Phenotypic characterization and biological activity of eMSCs

(A) Phenotypic analysis by flow cytometry of eMSCs using various classical and conventional markers for stromal and stem cells.

2- What is the expression profile of PD-1 and PD-L1 on these modified cells? This is important as pro-inflammatory signals will trigger or enhance the expression of these ICs on the cell surface.

We thank the reviewer for this comment. We fully agree that assessing PD-1 and PD-L1 expression is relevant to understand the immunomodulatory potential of our modified cells. Additional flow cytometry analysis on cultured cells revealed that these modified cells do not express PD-1 nor PD-L1, as shown by the histograms we added

in the **supplementary figure S1A** (see above). We did not, however, assess PD-1 or PD-L1 expression on eMSCs after their injection into tumors. Since these cells rapidly decline after injection (**see figure 1D-H**), their presence and effect are transient. Therefore, even if they were to express these markers, it is unlikely that this would lead to immunosuppression.

3- The authors gene-engineered their cells to express FTL3L and other chemokines. They should however provide evidence that there is no autocrine loop taking place whereby the gene-engineered cells express the cognate receptors and respond to the produced ligands.

We thank the reviewer for this comment. Additional flow cytometry analysis revealed that the eMSCs do not express FLT3 (or CXCR3 and CCR5), as shown by the histograms we added in the **supplementary figure S1A** (see above), indicating that no autocrine loop could take place. These findings are in line with the general notion that FLT3 is a receptor tyrosine kinase restricted to the hematopoietic lineage and that CXCR3 and CCR5 are important for lymphocyte traffic.

4- In figure 1, the authors seem to compare human to murine FLT3L. Is that correct? If so, why is that?

We would like to clarify that only human FLT3L was used throughout the study. We deliberately chose the human ligand because it displays high cross-reactivity between human and murine FLT3 receptors, making it suitable for use in our mouse model. In addition, using cells that already express human ligands facilitates subsequent human proof-of-concept studies in humanized mice. We acknowledge that the labeling in our figure may have been misleading. To avoid any confusion, we have removed the “hu” designation from the first figure in the revised version of the manuscript.

5- The authors always show tumor growth up to day 15. This is not impressive at all. Why not showing the growth until day 30 or 40?

Most untreated mice reached the ethical endpoint by day 15 (tumor volume of 1500 or 2000mm³, depending on mouse facility and ethical agreements). Therefore, tumor growth curves are shown only up to the last time point when all groups contained the same number of mice, ensuring fair and reliable comparison between groups. For a longer follow-up, we included survival data, which provide complementary information on treatment efficacy (**see figure 2D**, previously figure 1N). We show that eMSC-FLT3L immunotherapies actually increase survival alone (**see figure 2D**, previously

figure 1N), or in combination with immune checkpoint blockade (see figure 5P, 9G, previously figure 4Q and 8G).

6- By looking at Fig. 1L and N, one can say that the tumors that seem to be controlled by the eMSC-FLT3L + Poly I:C have regrown, which goes with my previous concern.

We agree with the reviewer's observation. Indeed, only a small proportion of mice achieved complete tumor eradication following treatment. In most cases, the eMSC-FLT3L + poly(I:C) therapy led to a marked delay in tumor growth but did not result in full tumor elimination. Consequently, tumors tended to regrow once the treatment was discontinued. These findings indicate that while the therapy exerts a strong anti-tumor effect, it may require combination with additional therapies, such as the immune checkpoint blockades (see figure 5P, previously figure 4Q), to achieve more durable tumor control and partial complete response.

7- Why choosing the B16F10 model? The authors should validate their therapeutic in vivo potency in at least 2 additional tumor models.

We thank the reviewer for this comment, and we agree that validating the therapeutic efficacy of the therapy in additional tumor models would strengthen the paper. Therefore, eMSC-FLT3L + poly(I:C) therapy was validated in three other tumor models: the breast cancer E0771 (CRL-3461), the colon carcinoma MC38 (Corbett et al. 1975) and the lung cancer TC-1 cell line (Lin et al. 1996). In each of this model, eMSC_FLT3L and poly(I:C) immunotherapy delayed tumor growth (see figure 2L, P, T) and improved survival (see figure 2N, R, V). These data have been added to a new figure, now figure 2.

Figure 2: Intratumoral engraftment of autologous mesenchymal stromal cells expressing membrane bound FLT3L stimulates anti-tumor immunity in the presence of poly(I:C).

(K, O, S) Experimental design to evaluate anti-tumoral effects of eMSC-FLT3L and poly(I:C) in MC38 (K), TC-1 (O) and E0771 (S) tumor models.

(L, P, T) Tumor growth curves (n=6-8 mice per group, one experiment)

(M, Q, U) Individual tumor size variation compared to the day of the first eMSC-FLT3L injection. (n=6-8 mice per group, one experiment, *p < 0.05, **p < 0.01 two-way ANOVA-test with Šídák's multiple comparisons test).

(N, R, V) Survival curves. (n=6-8 mice per group, one experiment, *p < 0.05, log-rank (Mantel-Cox) test).

8- The authors used the term cross-priming to refer to CD8 T-cell activation. Usually this process occurs if a given antigen is cross-presented. Have the authors provided any evidence for that?

We agree that we did not formally demonstrate that CD8⁺ T cells specific for tumor-associated antigens would depend on cross priming. Indeed, it could be hypothesized that MHC-I-peptide complexes formed on tumor cells could be transferred to XCR1⁺ cDC1s for the activation of tumor-specific T cells. This process is termed as cross dressing. To test this possibility, we attempted to develop a model of tumor devoid of

MHCI by CRISPR Cas9 inactivation of β 2-microglobulin, together with the expression of the OVA model antigen enabling to track the T cell response. However, unfortunately, these tumor cell lines were spontaneously rejected, preventing the possibility to perform our experiment. Nonetheless, the studies that highlighted a role of cross-dressing identified that it was a cDC1-independent process taking place in interferon-activated type 2 cDCs (Duong et al. 2022). By contrast, cDC1-associated genes like WDFY4 control cross presentation by cDC1s and the cross priming of CD8⁺ T cells mediating tumor control (Theisen et al. 2018). Altogether, we assume that cDC1-dependent tumor control (**see figure 3D-E**, previously figure 2D-E) is afforded by cross priming by cDC1s.

9- Why are TIM3-TCF1+CD8⁺ T_{pex} referred to as "exhausted". No convincing data are presented in that regard.

T_{pex} stands for precursor of exhausted cells, as mentioned in the text, which refers to CD8⁺ T cells possibly giving rise to exhausted cells expressing multiple inhibitory receptors (PD1, TIM3, LAG3) and losing polyfunctional cytokine response upon TCR recall. To avoid confusion, we have termed these cells as T_{SL} for stem-like T cells in the revised manuscript.

10- Why are surviving animals not rechallenged with tumors?

This is an interesting suggestion. Given the limited number of mice that completely cleared their tumors, we were unable to assess both tumor-specific memory T-cell responses and the functional efficacy of this memory. Therefore, we re-challenged the surviving animals with the SIINFEKL peptide to determine whether long-term immune memory had been established in protected mice. As shown by data in **figure 5I** (previously figure 4I), protected mice (and not control, naïve animals), displayed potent memory CD8⁺ T cell responses against a tumor-associated antigen. We conclude that tumor control is concomitant to the activation of tumor specific CD8⁺ T cell memory responses.

11- Are the eMSCs presented in Fig. 6A expressing FLT3L?

To clarify this point, several engineered cell types were generated: eMSC-FLT3L, eMSC-CXCL9, eMSC-CCL5, and eMSC-FLT3L-CXCL9-CCL5. In this figure, constructs expressing either eMSC-CXCL9 or eMSC-CCL5 were initially shown. To provide a more complete picture and avoid any misunderstanding, the results obtained

with all the different constructs were also included in the **figure 7A-C** (previously figure 6A-C).

A

Figure 7: Intratumoral delivery of eMSC co-expressing FLT3L, CXCL9 and CCL5 stimulates cDC1 infiltration in mice and humanized mice.

(A) Constructions used to produce eMSC-CCL5, eMSC-CXCL9 and eMSC-FLT3L-CXCL9-CCL5.

(B-C) Quantification by ELISA of CCL5 (B) and CXCL9 (C) released by the eMSCs transduced with the various ligands. (n=2-6 for the quantification of CCL5 and n=3-8 for the quantification of CXCL9, one experiment).

B

C

12- Fig. 6B/C are missing numbers on the Y axis. Plz fix it.

Missing numbers were added.

13- Figure 6M is confusing. We see percentages between 80-100% in all groups, which is high! The authors should better explain this figure and its relation to Fig. 6N and O.

We have used 3 antibodies to characterize human cDC1s: CD141, BTLA and XCR1. Of note, XCR1 can be upregulated during the terminal differentiation of cDC1s in tissues. As can be seen, the percentage of cDC1s expressing XCR1 within CD141⁺BTLA⁺ is high in both eMSC-containing control (100%) vs eMSC-FLT3L-CXCL9-CCL5 containing plugs (92%). However, the % of cDC1s within total CD45⁺ is highly increased in eMSC-FLT3L-CXCL9-CCL5 group as compared to eMSC-control (**see figure 7P**, previously figure 6O). Indeed, the treated eMSC-FLT3L-CXCL9-CCL5 group consistently shows higher percentages throughout the gating steps (**see figure 7N**, previously figure 6M), supporting the overall trend and the results shown in **figure 7O-P** (previously figure 6N-O).

14- Fig. 8F is disappointing especially if compared to the first set of in vivo data in Fig. 1. Once would have expected better tumor control at that point... In fact, the authors should combine the eMSC-FLt3L + Poly I:C with the anti-PD1 and anti-CTLA4 to see if this combo therapy induces stronger or more effective anti-tumoral responses.

Yes, indeed! And the result can be found in **figure 5N-P** (previously figure 4O-Q). The eMSC-FLT3L + poly(I:C) immunotherapy indeed induces stronger and more effective anti-tumoral responses when combined with ICB, as compared with the eMSC-FLT3L-CXCL9-CCL5 treatment. This outcome is somewhat expected, as poly(I:C) not only promotes chemokine secretion but also activates multiple signaling pathways, including the induction of type I IFNs downstream TLR3 in cDC1s and MDA5 in other cells. Type I IFN signaling, in turn activates the immunogenic maturation of cDCs (Longhi et al. 2009) (see **figure 3B-C, 4B-C**, previously figure 2B-C, 3B-C).

15- The discussion is very thin and needs more development within the context of cancer immunotherapy. In this current format, it looks more like a summary of findings. Plz discuss the drawback, limits, pitfalls of this technology and put it in a clinical context or compare it to other experimental approaches.

16- Why was the use of MSCs in the context of cancer vaccination not discussed? There are series of papers describing the use of gene-engineered or pharmacologically reprogrammed MSCs in the context of cancer vaccination. Such concept should be discussed as well.

We agree with the comment. The discussion has been completely rewritten, highlighting: i) the specific advantage of eMSCs for the local delivery of FLT3L and other factors within tumors; ii) practical and technological bottleneck in the road to translate the system to an immunotherapeutic intervention in clinical settings. Please, find below in green the revised paragraphs, taking these comments into account.

“In this study, we developed a novel immunotherapeutic platform based on the intratumoral engraftment of ex vivo–engineered, autologous mesenchymal stromal cells (eMSCs). In the first part, we demonstrate a potent therapeutic synergy between eMSCs expressing a membrane-bound form of FLT3 ligand (FLT3L) and the adjuvant poly(I:C). These findings are consistent with previous reports showing that combined administration of recombinant FLT3L and poly(I:C) enhances cDC1 expansion and maturation, promotes tumor regression, and improves responses to immune checkpoint blockade (Salmon et al. 2016; Hammerich et al. 2019; Oba et al. 2020). However, systemic delivery or secretion of FLT3L has been shown to perturb hematopoiesis and induce peripheral and intratumoral Treg expansion (Darrasse-Jze et al. 2009; Swee et al. 2009). Moreover, the requirement for repeated injections of poly(I:C) and/or FLT3L in these studies raises safety and feasibility concerns for clinical translation. This calls for the development of delivery methods limiting FLT3L protein production to the TME.

Here, we show that intratumoral delivery of eMSCs expressing membrane bound FLT3L elicits potent anti-tumor immunity while restricting FLT3L activity to the tumor site, thereby avoiding systemic toxicities. Treated mice exhibited no elevation in serum FLT3L levels or Treg frequencies, and no evidence of weight loss or liver injury, suggesting that this localized, cell-based strategy mitigates systemic adverse effects commonly associated with DC-targeted immunotherapies (Siwicki et al. 2021). Autologous eMSC-FLT3L proposes an alternative option to the intratumoral injection of recombinant FLT3L (Salmon et al. 2016), viral vectors encoding FLT3L (Snchez-Paulete et al. 2018), CAR-T cell transduced FLT3L (Lai et al. 2020) and Herpes Virus oncolytic virus encoding FLT3L (Wan et al. 2025). Indeed, local retention of eMSC has the advantage to restrict expression of FLT3L to the TME, a feature not afforded by CAR T cell mediated systemic delivery (Lai et al. 2020). In addition, our system, in contrast to viral vectors, is not susceptible to immunity.

This study expands the therapeutic scope of engineered mesenchymal stromal cells (eMSCs) to include modulation of the dendritic cell (DC) lineage. Previous studies have shown that IL-2-based, T cell-targeting eMSCs confer substantial benefits by promoting the expansion of non-exhausted PD-1⁺TIM-3⁻ CD8⁺ T cells (Stagg et al. 2004; Bae et al. 2022). Likewise, eMSCs engineered to express type I interferons or IL-12 have been reported to enhance the infiltration and cytotoxic effector functions of intratumoral CD8⁺ T cells (Ryu et al. 2011; Choi et al. 2017; ulach et al. 2021; Zhang et al. 2022) and NK cells (Ren et al. 2008). It remains to be determined whether eMSCs that act directly on lymphocytes could synergize with DC-targeting eMSCs such as those developed in this study. Exploring such combinatorial strategies may further amplify both antigen presentation and effector responses, providing a versatile platform for next-generation cell-based immunotherapies.

An important finding of this study is that elevation of FLT3L levels alone in the TME is not sufficient to induce DC recruitment and bring immunotherapeutic benefits. We found that poly(I:C) addition synergizes with eMSC-FLT3L. Therefore, we next investigated the mechanisms underlying the enhanced efficacy of the eMSC-FLT3L + poly(I:C) therapy and identified a marked upregulation of multiple chemokines within the tumor microenvironment (TME), notably CXCL9 and CCL5, which serve as ligands for CXCR3 and CCR5, respectively. Pharmacological blockade experiments demonstrated that both CXCR3 and CCR5 are essential for the intratumoral accumulation of cDC1s and for the establishment of effective antitumor immunity. To minimize the potential toxicities associated with poly(I:C), we replaced poly(I:C) by the co-expression of CXCL9 and CCL5 on eMSC-FLT3L. This substitution similarly enhanced the infiltration of cDC1s, T cells, and NK cells, while maintaining robust

antitumor activity. These results align with previous studies highlighting the role of CXCR3- and CCR5-mediated recruitment in sustaining immune effector infiltration into the TME (Chow et al. 2019; Dangaj et al. 2019). Importantly, although CXCR3 and CCR5 are expressed on circulating pre-DCs (Cook et al. 2018) as well as on activated T and NK cells, our findings indicate that T and NK cells are dispensable for the CXCL9/CCL5-driven recruitment of DCs. It is known that intratumoral chemokines may derive from multiple sources, including NK cells and cDC1s themselves (Böttcher et al. 2018; Chow et al. 2019; Meiser et al. 2023) and are essential to support the sustained infiltration of cDC1 within the TME. Here, we show that this pathway of cDC1 recruitment associated to immunogenic tumors is activatable in the context of poorly immunogenic tumors by eMSC-FLT3L-CCL5-CXCL9 engraftment. We demonstrate that local chemokine release by eMSC increase the interaction of cDC1s with eMSC-FLT3L. Translatability of the approach to human cDC1s is demonstrated because eMSC-FLT3L-CCL5-CXCL9 treatment supports the local recruitment of human cDC1s in the dermis of BRGSF immunodeficient mice reconstituted with human CD34⁺ hematopoietic stem cells. However, the identity of the circulating progenitor population responding to this therapy remains to be elucidated, as several candidate precursors have been proposed as human equivalents of murine pre-DCs (Lee et al. 2015; Villani et al. 2017; See et al. 2017).”

“The prospect of locally implanted, eMSCs acting as versatile, sustained factories for therapeutic molecule delivery is highly attractive for cancer immunotherapy. This is particularly relevant for chemokines, which generate spatially organized gradients governing leukocyte trafficking and tissue compartmentalization. Nonetheless, several technological and translational challenges must be addressed to realize the clinical potential of eMSCs.

First, standardized and GMP-compatible protocols for autologous eMSCs isolation, engineering, and production must be established. Such methods should ensure reproducible recovery of stromal populations of controlled complexity that are amenable to *ex vivo* expansion, viral transduction, and post-transduction enrichment. Although the present study primarily employed murine embryonic fibroblasts as an MSC source, we demonstrate that adipose-derived stromal cells can be expanded, transduced, and used to generate immunologically competent eMSCs. Other groups have successfully implemented bone marrow-derived eMSCs for immunotherapy (Stagg et al. 2004; Ren et al. 2008; ulach et al. 2021; Zhang et al. 2022). Validation in human systems will be critical for clinical translation. Furthermore, the phenotypic heterogeneity resulting from *ex vivo* expansion and transduction remains to be fully characterized. Despite homogeneous transgene expression (e.g., via IRES-linked

fluorescent reporters), we can not exclude that this cellular preparation is functionally heterogeneous and that only a subset of eMSCs may be functionally active *in vivo*. Integration of single-cell RNA sequencing and spatial transcriptomics in eMSC-treated tumors could provide essential insight into stromal diversity, fate, and functional states.

Second, optimization of eMSC delivery methods could improve therapeutic persistence and efficacy. In our current model, intratumoral injection of cell suspensions resulted in transient eMSC persistence, with only a minor fraction detectable beyond ten days. Future studies should aim at enhancing eMSC residency through tissue engineering strategies, including synthetic scaffolds or extracellular matrix integration (Choi et al. 2017). Cell-intrinsic manipulation of stress pathway might also offer some opportunities to improve the survival of eMSC in the TME. For instance, NQO1 expression might improve resistance to oxidative stress (Bae et al. 2022). Would these manipulations improve persistence, it would be important to determine whether prolonged eMSC survival improves T and NK cell-dependent tumor control. While intratumoral delivery provides localized control, it may not be feasible for all cancer types. Developing systemic delivery approaches for eMSCs could extend therapeutic reach to deep or inaccessible lesions while improving patient tolerability. Indeed, intravital imaging studies have shown that intravenously administered MSCs exhibit intrinsic tropism for sites of tissue remodeling, including wounds and solid tumors (Kidd et al. 2009; Kalimuthu et al. 2017). The molecular mechanisms underlying this selective homing remain poorly defined but could be exploited to deliver eMSCs to otherwise unreachable tumor sites.

Third, eMSC-based therapies designed to stimulate DC-dependent antitumor immunity must also incorporate cues that promote antigen release and DC maturation, the two essential steps required to initiate the cancer-immunity cycle. In our study, eMSCs expressing FLT3L and CXCL9/CCL5 effectively enhanced local DC infiltration but did not efficiently drive DC maturation, potentially limiting migration to tumor-draining lymph nodes and cross-priming of CD8⁺ T cells. Embedding maturation-inducing signals within eMSCs may overcome this limitation. In parallel, promoting local tumor cell death could enhance antigen availability for DC uptake. One promising avenue involves combining eMSCs with oncolytic virotherapy, which induces immunogenic cell death and antigen release. eMSCs have been widely explored as delivery vehicles for oncolytic viruses, leveraging their tumor-homing capacity (Pereboeva and Curiel 2004; Hakkarainen et al. 2007). This approach has been implemented for several viral platforms, including herpes simplex virus (Du et al. 2017), myxoma virus (Josiah et al. 2010), Newcastle disease virus (Kazimirsky et al. 2016), measles virus (Sonabend et al. 2008), and adenovirus (Sonabend et al. 2008; Yoon et

al. 2019). We propose that eMSC-based oncolytic virotherapy could be synergistically combined with DC-targeting eMSCs, as suggested by Svensson-Arvelund *et al.*, who observed cooperation between recombinant FLT3L administration and NDV-mediated oncolysis (Svensson-Arvelund *et al.* 2022). Moreover, type I IFN-releasing eMSCs could complement DC-directed eMSCs to further enhance local immune activation (Ren *et al.* 2008; Choi *et al.* 2017; Zhang *et al.* 2022).

In summary, while technical and regulatory limitations remain, the translational potential of eMSC-based immunotherapies is considerable. Their inherent tumor tropism, compatibility with *ex vivo* genetic engineering, and capacity for localized, sustained delivery make eMSCs a promising cellular platform for next-generation combination immunotherapies aimed at amplifying both innate and adaptive antitumor immunity.”

Reviewer #2:

This manuscript by Dr. Guernonprez and colleagues presents an elegant and innovative strategy for enhancing anti-tumor immunity through intra-tumoral delivery of engineered mesenchymal stromal cells expressing FLT3L and the chemokines CXCL9 and CCL5. The experiments are well-designed, and the mechanistic insights into cDC1 recruitment and T cell priming are compelling. I particularly appreciated the use of both mouse and humanized models, which enhances the translational relevance of the findings.

We sincerely thank the reviewer for their positive and encouraging comments. We greatly appreciate their recognition of our experimental design, the mechanistic insights into cDC1 recruitment and T cell priming, as well as the translational relevance provided by the use of both mouse and humanized models.

Nevertheless, a few key points could be addressed to strengthen the manuscript:

1- Cellular source of chemokines: The cellular source of CXCL9 and CCL5 following poly(I:C) treatment remains undefined. Further characterization of this axis would help clarify the underlying mechanism.

We thank the reviewer for bringing this interesting question. While identifying the precise cellular source of CXCL9 and CCL5 following poly(I:C) treatment would indeed be interesting, it is beyond the scope of the present study, which focuses on the functional impact of engineered eMSCs on cDC1 proliferation and maturation and T cell activation. We therefore did not investigate this aspect in the current work. Multiple studies identify DCs and cDC1 specifically, as important producers of CXCL9 (Spranger et al. 2017; Chow et al. 2019; Dangaj et al. 2019). However, multiple sources might participate to CCL5 production including T cells, N cells and macrophages (Böttcher et al. 2018). These aspects have been added to the discussion. Please, find below in green the revised paragraphs, taking this comment into account.

“To minimize the potential toxicities associated with poly(I:C), we replaced poly(I:C) by the co-expression of CXCL9 and CCL5 on eMSC-FLT3L. This substitution similarly enhanced the infiltration of cDC1s, T cells, and NK cells, while maintaining robust antitumor activity. These results align with previous studies highlighting the role of CXCR3- and CCR5-mediated recruitment in sustaining immune effector infiltration into the TME (Chow et al. 2019; Dangaj et al. 2019). Importantly, although CXCR3 and CCR5 are expressed on circulating pre-DCs (Cook et al. 2018) as well as on activated

T and NK cells, our findings indicate that T and NK cells are dispensable for the CXCL9/CCL5-driven recruitment of DCs. It is known that intratumoral chemokines may derive from multiple sources, including NK cells and cDC1s themselves (Böttcher et al. 2018; Chow et al. 2019; Meiser et al. 2023) and are essential to support the sustained infiltration of cDC1 within the TME. Here, we show that this pathway of cDC1 recruitment associated to immunogenic tumors is activatable in the context of poorly immunogenic tumors by eMSC-FLT3L-CCL5-CXCL9 engraftment. We demonstrate that local chemokine release by eMSC increase the interaction of cDC1s with eMSC-FLT3L.”

2- Safety considerations: Although systemic cytokine levels appear unaffected, the long-term biodistribution, persistence, and potential immunogenicity of the eMSCs warrant additional safety evaluation.

We agree that this is a key question, that was overlooked in the first version of the manuscript.

- i) **eMSC biodistribution:** To further assess the long-term biodistribution and persistence of eMSCs, nano-luciferase–expressing eMSCs were generated and monitored for 8 days in the B16-OVA model using bioluminescence imaging. We observed that eMSCs persisted within the tumor for at least 8 days, although at very low levels, and that their activity remained restricted to the tumor site. These findings suggest that, once injected into the tumor, eMSCs do not disseminate throughout the body but rather remain localized. These data have been added to the **figure 1F-H**.

Figure 1: Intratumoral delivery of engineered mesenchymal stromal cells expressing membrane bound FLT3L does not alter intratumoral and systemic cDC1 populations.

(F) Experimental design to evaluate the biodistribution and persistence of eMSCs within B16-OVA tumors. 1×10^6 eMSC-NanoLuc were injected *i.t.* on day 0. On days 0 (4h after the injection), 4, and 8, mice received an intravenous injection of $0.44 \mu\text{moles}$ of the fluorofurimazine substrate, followed by bioluminescence imaging using the IVIS system.

(G) Bioluminescence imaging and quantification of the average radiance of eMSC-NanoLuc at days 0, 4 and 8 following their injection.

(H) Bioluminescence imaging and quantification of the average radiance of eMSC-NanoLuc in individual organs of mice having received eMSC-NanoLuc. He:heart; Li:liver; Ki:kidney, Sp:spleen, Lu:lungs, Tu:tumor, tdLN:tumor-draining lymph node, Mu:muscle.

ii) **Systemic toxicity assessment:** To further assess if eMSC immunotherapy would induce a systemic, deleterious inflammatory response, we compared the toxicity of our therapy with that induced by the administration of a high (toxic) dose of anti-CD40 antibody (Blake et al. 2021; Siwicki et al. 2021; Salomon et al. 2022). We assessed serum levels of ALT and AST, markers of hepatocellular injury, together with assessment of liver damage using H&E to assess liver necrotic regions. Changes in body weight were assessed over time. We observed that treatment with anti-CD40 antibody induced significant toxicity (high levels of AST and ALT, liver damage), whereas eMSC-based therapies did not appear to be toxic. These data have been added to a new **figure 2E-G** and to the **supplementary figure S2C-D**.

Altogether, we conclude that eMSC do not induce a detectable level of systemic inflammation.

Figure 2: Intratumoral engraftment of autologous mesenchymal stromal cells expressing membrane bound FLT3L stimulates anti-tumor immunity in the presence of poly(I:C).

(E) Experimental design to evaluate the toxicity of the therapy in B16-OVA bearing mice. On day 7, eMSC-FLT3L were i.t. injected. On day 9, mice were treated i.t. with poly(I:C) or i.p. with an anti-CD40 antibody. Serum was collected two days later.

(F) Body weight loss of mice throughout the experiment. (n=3-4 mice per group, one experiment, *p < 0.05, **p < 0.01, two-way ANOVA-test with Tukey's multiple comparisons test).

(G) Level of ALT in serum collected 48h after poly(I:C) or anti-CD40 treatment.

Sup Figure 2: eMSC-FLT3L + poly(I:C) immunotherapy is more effective than recombinant FLT3L + poly(I:C) treatment, without inducing toxicity.

(C) Level of AST in serum collected 48h after poly(I:C) or anti-CD40 treatment.

(D) H&E staining of fixed liver tissue from mice treated with eMSC-FLT3L + poly(I:C) therapy or anti-CD40. Necrotic lesions are shown with the dashed black lines.

3- Model generalizability: The study would benefit from validation in additional tumor models to demonstrate generalizability beyond B16-OVA. This model is somewhat peculiar, as most OVA-expressing tumors are highly immunogenic and are often spontaneously rejected in immunocompetent mice. This limitation could be addressed in a follow-up study in which intra-tumoral delivery of FLT3L-expressing eMSCs is tested in other transplantable or spontaneous tumor models.

We thank the reviewer for this comment, and we agree that validating the therapeutic efficacy of the therapy in additional tumor models would strengthen the paper. Therefore, eMSC-FLT3L + poly(I:C) therapy was validated in three other tumor models: the breast cancer E0771 (CRL-3461), the colon carcinoma MC38 (Corbett et al. 1975) and the lung cancer TC1 cell line (Lin et al. 1996). In each of this model, eMSC_FLT3L and poly(I:C) immunotherapy delayed tumor growth (see figure 2L, P, T) and improved survival (see figure 2N, R, V). These data have been added to a new figure, now figure 2.

Figure 2: Intratumoral engraftment of autologous mesenchymal stromal cells expressing membrane bound FLT3L stimulates anti-tumor immunity in the presence of poly(I:C).

(K, O, S) Experimental design to evaluate anti-tumoral effects of eMSC-FLT3L and poly(I:C) in MC38 (K), TC-1 (O) and E0771 (S) tumor models.

(L, P, T) Tumor growth curves (n=6-8 mice per group, one experiment)

(M, Q, U) Individual tumor size variation compared to the day of the first eMSC-FLT3L injection. (n=6-8 mice per group, one experiment, *p < 0.05, **p < 0.01 two-way ANOVA-test with Šídák's multiple comparisons test).

(N, R, V) Survival curves. (n=6-8 mice per group, one experiment, *p < 0.05, log-rank (Mantel-Cox) test).

4- Mechanistic synergy with checkpoint blockade: The authors should discuss more thoroughly the mechanism by which this therapy synergizes with checkpoint blockade.

We thank the reviewer for this valuable comment. In response, we have incorporated a new paragraph into the discussion section of the revised manuscript. The revised text, addressing this point, is in green below.

“When combining eMSC-FLT3L immunotherapies to aPD-1/aCTLA-4 ICB immunotherapy, we manage to partially overcome primary resistance to immune checkpoint blockade and extend survival. We hypothesize that at least 3 mechanisms

could account for this effect of eMSC-FLT3L immunotherapy:

- i) eMSCs might activate DCs engaging with poorly differentiated TCF1⁺ T_{SL} CD8⁺ T cells enabling to generate ICB-responding pool within the tumors (Im et al. 2016; Siddiqui et al. 2019). In support of this, multiple studies have identified that mature DCs (and cDC1s in particular) participate to a niche supporting TCF1⁺ poorly differentiated T cells (Schenkel et al. 2021; Meiser et al. 2023; Magen et al. 2023; Chen et al. 2024). This might possibly happen via the accumulation of mature DCs within TLS for instance (Mattiuz et al. 2024).
- ii) eMSC might promote the formation of a pool of activated DCs within the tumor engaging intratumoral effector CD8⁺ T or NK cells. Garris et al. have shown that ICB rapidly activates intratumoral DCs via the γ to become IL12/IL15⁺ DCs (Garris et al. 2018). By increasing DC density, eMSC could promote the generation of intratumoral IL12/IL15/CXCL16⁺ perivascular DCs which have been shown to play a key role in maintaining the fitness of intratumoral CXCR6⁺ CD8⁺ T cell effectors (Di Pilato et al. 2021).
- iii) Lastly, eMSC might increase the number of migratory DCs cross-priming novel CD8⁺ T cells in tumor draining lymph nodes. This might help broadening and diversifying T cell response which is a feature of productive ICB immunotherapy (Yost et al. 2019; Liu et al. 2022). In support of this notion, we have evidenced that eMSC-FLT3L + poly(I:C) therapy increases the flux of migratory DCs to tumor-draining lymph nodes” (see figure 4B-C).

5- Clinical implications: The authors should discuss better the translational potential of eMSC therapy, as the intratumor route of delivery would not be amenable to most human cancers.

Addressing these points will enhance the impact of the study. However, given the novelty and important implications of the work, I recommend the manuscript undergo minor revision prior to acceptance.

We agree with the comment. The discussion has been extensively modified, highlighting practical and technological bottleneck in the road to translate the system to an immunotherapeutic intervention in clinical settings. Please, find below in green the revised paragraphs, taking this comment into account.

“The prospect of locally implanted, eMSCs acting as versatile, sustained factories for therapeutic molecule delivery is highly attractive for cancer immunotherapy. This is particularly relevant for chemokines, which generate spatially organized gradients governing leukocyte trafficking and tissue compartmentalization.

Nonetheless, several technological and translational challenges must be addressed to realize the clinical potential of eMSCs.

First, standardized and GMP-compatible protocols for autologous eMSCs isolation, engineering, and production must be established. Such methods should ensure reproducible recovery of stromal populations of controlled complexity that are amenable to *ex vivo* expansion, viral transduction, and post-transduction enrichment. Although the present study primarily employed murine embryonic fibroblasts as an MSC source, we demonstrate that adipose-derived stromal cells can be expanded, transduced, and used to generate immunologically competent eMSCs. Other groups have successfully implemented bone marrow-derived eMSCs for immunotherapy (Stagg et al. 2004; Ren et al. 2008; ulach et al. 2021; Zhang et al. 2022). Validation in human systems will be critical for clinical translation. Furthermore, the phenotypic heterogeneity resulting from *ex vivo* expansion and transduction remains to be fully characterized. Despite homogeneous transgene expression (e.g., via IRES-linked fluorescent reporters), we can not exclude that this cellular preparation is functionally heterogeneous and that only a subset of eMSCs may be functionally active *in vivo*. Integration of single-cell RNA sequencing and spatial transcriptomics in eMSC-treated tumors could provide essential insight into stromal diversity, fate, and functional states.

Second, optimization of eMSC delivery methods could improve therapeutic persistence and efficacy. In our current model, intratumoral injection of cell suspensions resulted in transient eMSC persistence, with only a minor fraction detectable beyond ten days. Future studies should aim at enhancing eMSC residency through tissue engineering strategies, including synthetic scaffolds or extracellular matrix integration (Choi et al. 2017). Cell-intrinsic manipulation of stress pathway might also offer some opportunities to improve the survival of eMSC in the TME. For instance, NQO1 expression might improve resistance to oxidative stress (Bae et al. 2022). Would these manipulations improve persistence, it would be important to determine whether prolonged eMSC survival improves T and NK cell-dependent tumor control. While intratumoral delivery provides localized control, it may not be feasible for all cancer types. Developing systemic delivery approaches for eMSCs could extend therapeutic reach to deep or inaccessible lesions while improving patient tolerability. Indeed, intravital imaging studies have shown that intravenously administered MSCs exhibit intrinsic tropism for sites of tissue remodeling, including wounds and solid tumors (Kidd et al. 2009; Kalimuthu et al. 2017). The molecular mechanisms underlying this selective homing remain poorly defined but could be exploited to deliver eMSCs to otherwise unreachable tumor sites.

Third, eMSC-based therapies designed to stimulate DC-dependent antitumor immunity must also incorporate cues that promote antigen release and DC maturation, the two essential steps required to initiate the cancer-immunity cycle. In our study, eMSCs expressing FLT3L and CXCL9/CCL5 effectively enhanced local DC infiltration but did not efficiently drive DC maturation, potentially limiting migration to tumor-draining lymph nodes and cross-priming of CD8⁺ T cells. Embedding maturation-inducing signals within eMSCs may overcome this limitation. In parallel, promoting local tumor cell death could enhance antigen availability for DC uptake. One promising avenue involves combining eMSCs with oncolytic virotherapy, which induces immunogenic cell death and antigen release. eMSCs have been widely explored as delivery vehicles for oncolytic viruses, leveraging their tumor-homing capacity (Pereboeva and Curiel 2004; Hakkarainen et al. 2007). This approach has been implemented for several viral platforms, including herpes simplex virus (Du et al. 2017), myxoma virus (Josiah et al. 2010), Newcastle disease virus (Kazimirsky et al. 2016), measles virus (Sonabend et al. 2008), and adenovirus (Sonabend et al. 2008; Yoon et al. 2019). We propose that eMSC-based oncolytic virotherapy could be synergistically combined with DC-targeting eMSCs, as suggested by Svensson-Arvelund *et al.*, who observed cooperation between recombinant FLT3L administration and NDV-mediated oncolysis (Svensson-Arvelund et al. 2022). Moreover, type I IFN-releasing eMSCs could complement DC-directed eMSCs to further enhance local immune activation (Ren et al. 2008; Choi et al. 2017; Zhang et al. 2022).

In summary, while technical and regulatory limitations remain, the translational potential of eMSC-based immunotherapies is considerable. Their inherent tumor tropism, compatibility with *ex vivo* genetic engineering, and capacity for localized, sustained delivery make eMSCs a promising cellular platform for next-generation combination immunotherapies aimed at amplifying both innate and adaptive antitumor immunity.”

Minor specific points:

In the experiments shown in Figures 3B, S2, and S3, the observed effects on cDC1 maturation and T cell activation could be attributed to poly I:C treatment alone. The authors should include a comparison between eMSC + poly I:C and poly I:C alone to demonstrate that the presence of eMSCs is essential for enhancing cDC1 maturation and IFN- γ secretion by T cells.

The figures mentioned (now **figure 4B, S3A and S4C-F**) already include the control condition “eMSC-Ctrl + poly(I:C)”, which can be considered equivalent to “poly(I:C) alone”, since eMSC-Ctrl do not exert any specific immunomodulatory activity. These control cells express only a fluorescent reporter and do not express

any additional ligands. Therefore, the observed effects cannot be attributed to poly(I:C) treatment alone but rather to the combination with the engineered eMSCs expressing FLT3L.

Reviewer #3:

The manuscript by Gorline et al analyses the anti-cancer efficacy of engineered mesenchymal stem cells expressing factors that enhance DC and T/N cell activation/infiltration in a mouse tumour model harbouring an ectopic immunodominant antigen. The use of humanized mice/human cells is highly appreciated and the authors perform a comprehensive analysis of the majority of relevant aspects of DC-mediated induction of anti-cancer T cell immunity. The authors basically report the effective repurposing of existing knowledge, the combination of a previously reported system (eMSCs) with known anti-cancer factors for the treatment of cancer in mice. Despite an eventual lack of novelty, the manuscript is clearly interesting from a translational point of view.

In light of this, the choice of mouse cancer models that are most closely resembling human disease to demonstrate therapeutic efficacy may be of high relevance.

Unfortunately, the manuscript practically exclusively relies on the B16-OVA cancer model. Moreover, several treatment controls are missing, which are relevant to understand if the efficacy of the new treatment strategy developed by the authors outcompetes previously used therapies. If those points are strengthened (see my suggestions to do so below), the manuscript will be a valuable addition to current efforts to improve cancer immunotherapy.

We sincerely thank the reviewer for their thoughtful and detailed assessment of our work. We appreciate their recognition of the comprehensive analysis we performed and the translational relevance of using humanized mice. We have carefully considered the reviewer's suggestions and have conducted additional experiments and analyses to expand the relevance of our findings, as detailed below.

Main comment 1 – only one pre-clinical cancer model expressing an ectopic immunodominant antigen is used in the manuscript.

The strongest suit of the present manuscript is its therapeutic relevance. The key findings of every Figure should, therefore, be demonstrated in one or more cancer model that closely resembles human disease. Ideally not a cell line (for example spontaneous models, if possible, or at least engraftment of primary cancer cells) and certainly not cancer cells expressing ectopic antigens.

We thank the reviewer for this comment, and we agree that validating the therapeutic efficacy of the therapy in additional tumor models would strengthen the paper. Therefore, eMSC-FLT3L + poly(I:C) therapy was validated in three other tumor models:

the breast cancer E0771 (CRL-3461), the colon carcinoma MC38 (Corbett et al. 1975) and the lung cancer TC1 cell line (Lin et al. 1996). In each of this model, eMSC_FLT3L and poly(I:C) immunotherapy delayed tumor growth (see figure 2L, P, T) and improved survival (see figure 2N, R, V). These data have been added to a new figure, now figure 2.

Figure 2: Intratumoral engraftment of autologous mesenchymal stromal cells expressing membrane bound FLT3L stimulates anti-tumoral immunity in the presence of poly(I:C).

(K, O, S) Experimental design to evaluate anti-tumoral effects of eMSC-FLT3L and poly(I:C) in MC38 (K), TC-1 (O) and E0771 (S) tumor models.

(L, P, T) Tumor growth curves (n=6-8 mice per group, one experiment)

(M, Q, U) Individual tumor size variation compared to the day of the first eMSC-FLT3L injection. (n=6-8 mice per group, one experiment, *p < 0.05, **p < 0.01 two-way ANOVA-test with Šídák's multiple comparisons test).

(N, R, V) Survival curves. (n=6-8 mice per group, one experiment, *p < 0.05, log-rank (Mantel-Cox) test).

I believe the only Figure that didn't use B16-OVA is Figure 1F-J with B16/F10 tumors, which is an experiment to "exclude" undesired effects. Probably in exactly such "excluding experiments", the authors should be consistent in tumour model and use B16-OVA.

We thank the reviewer for this comment. This experiment was done again using the B16-OVA model and similar results were obtained, as shown in **supplementary figure S1C**. Overall, we conclude that eMSC_FLT3L does not impact on systemic FLT3L levels.

Sup Figure 1: Phenotypic characterization and biological activity of eMSCs.

(C) Circulating huFLT3L levels measured by ELISA in mice serum 4 days after eMSCs injection. (n=3-4 mice per group in one experiment. *p < 0.05, Kruskal-Wallis test with Dunn's multiple comparisons).

Major comment 2 – missing controls (especially known factors as mono-treatment) and no repetition of experiments throughout the manuscript

Intratumoral injection of PolyI:C and its clinically approved derivatives is a frequently explored approach and known to be effective in pre-clinical mouse models – via a plethora of pathways (PMID: 31046839, 34824158, 34531246, 40079116).

We agree with the comment of the reviewer since multiple studies indeed highlighted some effects of poly(I:C) as a monotherapy. Here is a detailed response to the questions.

The authors only explore chemokine induction by polyI:C, which raises a few questions on the effects of polyI:C (dependent or not on eMSC-FLT3L combination treatment): How do eMSC-FLT3L compare with intratumoral injection of rec-huFLT3L in combination with polyI:C?

We thank the reviewer for this insightful comment. Intratumoral injection of recombinant FLT3L also delayed tumor growth and improved survival, although these effects were more pronounced with eMSC-FLT3L treatment, as shown with the tumor growth (see **supplementary figure S2A**) and survival curves (see **supplementary figure S2B**).

We conclude that eMSC-FLT3L bring a significant although partial advantage upon recombinant FLT3L for the local delivery of FLT3L within the TME. This advantage might be more pronounced in slightly growing tumors and if eMSCs persistence within the TME might be improved.

Sup Figure 2: eMSC-FLT3L + poly(I:C) immunotherapy is more effective than recombinant FLT3L + poly(I:C) treatment, without inducing toxicity.

(A) Tumor growth curves until day 15 (n=10 mice per group, two independent experiments, ****p < 0.0001, two-way ANOVA-test with Tukey's multiple comparisons test).

(B) Survival curves. (n=10 mice per group in two independent experiments, **p < 0.01, ****p < 0.0001, log-rank (Mantel-Cox) test)

Figure 4A-D and O-Q: the eMSC-FLT3L and, most importantly, the reference-control poly(I:C) mono-therapy groups are missing. That is important because many of those effects have previously been shown to be induced by poly(I:C) alone (see citations above).

We thank the reviewer for this valuable comment. In response, we have repeated experiments and have added the relevant controls. Quantifications for all experimental groups (eMSC-Ctrl, eMSC-FLT3L, poly(I:C), and eMSC-FLT3L + poly(I:C)) have been added to the **figure 5D-E** and previous figure quantifying the untreated and poly(I:C) groups only (previously figure 4D-E) has been moved to the **supplementary figure S5A-B**. We conclude that the effect we observe is indeed uniquely induced by the eMSC-FLT3L + poly(I:C) immunotherapy in our model.

Figure 5: Intratumoral delivery of eMSC-FLT3L and poly(I:C) stimulates the infiltration of T and NK cells within tumor beds that are required for immunotherapy efficiency and overcomes primary resistance to PD-1/CTLA-4 blockade.

(D-E) Representative flow cytometry plots (D) and quantification (E) of activated (T_{ACT}), exhausted (T_{EX}), and stem-like (T_{SL}) CD8⁺ T cells. (n=4 mice per group, one experiment, *p < 0.05, **p < 0.01, one-way ANOVA with Tukey's multiple comparisons test).

Regarding the figure combining the eMSC-FLT3L + poly(I:C) therapy to the ICB (figure 5N-P, previously figure 4O-Q), we showed that in our model, eMSC-FLT3L and poly(I:C) monotherapies do not independently delay tumor growth. The purpose of this experiment was to test the combination of an effective therapy with immunotherapies. Moreover, given that the efficacy of ICB therapy depends on enhanced intratumoral T-cell infiltration, and that only the eMSC-FLT3L + poly(I:C) combination promoted an increase in CD8⁺ T cells, our subsequent analyses focused on this combination in the context of ICB.

Figure 5D – is the upregulation of CXCL9, CXCL11 and CCL5 unique to eMSC-FLT3L + poly(I:C) or also found by poly(I:C) mono-treatment?

We thank the reviewer for this comment. In our model, the upregulation of CXCL9, CXCL11 and CCL5 is more pronounced with eMSC-FLT3L + poly(I:C). However, additional experiment including the poly(I:C) monotherapy identifies that poly(I:C) acts as an inducer of CXCL9 chemokine on its own (see supplementary figure S6A).

Sup Figure 6: poly(I:C) monotherapy is sufficient to upregulate CXCL9 expression in tumor homogenates.

(A) Murine CXCL9 ELISA done on tumor homogenates 24h after poly(I:C) injection. (n=3-4, one experiment).

Nonetheless, FLT3L and chemokines both synergize for intratumoral cDC1 infiltration and expansion (see figure 3C, previously figure 2C).

Many experiments using treatment with eMSC-FLT3L-CXCL9-CCL5 are missing the eMSC-FLT3L control, for example:

Figure 6D-I: are those untreated compared to treated tumors? The control of eMSC-FLT3L-treated tumours is missing, especially for Fig. 6H-I.

Upon reviewer request, we have added the missing controls. **Figure 7G-H** (previously figure 6F-G) and **supplementary figure S7A** (previously figure 6H), now compares eMSC-FLT3L to eMSC-FLT3L-CXCL9-CCL5. As shown in the first figures, the untreated, eMSC-Ctrl, and eMSC-FLT3L conditions did not affect tumor growth (see **figure 2B**, previously figure 1L) or intratumoral dendritic cell infiltration (see **figure 3C**, previously figure 2C). Here, we found that eMSC-FLT3L-CXCL9-CCL5 but not eMSC-FLT3L increase cDC1 density within the tumor. This highlights a cooperativity between chemokines enabling pre-cDC1 recruitment (see **figure 7D-E**, previously figure 6D-E) and growth factor signaling pathway FLT3L/FLT3 enabling survival and proliferation.

Figure 7: Intratumoral delivery of eMSC co-expressing FLT3L, CXCL9 and CCL5 stimulates cDC1 infiltration in mice and humanized mice.

(F) Experimental design to evaluate cDC1 recruitment within tumors at a late time point. On day 9, 11 and 14 after B16-OVA injection, eMSCs were i.t. injected. Tumors were harvested on day 17 and analyzed by flow cytometry.

(G-H) Quantification by flow cytometry of the absolute number of cDC1/g tumor (G) and their frequency in CD45⁺ live cells (H) at day 15, after 3 intratumoral injections of eMSC-FLT3L-CXCL9-CCL5. Results are shown as fold change to control (Untreated). (n=16-17 mice for the untreated and eMSC-FLT3L-CXCL9-CCL5 groups, n=5 mice for the eMSC-FLT3L group, three independent experiments, *p < 0.05, ***p < 0.001, ****p < 0.0001, one-way ANOVA with Tukey's multiple comparisons test .

Sup Figure 7: eMSC-FLT3L-CXCL9-CCL5 are localized at the border of the tumor and attract hucDC2s in synthetic niches in reconstituted BRGSF mice.

(A) Quantification by flow cytometry of the mean fluorescence intensity (MFI) of CD40 and CD86 at day 15, after 3 intratumoral injections of eMSC-FLT3L-CXCL9-CCL5. Results are shown as fold change to control (Untreated). (n=16-17 mice for the

untreated and eMSC-FLT3L-CXCL9-CCL5 groups, n=5 mice for the eMSC-FLT3L group, three independent experiments, ***p < 0.001, one-way ANOVA with Tukey's multiple comparisons test).

To further test this model, we have performed novel confocal imaging on explanted live tissue section in Xcr1cre x ROSA^{ltd}TOMATO mice, that received tumors and eMSC-FLT3L vs eMSC-FLT3L-CXCL9-CCL5 immunotherapies. As shown in **figure 7I-J**, we found that chemokine expression by stromal cells significantly increase the interactions between eMSCs-FLT3L and cDC1s. We conclude that chemokine gradients organized by eMSCs lead to the efficient attraction of cDC1s at cellular sources of membrane bound FLT3L. This represents, we believe, an important addition to the study.

Figure 7: Intratumoral delivery of eMSC co-expressing FLT3L, CXCL9 and CCL5 stimulates cDC1 infiltration in mice and humanized mice.

(I) Representative immunofluorescence images of YUMM-OVA tumors injected in XCR1^{tdTomato} mice and treated with eMSC-FLT3L or eMSC-FLT3L-CXCL9-CCL5. The images shown here represent masks for cDC1s (red) and eMSCs (yellow), generated from the maximum intensity projection using ImageJ.

(J) Quantification of the number of cDC1s surrounding an eMSC, manually determined. (one experiment, two-tailed Mann-Whitney test, ****p < 0.0001).

Figure 6L-O: highly appreciated, but probably eMSC-FLT3L would have been the appropriate control here? I understand this is very challenging to repeat, but could the authors explain their choice of control here?

We thank the reviewer for this question, and we agree that including an eMSC-FLT3L control group would have further strengthened the experiment. However, given the

technical difficulty and resource intensity of this model, additional replicates were not feasible. Importantly, as shown in **figure 3C** (previously figure 2C), both eMSC-Ctrl and eMSC-FLT3L exhibited comparable capacities to enhance intratumoral dendritic cell frequencies. Therefore, we used eMSC-Ctrl as the control condition in this experiment.

Finally, the majority of experiments have only been performed once. Every experiment (especially in vivo tumour-treatments) have to be performed 2 independent times to generate robust and reproducible data.

We agree with this comment. Nonetheless, the key findings of the manuscript are supported by experiments that were independently repeated several times, confirming the robustness and reproducibility of the results. Experiments corresponding to the main figures that were initially performed once have since been repeated, and the pooled data are now presented. For example, the quantification of intratumoral and tumor-draining lymph node–associated cDC1 and cDC2 numbers and phenotypes at day 10 (**see figure 3C,4C and supplementary figure S3A**, previously figure 2C, 3C and S2A), were confirmed in independent experiments.

Figure 3: Intratumoral delivery of eMSC-FLT3L and poly(I:C) stimulates the infiltration of cDC1s, that are required for immunotherapy efficiency.

(C) Quantification of the absolute number of intra-tumoral cDC1/g tumor and the mean fluorescence intensity (MFI) of CD40 and CD86. Results are shown as fold change to control (eMSC-Ctrl). (n=6-7 mice per group, two independent experiments for day 10, n=8-9 mice per group, two independent experiments for day 15, *p < 0.05, **p < 0.01, ***p < 0.001, ****p < 0.0001, one-way ANOVA-test with Tukey's multiple comparisons).

Sup Figure 3: Intratumoral delivery of eMSC-FLT3L and poly(I:C) stimulates the infiltration of cDC2s in the tumor.

(A) Quantification of the absolute number of intratumoral cDC2/g tumor and the mean fluorescence intensity (MFI) of CD40 and CD86. Results are shown as fold change to control (eMSC-Ctrl). (n=6-7 mice per group, two independent experiments for day 10, n=8-9 mice per group, two independent experiments for day 15, *p < 0.05, **p < 0.01, one-way ANOVA-test with Tukey's multiple comparisons).

Figure 4: Intratumoral delivery of eMSC-FLT3L and poly(I:C) increase mature migratory cDC1s in tumor-draining lymph nodes and enables cross-priming of tumor-specific CD8⁺ T cells.

(C) Quantification of the absolute number of migratory cDC1s and the mean fluorescence intensity (MFI) of CD40 and CD86. Results are shown as fold change to control (eMSC-Ctrl). (n=7-8 mice per group, two independent experiments, *p < 0.05, ***p < 0.001, ****p < 0.0001, one-way ANOVA with Tukey's multiple comparisons test).

The involvement of cDC1s in driving the eMSC-FLT3L + poly(I:C) immunotherapy was also confirmed (see figure 3D-E, previously figure 2D-E).

Figure 3: Intratumoral delivery of eMSC-FLT3L and poly(I:C) stimulates the infiltration of cDC1s, that are required for immunotherapy efficiency.

(D) Tumor growth curves of XCR1^{DTA} mice or C57BL/6 WT mice treated or not with eMSC-FLT3L + poly(I:C). (A line represents the mean and SD is shown with the colored dotted lines, n=4 mice XCR1^{DTA} - No therapy and n=8 mice for the three other groups, representative of two independent experiments, ****p < 0.0001 two-way ANOVA-test with Tukey's multiple comparisons).

(E) Survival curves. (n=10-16 mice per group, two independent experiments, *p < 0.05, log-rank (Mantel-Cox) test).

Furthermore, instead of showing only representative plots from independent experiments, we have pooled the data from individual experiments when applicable. For example, in **Figure 4I** (previously figure 3H), we now present the combined absolute numbers of tetramer-positive CD8⁺ T cells in the tumor-draining lymph nodes at day 15.

Figure 4: Intratumoral delivery of eMSC-FLT3L and poly(I:C) increase mature migratory cDC1s in tumor-draining lymph nodes and enables cross-priming of tumor-specific CD8⁺ T cells.

(I) Quantification (I) of OVA-specific CD8⁺ Tetramer⁺ T cells at day 15. Results are shown as fold change to control (eMSC-Ctrl). (n=8-9 mice per group, two independent experiments, **p < 0.01, Kruskal-Wallis test with Dunn's multiple comparisons).

The figure legends have been updated to indicate the number of experimental repeats and the number of independent mice analyzed.

Major comment 3 – some conclusions should be revised and may require further testing

The unique induction of activated and non-exhausted CD8⁺ T cells in the eMSC-FLT3L+ PolyI:C-treated group is not clear, because a lot of data from the other groups are omitted/not shown. Also, it is not clear if the efficacy of eMSC-FLT3L+ PolyI:C treatment is CD8⁺ T cell dependent or Treg cell dependent. Please consider the following for a better understanding of the effects of the therapy:

We agree with the comment of the reviewer. Here is a detailed response to the questions.

Figure 3F – please add a quantification of all experimental groups – not only the untreated and combo.

We thank the reviewer for this valuable comment. In response, we have repeated experiments and have added the relevant controls. Quantifications for all experimental groups (eMSC-Ctrl, eMSC-FLT3L, poly(I:C), and eMSC-FLT3L + poly(I:C)) have been added to the **figure 4D-E** and previous figure quantifying the untreated and poly(I:C) groups only (previously figure 3D-F) have been moved to the **supplementary figure S4A-B**. We conclude that the effect we observe is indeed uniquely induced by the eMSC-FLT3L + poly(I:C) immunotherapy in our model.

Figure 4: Intratumoral delivery of eMSC-FLT3L and poly(I:C) increase mature migratory cDC1s in tumor-draining lymph nodes and enables cross-priming of tumor-specific CD8⁺ T cells.

(D) Representative flow cytometry plots of CD8 T cells in the tumor-draining lymph node at day 15.

(E) Quantification of the absolute number of activated, exhausted and stem-like CD8 T cells. Results are shown as fold change to control (eMSC-Ctrl). (n=12-14 mice per group, three independent experiments, **p < 0.01, ***p < 0.001, Kruskal-Wallis test with Dunn's multiple comparisons. n=4 mice per group, one experiment, **p < 0.01, Kruskal-Wallis test with Dunn's multiple comparisons).

Figure 3G – please add a quantification of Tregs and IFN γ -producing CD8⁺ T cells for all experimental groups – not only the ratio with CD8 T cells

This quantification was already shown in the previous version of the manuscript and can now be found in the **supplementary figure S4C-D** (previously figure 3A-B).

Sup Figure 4: Intratumoral delivery of eMSC-FLT3L and poly(I:C) increase proliferative stem-like CD8⁺ T cells, IFN γ ⁺CD8⁺ and IFN γ ⁺CD4⁺ but not T regulatory cells.

(C-D) Quantification of the absolute number of IFN γ ⁺CD8⁺ T cells **(C)**, CD4⁺Foxp3⁺ T regulatory cells (Tregs) **(D)** at day 15. (n=5 mice per group in one experiment, *p < 0.05, one-way ANOVA-test with Dunnett's multiple comparison test).

In Figure 4K-N, CD8 blockade should be tested alone to confirm an involvement of CD8 T cells. Also, to distinguish total CD4 from Treg cells, please block Tregs specifically using Foxp3-ko mice or a blocking antibody (for example anti-CD25).

We thank the reviewer for this insightful comment and, as suggested by the reviewer, we performed additional experiments to address this. By selectively depleting CD8⁺ T cells, we confirmed their involvement in the therapeutic efficacy. In contrast, depletion of all CD4⁺ T cells or regulatory T cells (Tregs) alone did not significantly affect tumor growth (see **supplementary figure 5E**) or survival (see **supplementary figure 5F**), indicating that the efficacy of this therapy is primarily CD8⁺ T cell dependent.

Sup Figure 5: Intratumoral delivery of eMSC-FLT3L and poly(I:C) stimulates infiltration and proliferation of CD4+ T cells but not of T regulatory cells.

(E) Tumor growth curves. (n=7 mice per group, one experiment, ****p < 0.0001, two-way ANOVA-test with Tukey's multiple comparisons).

(F) Survival curves. (n=7 mice per group, one experiment, *p < 0.05, log-rank (Mantel-Cox) test).

Additional comments

In their explanations on conventional DCs (cDCs) the authors should probably adhere to the currently established nomenclature as much as possible and use the abbreviations cDC1 and cDC2 (instead of DC1 and DC2) to prevent confusion for the reader.

We agree with this comment. The terms DC1 and DC2 have been replaced with cDC1 and cDC2 throughout the manuscript.

Define what "rec-huFLT3L" is. I understand recombinant human Flt3L.

Yes, rec-huFLT3L means recombinant human FLT3L. This modification has been added to the figure legend and the text in the revised manuscript.

The authors assess "Intra-tumoral delivery of eMSC-FLT3L" in Figure 1A-J. However, the experimental setup is a co-injection of tumour cells and MSCs from the beginning. The caveat of this strategy should be stated.

There seems to be a misunderstanding regarding the different experimental setups shown in the figures.

In **figure 1B–C** (previously figure 1B-C), eMSCs were co-injected with murine hematopoietic cells embedded in an extracellular matrix, but no tumor was present at that stage. This specific experiment was designed to test if eMSC-FLT3L would induce DC differentiation from co-injected DC progenitors. In contrast, in the experiment shown in figures **1D** (previously 1D), **1F**, **1I** (previously 1F) **or 2A** (previously 1K), **and later on**, tumor cells were injected first, and eMSCs were administered later, once the tumors became palpable (between 50 and 100 mm³). Thus, this setup does not correspond to a co-injection of tumor cells and eMSCs. To ensure clarity and prevent any confusion, we have added the indication “C57BL/6 bearing B16-OVA” in **figures 1D and 1F**. These panels refer to the persistence experiments, where the timeline is based on the day of eMSC injection instead of the day following B16-OVA tumor inoculation, which can be confusing. In summary, our experimental setting for immunotherapy is based on the intratumoral injection of eMSC-FLT3L in already growing tumors.

As the treatment with eMSC-FLT3L is of a transient nature, how to the authors envisage clinical translation? Please discuss and explain.

This is indeed a key issue. To answer this, we have first added additional experiments addressing the persistence and biodistribution of eMSCs using *in vivo* imaging. Nano-luciferase–expressing eMSCs were generated and monitored for 8 days in the B16-OVA model using bioluminescence imaging. We observed that eMSCs persisted within the tumor for at least 8 days, although at very low levels, and that their activity remained restricted to the tumor site. These findings suggest that, once injected into the tumor, eMSCs do not disseminate throughout the body but rather remain localized. These data have been added to the **figure 1F-H**.

Figure 1: Intratumoral delivery of engineered mesenchymal stromal cells expressing membrane bound FLT3L does not alter intratumoral and systemic cDC1 populations.

(F) Experimental design to evaluate the biodistribution and persistence of eMSCs within B16-OVA tumors. 1×10^6 eMSC-NanoLuc were injected *i.t.* on day 0. On days 0 (4h after the injection), 4, and 8, mice received an intravenous injection of $0.44 \mu\text{moles}$ of the fluorofurimazine substrate, followed by bioluminescence imaging using the IVIS system.

(G) Bioluminescence imaging and quantification of the average radiance of eMSC-NanoLuc at days 0, 4 and 8 following their injection.

(H) Bioluminescence imaging and quantification of the average radiance of eMSC-NanoLuc in individual organs of mice having received eMSC-NanoLuc. He:heart; Li:liver; Ki:kidney, Sp:spleen, Lu:lungs, Tu:tumor, tdLN:tumor-draining lymph node, Mu:muscle.

From this experiment, we conclude that engraftment of eMSCs leads to persistence over 8 days and we hope that optimization of engraftment methods might extend this persistence.

In addition, as this point is highly important in the study, the discussion has been extensively modified, highlighting practical and technological bottleneck in the road to translate the system to an immunotherapeutic intervention in clinical settings. Please, find below in green the revised paragraphs, taking this comment into account.

“The prospect of locally implanted, eMSCs acting as versatile, sustained factories for therapeutic molecule delivery is highly attractive for cancer immunotherapy. This is particularly relevant for chemokines, which generate spatially organized gradients governing leukocyte trafficking and tissue compartmentalization. Nonetheless, several technological and translational challenges must be addressed to realize the clinical potential of eMSCs.

First, standardized and GMP-compatible protocols for autologous eMSCs isolation, engineering, and production must be established. Such methods should ensure reproducible recovery of stromal populations of controlled complexity that are amenable to *ex vivo* expansion, viral transduction, and post-transduction enrichment. Although the present study primarily employed murine embryonic fibroblasts as an MSC source, we demonstrate that adipose-derived stromal cells can be expanded, transduced, and used to generate immunologically competent eMSCs. Other groups have successfully implemented bone marrow-derived eMSCs for immunotherapy (Stagg et al. 2004; Ren et al. 2008; ulach et al. 2021; Zhang et al. 2022). Validation in human systems will be critical for clinical translation. Furthermore, the phenotypic heterogeneity resulting from *ex vivo* expansion and transduction remains to be fully characterized. Despite homogeneous transgene expression (e.g., via IRES-linked fluorescent reporters), we can not exclude that this cellular preparation is functionally heterogeneous and that only a subset of eMSCs may be functionally active *in vivo*. Integration of single-cell RNA sequencing and spatial transcriptomics in eMSC-treated tumors could provide essential insight into stromal diversity, fate, and functional states.

Second, optimization of eMSC delivery methods could improve therapeutic persistence and efficacy. In our current model, intratumoral injection of cell suspensions resulted in transient eMSC persistence, with only a minor fraction detectable beyond ten days. Future studies should aim at enhancing eMSC residency through tissue engineering strategies, including synthetic scaffolds or extracellular matrix integration (Choi et al. 2017). Cell-intrinsic manipulation of stress pathway might also offer some opportunities to improve the survival of eMSC in the TME. For instance, NQO1 expression might improve resistance to oxidative stress (Bae et al. 2022). Would these manipulations improve persistence, it would be important to determine whether prolonged eMSC survival improves T and NK cell-dependent tumor control. While intratumoral delivery provides localized control, it may not be feasible for all cancer types. Developing systemic delivery approaches for eMSCs could extend therapeutic reach to deep or inaccessible lesions while improving patient tolerability. Indeed, intravital imaging studies have shown that intravenously administered MSCs exhibit intrinsic tropism for sites of tissue remodeling, including wounds and solid

tumors (Kidd et al. 2009; Kalimuthu et al. 2017). The molecular mechanisms underlying this selective homing remain poorly defined but could be exploited to deliver eMSCs to otherwise unreachable tumor sites.

Third, eMSC-based therapies designed to stimulate DC-dependent antitumor immunity must also incorporate cues that promote antigen release and DC maturation, the two essential steps required to initiate the cancer-immunity cycle. In our study, eMSCs expressing FLT3L and CXCL9/CCL5 effectively enhanced local DC infiltration but did not efficiently drive DC maturation, potentially limiting migration to tumor-draining lymph nodes and cross-priming of CD8⁺ T cells. Embedding maturation-inducing signals within eMSCs may overcome this limitation. In parallel, promoting local tumor cell death could enhance antigen availability for DC uptake. One promising avenue involves combining eMSCs with oncolytic virotherapy, which induces immunogenic cell death and antigen release. eMSCs have been widely explored as delivery vehicles for oncolytic viruses, leveraging their tumor-homing capacity (Pereboeva and Curiel 2004; Hakkarainen et al. 2007). This approach has been implemented for several viral platforms, including herpes simplex virus (Du et al. 2017), myxoma virus (Josiah et al. 2010), Newcastle disease virus (Kazimirsky et al. 2016), measles virus (Sonabend et al. 2008), and adenovirus (Sonabend et al. 2008; Yoon et al. 2019). We propose that eMSC-based oncolytic virotherapy could be synergistically combined with DC-targeting eMSCs, as suggested by Svensson-Arvelund *et al.*, who observed cooperation between recombinant FLT3L administration and NDV-mediated oncolysis (Svensson-Arvelund et al. 2022). Moreover, type I IFN-releasing eMSCs could complement DC-directed eMSCs to further enhance local immune activation (Ren et al. 2008; Choi et al. 2017; Zhang et al. 2022).

In summary, while technical and regulatory limitations remain, the translational potential of eMSC-based immunotherapies is considerable. Their inherent tumor tropism, compatibility with *ex vivo* genetic engineering, and capacity for localized, sustained delivery make eMSCs a promising cellular platform for next-generation combination immunotherapies aimed at amplifying both innate and adaptive antitumor immunity.”

Figure 4 is missing panel M

This error has been corrected.

Figure 4D – add percentages to representative FACS plot please

Percentages have been added to the representative FACS plots (see supplementary figure S5A).

Results from Figure 4H-J should be reinterpreted: it is clear that tumour-experienced mice will develop a T cell memory and it is not clear how much the therapy contributed to that. For confirmation, tumour-resected mice would be necessary. And many more markers than displayed in Figure 4J would be necessary to confirm a resident-memory phenotype. I do not think this is a necessary message of the manuscript, but please omit any statements like “We conclude that eMSC-FLT3L + poly(I:C) immunotherapy induce local, systemic and long- term immunity against tumor antigens in the CD8⁺ T cell compartment” throughout the manuscript.

We agree with the comment and have rephrased accordingly to distinguish between the induction of CD8⁺ T cells and actual anti-tumor immunity, as follows: “We conclude that eMSC-FLT3L + poly(I:C) immunotherapy induces a sustained memory CD8⁺ T cell response.”

Figure 7D-F: could the authors also show non-transferred, endogenous T cell infiltration into tumours of this experiment? That would be more relevant than quantifying adoptively transferred T cells.

The goal of this experiment was to selectively address if eMSC-derived chemokines would be able to increase the recruitment of activated T cells from the blood stream. To address this, it was important to selectively track blood derived activated T cells post intra-veinous transfer. Nonetheless and as requested, we have added the requested quantifications of endogenous T cell accumulation in the figure **8G-H**.

Figure 8: Intratumoral delivery of eMSC co-expressing FLT3L, CXCL9 and CCL5 stimulates the infiltration of activated CD4⁺ and CD8⁺ T cells.

(G-H) Quantification of the absolute number of endogenous CD44⁺CD4⁺ T cells/g tumor (G) and endogenous CD44⁺CD8⁺ T cells/g tumor (H). (n=5 mice per group, one experiment, one-way ANOVA test with Dunnett's multiple comparisons).

Figure 7G-H: at day 17, tumours already have a significantly different size (according to Figure 8). Hence, the differences in T cell infiltration could be a consequence rather than a cause. T cell infiltration into tumours should be analyzed at an earlier time-point, when tumours have equal sizes between groups.

This comment would be valid in absolute numbers. However, we have expressed results in absolute numbers normalized by tumor weight. This method enables to quantify T cell accrual independently of tumor size. In addition, kinetic constraints guided the choice of this time point. At this early time point, an immune response within the tumor is unlikely to be detectable. The sequence of events, from dendritic cell activation, migration into the tumor, and T-cell priming, to the subsequent infiltration of activated T cells, requires sufficient time, by which point tumors may already differ in size between groups.

The authors state: From these results, we conclude that the recruitment of DC1 induced by eMSC-FLT3L-CXCL9-CCL5 is independent on B, T and N /ILCs cells but dependent on FLT3 signalling. Yet, FLT3L overexpression alone is not sufficient to induce cDC1 recruitment by itself. This is puzzling and the authors should comment on this. To decipher this and understand if Flt3L is actually required (but not sufficient), the authors could consider to show if eMSC-CXCL9-CCL5 (without FLT3L expression) are able to recruit cDC1s to tumours (but that is not vital for the manuscript).

We think that eMSC-FLT3L-CXCL9-CCL5 is not working in FLT3ko mice because FLT3 deficiency is known to impact on the amount of bone marrow and circulating pre-cDC1s. Therefore, the interesting experiment suggested by the reviewer should be done in WT, FLT3 proficient mice. We predict that chemokines should lead to pre-cDC1s recruitment in tissues regardless of their local expansion. (Although it might be very difficult to quantify given the rarity of these cells, **see figure 7E**, previously figure 6E).

Figure 8E-F: a tumor model that is less susceptible to aPD1/aCTLA4 blockade should be used to evaluate synergy with eMSC-FLT3L-CXCL9-CCL5 treatment. I don't think this is necessary for the manuscript and the data could be removed.

Other reviewers didn't ask for it to be removed, therefore we have left it and wait for editor instructions regarding this specific topic.

REFERENCES

- Bae, Joonbeom, Longchao Liu, Casey Moore, et al. 2022. "IL-2 Delivery by Engineered Mesenchymal Stem Cells Re-Invigorates CD8+ T Cells to Overcome Immunotherapy Resistance in Cancer." *Nature Cell Biology* 24 (12): 1754–65. <https://doi.org/10.1038/s41556-022-01024-5>.
- Blake, Stephen J., Jane James, Feargal J. Ryan, et al. 2021. "The Immunotoxicity, but Not Anti-Tumor Efficacy, of Anti-CD40 and Anti-CD137 Immunotherapies Is Dependent on the Gut Microbiota." *Cell Reports Medicine* 2 (12): 100464. <https://doi.org/10.1016/j.xcrm.2021.100464>.
- Böttcher, Jan P., Eduardo Bonavita, Probir Chakravarty, et al. 2018. "NK Cells Stimulate Recruitment of cDC1 into the Tumor Microenvironment Promoting Cancer Immune Control." *Cell* 172 (5): 1022-1037.e14. <https://doi.org/10.1016/j.cell.2018.01.004>.
- Chen, Jonathan H., Linda T. Nieman, Maxwell Spurrell, et al. 2024. "Human Lung Cancer Harbors Spatially Organized Stem-Immunity Hubs Associated with Response to Immunotherapy." *Nature Immunology* 25 (4): 644–58. <https://doi.org/10.1038/s41590-024-01792-2>.
- Choi, Sung Hugh, Daniel W. Stuckey, Sara Pignatta, et al. 2017. "Tumor Resection Recruits Effector T Cells and Boosts Therapeutic Efficacy of Encapsulated Stem Cells Expressing IFN β in Glioblastomas." *Clinical Cancer Research : An Official Journal of the American Association for Cancer Research* 23 (22): 7047–58. <https://doi.org/10.1158/1078-0432.CCR-17-0077>.
- Chow, Melvyn T., Aleksandra J. Ozga, Rachel L. Servis, et al. 2019. "Intratumoral Activity of the CXCR3 Chemokine System Is Required for the Efficacy of Anti-PD-1 Therapy." *Immunity* 50 (6): 1498-1512.e5. <https://doi.org/10.1016/j.immuni.2019.04.010>.
- Cook, Stuart J., Quintin Lee, Alex Ch Wong, et al. 2018. "Differential Chemokine Receptor Expression and Usage by Pre-cDC1 and Pre-cDC2." *Immunology and Cell Biology* 96 (10): 1131–39. <https://doi.org/10.1111/imcb.12186>.
- Corbett, T. H., D. P. Griswold, B. J. Roberts, J. C. Peckham, and F. M. Schabel. 1975. "Tumor Induction Relationships in Development of Transplantable Cancers of the Colon in Mice for Chemotherapy Assays, with a Note on Carcinogen Structure." *Cancer Research* 35 (9): 2434–39.

- Dangaj, Denarda, Marine Bruand, Alizée J. Grimm, et al. 2019. "Cooperation between Constitutive and Inducible Chemokines Enables T Cell Engraftment and Immune Attack in Solid Tumors." *Cancer Cell* 35 (6): 885-900.e10. <https://doi.org/10.1016/j.ccell.2019.05.004>.
- Darrasse-Jèze, Guillaume, Stephanie Deroubaix, Hugo Mouquet, et al. 2009. "Feedback Control of Regulatory T Cell Homeostasis by Dendritic Cells in Vivo." *The Journal of Experimental Medicine* 206 (9): 1853–62. <https://doi.org/10.1084/jem.20090746>.
- Di Pilato, M., R. Kfuri-Rubens, J. N. Pruessmann, et al. 2021. "CXCR6 Positions Cytotoxic T Cells to Receive Critical Survival Signals in the Tumor Microenvironment." *Cell* 184 (17): 4512-4530 e22. 34343496. <https://doi.org/10.1016/j.cell.2021.07.015>.
- Du, Wanlu, Ivan Seah, Oumaima Bougazzoul, et al. 2017. "Stem Cell-Released Oncolytic Herpes Simplex Virus Has Therapeutic Efficacy in Brain Metastatic Melanomas." *Proceedings of the National Academy of Sciences of the United States of America* 114 (30): E6157–65. <https://doi.org/10.1073/pnas.1700363114>.
- Duong, Ellen, Tim B. Fessenden, Emi Lutz, et al. 2022. "Type I Interferon Activates MHC Class I-Dressed CD11b+ Conventional Dendritic Cells to Promote Protective Anti-Tumor CD8+ T Cell Immunity." *Immunity* 55 (2): 308-323.e9. <https://doi.org/10.1016/j.immuni.2021.10.020>.
- Garris, Christopher S., Sean P. Arlauckas, Rainer H. Kohler, et al. 2018. "Successful Anti-PD-1 Cancer Immunotherapy Requires T Cell-Dendritic Cell Crosstalk Involving the Cytokines IFN- γ and IL-12." *Immunity* 49 (6): 1148-1161.e7. <https://doi.org/10.1016/j.immuni.2018.09.024>.
- Hakkarainen, Tanja, Merja Särkioja, Petri Lehenkari, et al. 2007. "Human Mesenchymal Stem Cells Lack Tumor Tropism but Enhance the Antitumor Activity of Oncolytic Adenoviruses in Orthotopic Lung and Breast Tumors." *Human Gene Therapy* 18 (7): 627–41. <https://doi.org/10.1089/hum.2007.034>.
- Hammerich, Linda, Thomas U. Marron, Ranjan Upadhyay, et al. 2019. "Systemic Clinical Tumor Regressions and Potentiation of PD1 Blockade with in Situ Vaccination." *Nature Medicine* 25 (5): 814–24. <https://doi.org/10.1038/s41591-019-0410-x>.

- Im, Se Jin, Masao Hashimoto, Michael Y. Gerner, et al. 2016. "Defining CD8+ T Cells That Provide the Proliferative Burst after PD-1 Therapy." *Nature* 537 (7620): 417–21. <https://doi.org/10.1038/nature19330>.
- Josiah, Darnell T., Dongqin Zhu, Fernanda Dreher, John Olson, Grant McFadden, and Hannah Caldas. 2010. "Adipose-Derived Stem Cells as Therapeutic Delivery Vehicles of an Oncolytic Virus for Glioblastoma." *Molecular Therapy: The Journal of the American Society of Gene Therapy* 18 (2): 377–85. <https://doi.org/10.1038/mt.2009.265>.
- Kalimuthu, Senthilkumar, Ji Min Oh, Prakash Gangadaran, et al. 2017. "In Vivo Tracking of Chemokine Receptor CXCR4-Engineered Mesenchymal Stem Cell Migration by Optical Molecular Imaging." *Stem Cells International* 2017 (1): 8085637. <https://doi.org/10.1155/2017/8085637>.
- Kazimirsky, Gila, Wei Jiang, Shimon Slavin, Amotz Ziv-Av, and Chaya Brodie. 2016. "Mesenchymal Stem Cells Enhance the Oncolytic Effect of Newcastle Disease Virus in Glioma Cells and Glioma Stem Cells via the Secretion of TRAIL." *Stem Cell Research & Therapy* 7 (1): 149. <https://doi.org/10.1186/s13287-016-0414-0>.
- Kidd, Shannon, Erika Spaeth, Jennifer L. Dembinski, et al. 2009. "Direct Evidence of Mesenchymal Stem Cell Tropism for Tumor and Wounding Microenvironments Using in Vivo Bioluminescent Imaging." *Stem Cells (Dayton, Ohio)* 27 (10): 2614–23. <https://doi.org/10.1002/stem.187>.
- Kułach, Natalia, Ewelina Pilny, Tomasz Cichoń, et al. 2021. "Mesenchymal Stromal Cells as Carriers of IL-12 Reduce Primary and Metastatic Tumors of Murine Melanoma." *Scientific Reports* 11 (1): 18335. <https://doi.org/10.1038/s41598-021-97435-9>.
- Lai, Junyun, Sherly Mardiana, Imran G. House, et al. 2020. "Adoptive Cellular Therapy with T Cells Expressing the Dendritic Cell Growth Factor Flt3L Drives Epitope Spreading and Antitumor Immunity." *Nature Immunology* 21 (8): 914–26. <https://doi.org/10.1038/s41590-020-0676-7>.
- Lee, Jaeyop, Gaëlle Breton, Thiago Yukio Kikuchi Oliveira, et al. 2015. "Restricted Dendritic Cell and Monocyte Progenitors in Human Cord Blood and Bone Marrow." *The Journal of Experimental Medicine* 212 (3): 385–99. <https://doi.org/10.1084/jem.20141442>.

- Lin, K. Y., F. G. Guarnieri, K. F. Staveley-O'Carroll, et al. 1996. "Treatment of Established Tumors with a Novel Vaccine That Enhances Major Histocompatibility Class II Presentation of Tumor Antigen." *Cancer Research* 56 (1): 21–26.
- Liu, Baolin, Xueda Hu, Kaichao Feng, et al. 2022. "Temporal Single-Cell Tracing Reveals Clonal Revival and Expansion of Precursor Exhausted T Cells during Anti-PD-1 Therapy in Lung Cancer." *Nature Cancer* 3 (1): 108–21. <https://doi.org/10.1038/s43018-021-00292-8>.
- Longhi, M. Paula, Christine Trumpfheller, Juliana Idoyaga, et al. 2009. "Dendritic Cells Require a Systemic Type I Interferon Response to Mature and Induce CD4+ Th1 Immunity with Poly IC as Adjuvant." *The Journal of Experimental Medicine* 206 (7): 1589–602. <https://doi.org/10.1084/jem.20090247>.
- Magen, Assaf, Pauline Hamon, Nathalie Fiaschi, et al. 2023. "Intratumoral Dendritic Cell-CD4+ T Helper Cell Niches Enable CD8+ T Cell Differentiation Following PD-1 Blockade in Hepatocellular Carcinoma." *Nature Medicine* 29 (6): 1389–99. <https://doi.org/10.1038/s41591-023-02345-0>.
- Mattiuz, Raphaël, Jesse Boumelha, Pauline Hamon, et al. 2024. "Dendritic Cells Type 1 Control the Formation, Maintenance, and Function of Tertiary Lymphoid Structures in Cancer." Preprint, December 27. <https://doi.org/10.1101/2024.12.27.628014>.
- Meiser, Philippa, Moritz A. Knolle, Anna Hirschberger, et al. 2023. "A Distinct Stimulatory cDC1 Subpopulation Amplifies CD8+ T Cell Responses in Tumors for Protective Anti-Cancer Immunity." *Cancer Cell* 41 (8): 1498-1515.e10. <https://doi.org/10.1016/j.ccell.2023.06.008>.
- Oba, Takaaki, Mark D. Long, Tibor Keler, et al. 2020. "Overcoming Primary and Acquired Resistance to Anti-PD-L1 Therapy by Induction and Activation of Tumor-Residing cDC1s." *Nature Communications* 11 (1): 5415. <https://doi.org/10.1038/s41467-020-19192-z>.
- Pereboeva, Larisa, and David T. Curiel. 2004. "Cellular Vehicles for Cancer Gene Therapy." *BioDrugs* 18 (6): 361–85. <https://doi.org/10.2165/00063030-200418060-00003>.
- Ren, C., S. Kumar, D. Chanda, et al. 2008. "Cancer Gene Therapy Using Mesenchymal Stem Cells Expressing Interferon-Beta in a Mouse Prostate

- Cancer Lung Metastasis Model.” *Gene Therapy* 15 (21): 1446–53. <https://doi.org/10.1038/gt.2008.101>.
- Ryu, Chung Heon, Sang-Hoon Park, Soon A. Park, et al. 2011. “Gene Therapy of Intracranial Glioma Using Interleukin 12-Secreting Human Umbilical Cord Blood-Derived Mesenchymal Stem Cells.” *Human Gene Therapy* 22 (6): 733–43. <https://doi.org/10.1089/hum.2010.187>.
- Salmon, H el ene, Juliana Idoyaga, Adeeb Rahman, et al. 2016. “Expansion and Activation of CD103(+) Dendritic Cell Progenitors at the Tumor Site Enhances Tumor Responses to Therapeutic PD-L1 and BRAF Inhibition.” *Immunity* 44 (4): 924–38. <https://doi.org/10.1016/j.immuni.2016.03.012>.
- Salomon, Ran, Hagar Rotem, Yonatan Katzenelenbogen, et al. 2022. “Bispecific Antibodies Increase the Therapeutic Window of CD40 Agonists through Selective Dendritic Cell Targeting.” *Nature Cancer* 3 (3): 287–302. <https://doi.org/10.1038/s43018-022-00329-6>.
- S anchez-Paulete, Alfonso R.,  lvaro Teijeira, Jos e I. Quetglas, et al. 2018. “Intratumoral Immunotherapy with XCL1 and sFlt3L Encoded in Recombinant Semliki Forest Virus-Derived Vectors Fosters Dendritic Cell-Mediated T-Cell Cross-Priming.” *Cancer Research* 78 (23): 6643–54. <https://doi.org/10.1158/0008-5472.CAN-18-0933>.
- Schenkel, Jason M., Rebecca H. Herbst, David Canner, et al. 2021. “Conventional Type I Dendritic Cells Maintain a Reservoir of Proliferative Tumor-Antigen Specific TCF-1+ CD8+ T Cells in Tumor-Draining Lymph Nodes.” *Immunity* 54 (10): 2338-2353.e6. <https://doi.org/10.1016/j.immuni.2021.08.026>.
- See, Peter, Charles-Antoine Dutertre, Jinmiao Chen, et al. 2017. “Mapping the Human DC Lineage through the Integration of High-Dimensional Techniques.” *Science (New York, N.Y.)* 356 (6342): eaag3009. <https://doi.org/10.1126/science.aag3009>.
- Siddiqui, Imran, Karin Schaeuble, Vijaykumar Chennupati, et al. 2019. “Intratumoral Tcf1+PD-1+CD8+ T Cells with Stem-like Properties Promote Tumor Control in Response to Vaccination and Checkpoint Blockade Immunotherapy.” *Immunity* 50 (1): 195-211.e10. <https://doi.org/10.1016/j.immuni.2018.12.021>.
- Siwicki, Marie, Nicolas A. Gort-Freitas, Marius Messemaker, et al. 2021. “Resident Kupffer Cells and Neutrophils Drive Liver Toxicity in Cancer Immunotherapy.”

Science Immunology 6 (61): eabi7083.
<https://doi.org/10.1126/sciimmunol.abi7083>.

Sonabend, Adam M., Ilya V. Ulasov, Matthew A. Tyler, Angel A. Rivera, James M. Mathis, and Maciej S. Lesniak. 2008. "Mesenchymal Stem Cells Effectively Deliver an Oncolytic Adenovirus to Intracranial Glioma." *Stem Cells (Dayton, Ohio)* 26 (3): 831–41. <https://doi.org/10.1634/stemcells.2007-0758>.

Spranger, Stefani, Daisy Dai, Brendan Horton, and Thomas F. Gajewski. 2017. "Tumor-Residing Batf3 Dendritic Cells Are Required for Effector T Cell Trafficking and Adoptive T Cell Therapy." *Cancer Cell* 31 (5): 711-723.e4. <https://doi.org/10.1016/j.ccell.2017.04.003>.

Stagg, John, Laurence Lejeune, André Paquin, and Jacques Galipeau. 2004. "Marrow Stromal Cells for Interleukin-2 Delivery in Cancer Immunotherapy." *Human Gene Therapy* 15 (6): 597–608. <https://doi.org/10.1089/104303404323142042>.

Svensson-Arvelund, Judit, Sara Cuadrado-Castano, Gvantsa Pantsulaia, et al. 2022. "Expanding Cross-Presenting Dendritic Cells Enhances Oncolytic Virotherapy and Is Critical for Long-Term Anti-Tumor Immunity." *Nature Communications* 13 (1): 7149. <https://doi.org/10.1038/s41467-022-34791-8>.

Swee, Lee Kim, Nabil Bosco, Bernard Malissen, Rhodri Ceredig, and Antonius Rolink. 2009. "Expansion of Peripheral Naturally Occurring T Regulatory Cells by Fms-like Tyrosine Kinase 3 Ligand Treatment." *Blood* 113 (25): 6277–87. <https://doi.org/10.1182/blood-2008-06-161026>.

Theisen, Derek J., Jesse T. Davidson, Carlos G. Briseño, et al. 2018. "WDFY4 Is Required for Cross-Presentation in Response to Viral and Tumor Antigens." *Science (New York, N.Y.)* 362 (6415): 694–99. <https://doi.org/10.1126/science.aat5030>.

Villani, Alexandra-Chloé, Rahul Satija, Gary Reynolds, et al. 2017. "Single-Cell RNA-Seq Reveals New Types of Human Blood Dendritic Cells, Monocytes, and Progenitors." *Science (New York, N.Y.)* 356 (6335): eaah4573. <https://doi.org/10.1126/science.aah4573>.

Wan, Duo, Qi Zhang, Zhenrong Yang, et al. 2025. "Engineered Oncolytic Virus OH2-FLT3L Enhances Antitumor Immunity via Dendritic Cell Activation." *Molecular Therapy Oncology* 33 (2): 200975. <https://doi.org/10.1016/j.omton.2025.200975>.

- Yoon, A.-Rum, JinWoo Hong, Yan Li, et al. 2019. "Mesenchymal Stem Cell-Mediated Delivery of an Oncolytic Adenovirus Enhances Antitumor Efficacy in Hepatocellular Carcinoma." *Cancer Research* 79 (17): 4503–14. <https://doi.org/10.1158/0008-5472.CAN-18-3900>.
- Yost, Kathryn E., Ansuman T. Satpathy, Daniel K. Wells, et al. 2019. "Clonal Replacement of Tumor-Specific T Cells Following PD-1 Blockade." *Nature Medicine* 25 (8): 1251–59. <https://doi.org/10.1038/s41591-019-0522-3>.
- Zhang, Tao, Yu Wang, Qing Li, et al. 2022. "Mesenchymal Stromal Cells Equipped by IFN α Empower T Cells with Potent Anti-Tumor Immunity." *Oncogene* 41 (13): 1866–81. <https://doi.org/10.1038/s41388-022-02201-4>.

REVIEWERS' COMMENTS

Reviewer #1:

The authors have greatly addressed each of my comments. I have no more requests nor additional clarifications.

We sincerely thank the reviewer for the positive feedback and are pleased that our revisions have satisfactorily addressed the comments.

Reviewer #2:

The authors have extensively addressed my comments and those of the other reviewers. I have no additional suggestions.

We sincerely thank the reviewer for his careful evaluation and for acknowledging the revisions. We greatly appreciate your time and constructive feedback throughout the review process.

Reviewer #3:

I would like to congratulate the authors for strongly improving the manuscript and erasing most of my comments. Additional analyses and controls strengthen the scientific soundness of the data and conclusions. Also, it is appreciated that the authors validated the efficacy of eMSC-FLT3L + poly(I:C) in 3 more grafted cancer models.

Yet, the key findings in the remaining paper are unfortunately still limited to B16-OVA. However, more importantly, in light of the new data in response to my major comment 2, the comparison of Recombinant FLT3L + poly(I:C) (which are known and previously described treatments) and eMSC-FLT3L + poly(I:C) and mono-treatments, the therapeutic improvement of using eMSC-FLT3L (over recombinant Flt3l or poly(I:C) alone) is unfortunately overall less convincing in the revised version of the manuscript. Indeed, this direct comparison was not tested in more physiological cancer models (without ectopic antigen), where the therapeutic efficacy may actually be equal. Hence, based on the provided data in Supplementary Figure 2 (that shows no significant difference between Recombinant FLT3L + poly(I:C) and eMSC-FLT3L + poly(I:C)), I am unfortunately not convinced that there is a strong therapeutic advantage of using eMSC-FLT3L or eMSC-FLT3L-CXCL9-CCL5 of the other treatment (combinations) of poly(I:C) and rec-Flt3L. In my point of view, this should be better shown experimentally in at least a second physiological cancer model (not only B16-OVA). Other advantages of eMSC-FLT3L compared with rec-Flt3L could also be more closely explored or at least explained in detail to justify the use of an adoptive transfer of cells versus a recombinant protein/adjuvant in clinical practice (e.g. eventual toxicity, Treg induction in a comparative experiment, etc) – but those experiments are just a suggestion. Erasing this last doubt would really strengthen the relevance of the presented manuscript, that otherwise is very interesting.

We sincerely thank the reviewer for the positive and encouraging feedback. We are glad that the additional analyses and controls have reinforced the robustness of our findings. Your insights were very helpful in improving the manuscript.

Regarding eMSC-FLT3L + poly(I:C) vs recombinant protein FLT3L + poly(I:C):

The data presented in Supplementary Fig. 2 show that both strategies achieve comparable tumor control. However, in terms of survival, eMSC_FLT3L + poly(I:C) demonstrates improved efficacy, with some surviving mice (vs. untreated, ****), whereas recombinant FLT3L + poly(I:C) does not result in surviving animals (vs. untreated, **). We therefore conclude that eMSC-FLT3L provides a modest but measurable advantage over recombinant FLT3L, which is further supported by the absence of systemic effects on FLT3L levels (Fig. 1J). Overall, these data demonstrate benefit–risk ratio that favors eMSC-FLT3L over recombinant FLT3L.

We agree with Reviewer 3 on the need to extend these analyses to additional tumor models, including autochthonous models. We plan to undertake these experiments in subsequent studies, which will include careful titration of recombinant FLT3L and evaluation of alternative injection schedules. For instance, intratumoral route was used for recombinant FLT3L. We would need to also compare systemic delivery route (more amenable in clinical practice).

As noted in the revised Discussion section, we believe that cell delivery methods must be carefully optimized to ensure reliable implantation of eMSC-FLT3L within tumors. As this work is beyond the scope of the current manuscript, we consider it premature to draw conclusions regarding the relative clinical utility of eMSC-FLT3L versus recombinant FLT3L. Nevertheless, we believe that the current study provides a valuable proof-of-concept demonstrating the feasibility of using eMSCs to stimulate cDC infiltration in solid tumors.